# A positive-feedback-based mechanism for constriction rate acceleration during cytokinesis in *Caenorhabditis elegans*

Renat N Khaliullin[1]*, Rebecca A Green[1], Linda Z Shi[2], J Sebastian Gomez-Cavazos[1], Michael W Berns[2], Arshad Desai[1], Karen Oegema[1]*

[1]Department of Cellular and Molecular Medicine, Ludwig Institute for Cancer Research, University of California, San Diego, San Diego, United States; [2]Department of Bioengineering and Institute of Engineering in Medicine, University of California, San Diego, San Diego, United States

**Abstract** To ensure timely cytokinesis, the equatorial actomyosin contractile ring constricts at a relatively constant rate despite its progressively decreasing size. Thus, the per-unit-length constriction rate increases as ring perimeter decreases. To understand this acceleration, we monitored cortical surface and ring component dynamics during the first cytokinesis of the *Caenorhabditis elegans* embryo. We found that, per unit length, the amount of ring components (myosin, anillin) and the constriction rate increase with parallel exponential kinetics. Quantitative analysis of cortical flow indicated that the cortex within the ring is compressed along the axis perpendicular to the ring, and the per-unit-length rate of cortical compression increases during constriction in proportion to ring myosin. We propose that positive feedback between ring myosin and compression-driven flow of cortex into the ring drives an exponential increase in the per-unit-length amount of ring myosin to maintain a high ring constriction rate and support this proposal with an analytical mathematical model.

DOI: https://doi.org/10.7554/eLife.36073.001

*For correspondence:
renatkh@gmail.com (RNK);
koegema@ucsd.edu (KO)

**Competing interests:** The authors declare that no competing interests exist.

## Introduction

During cytokinesis in animal cells, constriction of an equatorial actomyosin ring cinches the mother cell surface to generate a dumbbell-shaped structure with an intercellular bridge that connects the two daughter cells (*Fededa and Gerlich, 2012*; *Green et al., 2012*). This cellular shape change is brought about by a cortical contractile ring that assembles around the cell equator following chromosome segregation in anaphase. To ensure that each cell inherits a single genomic complement, contractile ring assembly is directed by the small GTPase RhoA. RhoA is activated in an equatorial zone (the 'Rho zone') in response to signaling by the anaphase spindle (*Green et al., 2012*; *Jordan and Canman, 2012*; *Piekny et al., 2005*) and patterns the equatorial cortex by recruiting contractile ring components from the cytoplasm (*Vale et al., 2009*; *Yumura, 2001*; *Zhou and Wang, 2008*). RhoA activates Rho kinase, which promotes the assembly and recruitment of myosin II (*Matsumura et al., 2011*) and the cytokinesis formin that assembles the long actin filaments that make up the ring (*Otomo et al., 2005*). Contractile rings also contain membrane-associated septin filaments (*Bridges and Gladfelter, 2015*) and the filament cross linker anillin (*D'Avino, 2009*; *Piekny and Maddox, 2010*). Recent work in the *Caenorhabditis elegans* embryo suggests that the equatorial cortex is compressed along the pole-to-pole axis after this initial patterning, leading to the alignment of actin filament bundles as the ring forms (*Reymann et al., 2016*). After its assembly, the ring begins to constrict in the around-the-ring direction. Constriction is thought to be coupled to

the progressive disassembly of the ring (i.e. loss of components in proportion to reduction in length; *Carvalho et al., 2009*; *Murrell et al., 2015*; *Schroeder, 1990*).

Ring constriction must complete within a short cell cycle window during mitotic exit (*Canman et al., 2000*; *Martineau et al., 1995*; *Straight et al., 2003*). Timely constriction relies on the conserved ability of contractile rings to maintain a relatively constant overall closure rate despite their progressively decreasing perimeter (*Biron et al., 2004*; *Bourdages et al., 2014*; *Calvert et al., 2011*; *Carvalho et al., 2009*; *Ma et al., 2012*; *Mabuchi, 1994*; *Pelham and Chang, 2002*; *Zumdieck et al., 2007*). This property implies that the per-unit-length constriction rate increases as the rings get smaller.

Here, we explore the mechanisms underlying constriction rate acceleration during the first division of the *C. elegans* embryo. Since the contractile ring is an integral part of the larger cell cortex, we investigated the possibility that interactions between the ring and surrounding cortex contribute to constriction rate acceleration. By generating a 4D map of cortical surface dynamics, we show that the surface area of the cortex at the cell poles expands in response to the tension generated by the constricting ring to provide the increased cortical surface area required to generate the daughter cells. The ability of the polar cortex to expand in response to tension also allows ring myosin to compress cortical surface along the pole-to-pole axis perpendicular to the ring, leading to a continuous flow of cortical surface into the ring as it constricts. We show that, per unit length, the amount of ring components (myosin, anillin), the constriction rate, and the rate of cortical surface compression within the ring all increase with parallel exponential kinetics, suggesting control by positive feedback. Based on our experimental observations, we propose that feedback between ring myosin and compression-driven flow of cortex into the ring (Compression Feedback) drives the exponential increase in the per-unit-length amount of ring myosin, which in turn increases the per-unit-length constriction rate to keep the overall constriction rate high as the ring closes. We further show that an analytical mathematical formulation of the proposed feedback can recapitulate the observed dynamics of the ring during constriction.

## Results

### The cortex at the cell poles expands in response to tension generated by the constricting ring without limiting the constriction rate

During the first division of the *C. elegans* embryo, the surface area of the cell increases by ~40% to accommodate the shape change that generates the daughter cells. Work in multiple systems has shown that the molecular components of the cortex, cortex-associated granules in the cytoplasm, and particles adhered to cell surface receptors all move in a coordinated fashion during cytokinesis (*Cao and Wang, 1990*; *Dan, 1954*; *Dan and Dan, 1940*; *Dan et al., 1938*; *DeBiasio et al., 1996*; *Fishkind et al., 1996*; *Hird and White, 1993*; *Reymann et al., 2016*; *Swann and Mitchison, 1958*; *Wang et al., 1994*). In a classic set of experiments, Dan and colleagues monitored the pattern of cortical surface expansion and compression during cytokinesis by measuring the distance between surface adhered particles and/or pigmented cortex-associated cytoplasmic granules in sea urchin embryos. This analysis revealed that ring constriction is accompanied by a wave of cortical surface expansion that initiates at the cell poles (*Dan et al., 1938*; *Dan and Ono, 1954*; *Dan et al., 1937*; *Swann and Mitchison, 1958*). Note that expansion is an increase in cortical surface area and compression is reduction in cortical surface area. These terms do not imply a specific mechanism for how the increase or reduction occurs. For example, expansion can be accompanied by deposition of new cortical components that allow the cortex to maintain the same density, or by stretching and thinning of the existing cortex; similarly, compression can be accompanied by an increase in the density of cortical components or by their disassembly. Also, note that the analysis of cell surface dynamics described above refers to the movement, expansion and compression of the cortex and associated structures. How deposition of plasma membrane, the fluid lipid layer that overlies the cortex, is controlled and where it occurs is a separate question that we will not address here.

Although the experiments performed by Dan and colleagues (*Dan et al., 1938*; *Dan and Ono, 1954*; *Dan et al., 1937*; *Swann and Mitchison, 1958*) in sea urchin embryos provided a rough map of where expansion occurs, they did not allow quantification of the extent of change in cortical surface area or provide a map of cortical surface movements. To generate a quantitative map of cortical

surface dynamics during the first division of the *C. elegans* embryo, we employed an updated version of the classical approach in which we used myosin foci, rather than surface adhered particles, as fiduciary marks. We imaged the cortex at high time resolution (2 s intervals, cyan box in *Figure 1A*, *Video 1*) in embryos expressing a GFP fusion with the heavy chain of non-muscle myosin II (NMY-2; hereafter myosin::GFP; *Figure 1—figure supplement 1A,B*). In addition to its RhoA-dependent enrichment in the contractile ring, myosin is present in small puncta, distributed over the entire cortex, that flow together with actin filaments (LifeAct::mKate2, *Figure 1—figure supplement 1C*), validating their utility as fiduciary marks for monitoring cortical movements. To temporally and spatially align data collected in different embryos, ring constriction was also monitored at lower time resolution in the same embryos (36 s intervals, *Figure 1A*, *Figure 1—figure supplement 2*). Because the contractile ring closes asymmetrically within the division plane (*Maddox et al., 2007*; *Figure 1A*, *Figure 1—figure supplement 2*), cortical dynamics are not cylindrically symmetric. Therefore, we generated an average 4D map of cortical movement by computationally combining data from 93 embryos imaged in random rotational orientations (*Figure 1A*, *Figure 1—figure supplement 2*). We defined the top of the embryo as the side where the furrow ingresses first, the bottom as the opposite side, and referenced positions around the embryo circumference by the angle θ relative to the initial ingression axis (*Figure 1A*). For temporal alignment, we fit a line to normalized ring size ($\bar{R} := R/R_{emb}$) versus time between 30 and 80% closure for each embryo and extrapolated this line to 1 and 0 to define $t_0$ (cytokinesis onset) and $t_{CK}$ (time of cytokinesis), respectively (*Figure 1A*, *Figure 1—figure supplement 2*). Cortical movement could not be monitored in the division plane, because it is hidden inside the cell, or at the cell poles, due to their high curvature. Thus, this approach provided a quantitative picture of cortical movement in the central 2/3 of the embryo, with the exception of the division plane, throughout cytokinesis (*Figure 1B*; *Video 2*).

The 4D map allowed us to determine where cortical surface expansion occurs as the ring closes in the *C. elegans* embryo. Prior work monitoring the movement of surface adhered particles indicated that surface expansion occurs at the poles and immediately behind the contractile ring in sea urchin and *Xenopus* embryos, respectively (*Bluemink and de Laat, 1973*; *Byers and Armstrong, 1986*; *Danilchik et al., 2003*; *Gudejko et al., 2012*; *Selman and Perry, 1970*; *Swann and Mitchison, 1958*). In addition to these two patterns, we also considered the possibility that the cortex could expand uniformly, an assumption in mathematical models of cytokinesis (*Turlier et al., 2014*; *Zumdieck et al., 2007*). Each of these three patterns predicts a different profile for the Anterior-Posterior (AP) component of cortical velocity along the embryo. For uniform surface expansion, a gradient of velocities is predicted, where the velocity of the cortex immediately behind the ring is equal to the velocity of furrow ingression and the velocity decreases linearly as you move towards the cell poles. For surface expansion immediately behind the ring, no cortical movement is predicted on the observable embryo surface. If surface expansion is limited to the poles, the cortical velocity is predicted to be constant within the flow map region (*Figure 1—figure supplement 3*). The cortical velocity profile measured from the flow map indicated that the cortical surface at the cell poles expands as the ring constricts, whereas the cortex in the region between the poles and the division plane flows at constant velocity towards the division plane, without expansion or compression (*Figure 1B*). Note that the apparent velocity gradient that spans the division plane in the flow maps (*Figure 1B*, *dashed regions on velocity curves*) is a projection artifact due to the fact that the cortical surface turns to flow inwards as it approaches the furrow from either side (component of the velocity in the x-y plane decreases as the velocity vectors turn inwards). As expected, based on the asymmetric closure of the contractile ring within the division plane, the velocity of cortical flow was higher on the top of the embryo during the first half of cytokinesis when the furrow ingresses from the top (*Figure 1B*, black traces) and became higher on the bottom of the embryo towards the end when the furrow ingresses from the bottom (*Figure 1B*, grey traces; *Video 2*).

Cutting the cortex parallel to the division plane using a laser revealed that the cortex is under tension during cytokinesis (*Figure 2A*). To determine if cortical tension limits the constriction rate, we assayed the effect of the cortical cuts on ring closure. Cortical cuts spanning the visible area of cortex on the anterior side of the embryo (~10 μm in length) were made parallel to the division plane when the ring was at ~50% closure, and the effect on contractile ring closure rate was assessed by measuring the difference in ring sizes immediately before and 13 s after the cut. The cortical opening resulting from the ablation was approximately 35 μm²; if the cortical surface tension is the dominant force limiting the ring closure rate, this size opening would be expected to increase the

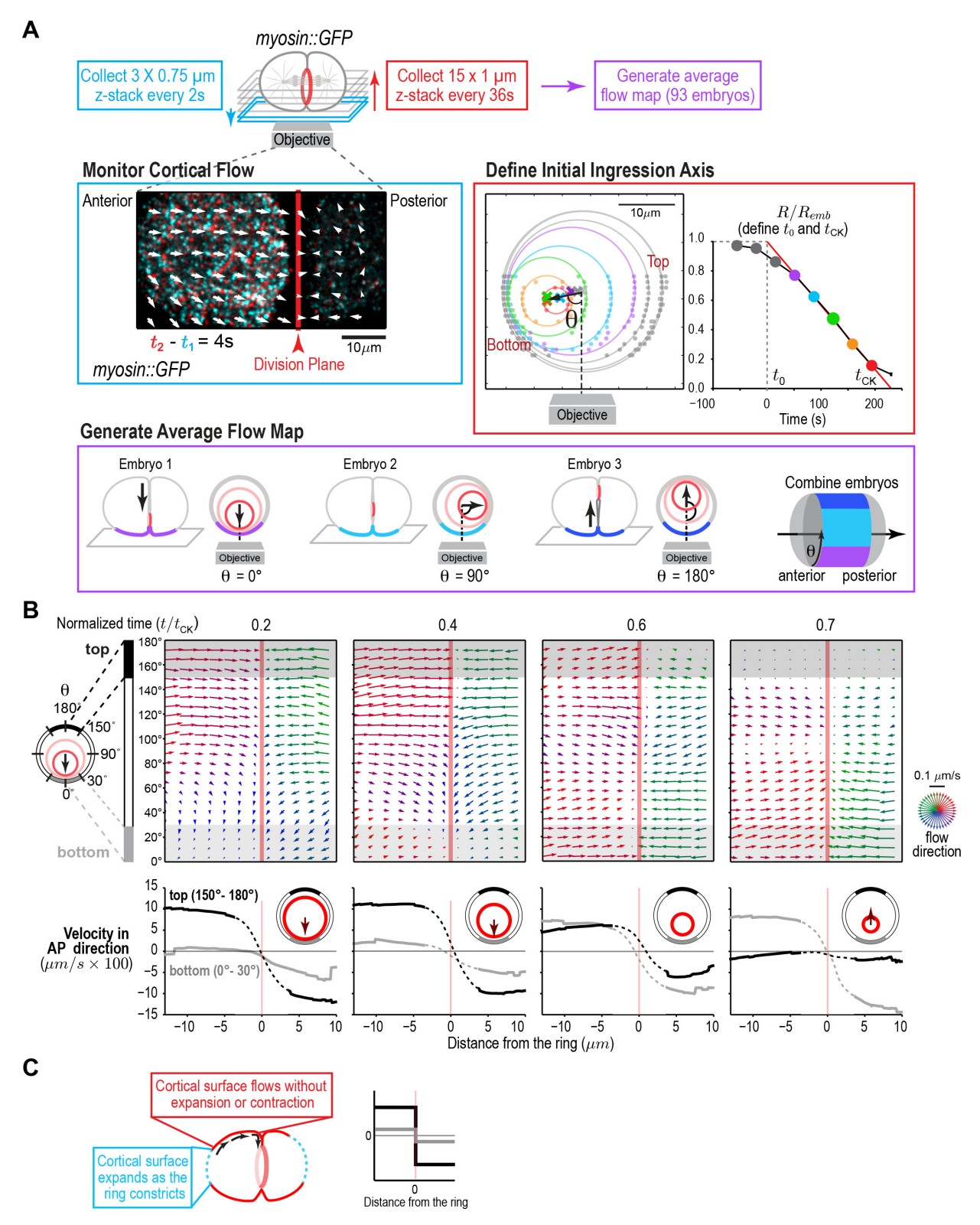

**Figure 1.** A quantitative map of cortical surface dynamics during the first cytokinesis in the *C. elegans* embryo reveals that the cortical surface at the cell poles expands as the ring constricts. (**A**) (*top*) Schematic of the experimental procedure. (*middle, left*) Superposition of images of the cortex acquired 4 s apart. Arrows indicate cortical flow (magnified 2.5X). (*middle, right*) The initial ingression axis, $t_0$, and $t_{CK}$ were defined as shown for a representative embryo. The angle θ specifies the position of the imaged cortex relative to the initial furrow ingression axis. Image and quantification are
*Figure 1 continued on next page*

*Figure 1 continued*

representative of the 93 imaged embryos. (*bottom*) Angular position was used to combine data from 93 embryos to generate an average flow map. (**B**) (*top*) Average flow at the indicated timepoints. Arrows show direction and magnitude of the displacement in 1 s (magnified 20X). (*middle*) Graphs are average velocity in the A-P direction versus position along the A-P axis for the cortex on the top (*black*) and bottom (*grey*) of the embryo (*shaded in flow maps*). Surface movement changes direction across the division plane, the apparent velocity gradient close to the division plane is a projection artifact due to the fact that the cortical surface turns inwards as it approaches the furrow from either side (dotted regions on velocity curves). (**C**) Schematics show the predicted cortical velocity profile along the AP axis if surface is gained at the poles; velocity would be constant in magnitude within the flow map region with opposite directions on the two sides of the ring, as is experimentally observed.

DOI: https://doi.org/10.7554/eLife.36073.002

The following figure supplements are available for figure 1:

**Figure supplement 1.** Actin and myosin move together with the cortical surface during cytokinesis.

DOI: https://doi.org/10.7554/eLife.36073.003

**Figure supplement 2.** An automated method for monitoring contractile ring closure.

DOI: https://doi.org/10.7554/eLife.36073.004

**Figure supplement 3.** Different profiles of cortical surface velocity along the A-P axis are predicted for different spatial patterns of surface expansion.

DOI: https://doi.org/10.7554/eLife.36073.005

constriction rate from the control rate of 0.22 ± 0.05 µm/s to ~0.25 µm/s over the 13 s interval (see Materials and methods for details). In contrast, the measured constriction rate was not increased after cutting (0.18 ± 0.03 µm/s; *Figure 2B,C*), indicating that cortical tension does not impose significant resistance to ring pulling. Cuts made perpendicular to the ring also had no effect on the constriction rate (0.19 ± 0.03 µm/s data not shown). Consistent with the results of the laser cutting experiments, inhibiting the Arp2/3 complex by depleting its ARX-2 subunit, which is expected to reduce effective cortical viscosity and thus cortical tension (*Chaudhuri et al., 2007*; *Davies et al., 2014*; *Tseng and Wirtz, 2004*), also did not alter the constriction rate (*Figure 2—figure supplement 1A*).

Putting the results of our flow map analysis together with our laser cutting and Arp2/3 inhibition experiments, we conclude that the cortical surface at the cell poles expands in response to tension generated by the constricting ring without providing significant resistance that would affect the rate of ring closure. In contrast, the cortex in the region between the ring and the poles flows toward the ring without expansion or compression. The differential response of the polar cortex to ring-generated tension is consistent with the idea of polar relaxation hypothesized in early conceptual models of cytokinesis (*Greenspan, 1978*; *Swann and Mitchison, 1958*; *Taber, 1995*; *White and Borisy, 1983*; *Wolpert, 1960*; *Zinemanas and Nir, 1987, 1988*), and suggests that the polar cortex has unique mechanical properties compared to the intervening cortex that does not expand (see Discussion). The fact that cortical tension does not limit the rate of ring constriction suggests that the constriction rate is instead limited by ring-internal friction. We conclude that the viscosity of the polar cortex is negligible compared to the viscosity internal to the ring; thus, ring myosin generated force primarily counters ring internal friction to drive ring constriction (*Figure 2—figure supplement 1B*). Ring constriction, in turn, affects cortical tension and drives expansion of the polar cortex.

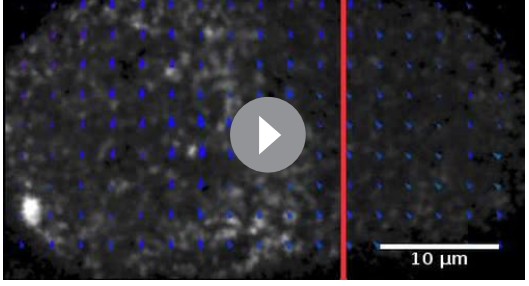

**Video 1.** Cortical flow imaged in a control embryo expressing myosin::GFP. Playback is 6x realtime. The video is constructed from maximum intensity projection of 3 × 0.75 µm plane z-stacks acquired at 2 s intervals. The red line marks the position of the division plane. The arrows represent the surface movement between consecutive frames at the base of the arrow. The length of the arrow is five times the magnitude of movement. The direction is also color coded according to the color wheel as shown in *Figure 1B*.

DOI: https://doi.org/10.7554/eLife.36073.006

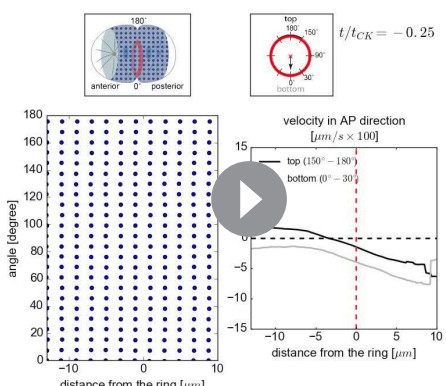

**Video 2.** Average cortical flow map calculated from time lapse imaging of the cell surface in 93 control embryos expressing myosin::GFP. (*top, left*) Schematic illustrates location of the cylindrical surface covered by the map. (*top, right*) Dynamic schematic illustrates ring size and position for each value of t/t$_{CK}$. (*bottom, left*) The movement of each blue dot corresponds to surface movement at its location. The y-axis is the angular position relative to the initial ingression axis. The x-axis is the distance from the division plane along the anterior-posterior axis. (*bottom, right*) Dynamic graph plots the magnitude of the component of surface velocity aligned along the anterior-posterior axis for the top (150–180°; black) and bottom (0–30°; grey) regions of the cortex.

DOI: https://doi.org/10.7554/eLife.36073.007

# Ring myosin compresses cortical surface along the axis perpendicular to the ring, pulling new cortical surface into the ring at a rate proportional to the amount of ring myosin

In the *C. elegans* embryo, as in other systems, spindle-based signaling following anaphase onset activates RhoA on the cortex in an equatorial zone (the Rho zone), leading to the recruitment of contractile ring proteins including myosin II, the septins, and anillin (*Jenkins et al., 2006*; *Maddox et al., 2005, 2007*; *Mangal et al., 2018*; *Motegi and Sugimoto, 2006*; *Schonegg et al., 2007*; *Tse et al., 2012*; *Werner et al., 2007*). An astral microtubule-based mechanism that clears contractile ring proteins from the polar cortex also limits contractile ring protein recruitment to a defined equatorial zone (*Mangal et al., 2018*; *Werner et al., 2007*). Prior work in the *C. elegans* embryo has suggested that the cortical surface within the Rho zone is compressed along the axis perpendicular to the ring as the contractile ring forms following anaphase onset, coincident with the alignment of actin filament bundles (*Reymann et al., 2016*). Compression, which is the reduction of cortical surface area, is detected as a gradient in the velocity of cortical surface flow. Consistent with the idea that cortical surface is compressed during contractile ring assembly, we observed a linear gradient in the velocity of cortical flow that spanned the cell equator in our flow map at early time points prior to furrow ingression (*Figure 3A*). The linear gradient indicated that, during contractile ring assembly, when the ring is on the embryo surface, cortical surface is uniformly compressed across a 10 µm wide region along the perpendicular-to-the-ring axis between the two relaxing poles.

After its assembly, the ring begins to shorten in the around-the-ring direction. Ring constriction has been proposed to be coupled to the progressive disassembly of the ring (constriction-coupled disassembly); in other words, ring components are lost in proportion to reduction in perimeter (*Carvalho et al., 2009*; *Murrell et al., 2015*; *Schroeder, 1990*). As the ring constricts, it pulls the cortex behind it, which leads to a flow of cortex into the division plane. We were interested in whether the compression of cortical surface area within the Rho zone/contractile ring along the perpendicular-to-the-ring axis is limited to contractile ring assembly, or whether it also continues during ring constriction. If compression stops, the constricting ring would generate the division plane by pulling the cortex behind it, and the cortical surface area entering the division plane would equal the area of the division plane. In contrast, if compression continues during constriction, the cortical surface area entering the division plane would be larger than the area of the division plane.

To distinguish between these possibilities, we used the 4D cortical flow map to measure the cortical surface area entering the division plane and compare it to the area of the division plane (accounting for the fact that two surfaces are generated-*red outline in Figure 3B*). This analysis revealed that the area of the cortical surface that entered the division plane during ring constriction was significantly greater than the area of the division plane (*Figure 3B*, *middle panel*). The flux of cortical area into the division plane was 1.5- to 2-fold higher than the rate of change in the area of the division plane throughout cytokinesis, suggesting that cortical surface area is continuously compressed within the Rho zone/contractile ring throughout constriction (*Figure 3B*, *right panel*). In control embryos,

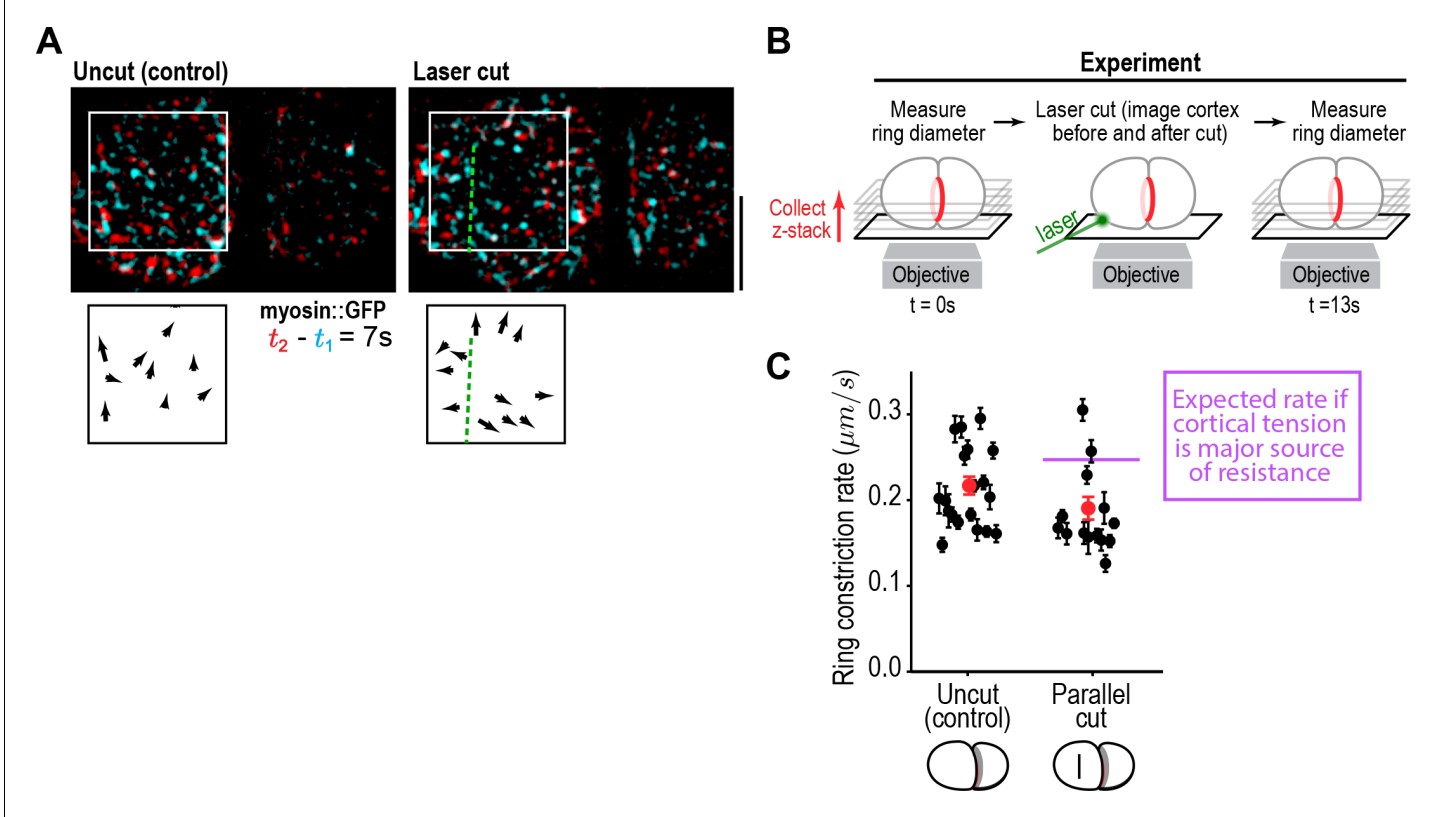

**Figure 2.** Cortical tension does not limit the rate of ring closure. (A) The success of cortical cuts was assessed by comparing surface images of cortical myosin before (*cyan*) and after (*red*) the cut to monitor the movement of myosin foci away from the cut site. Representative images are shown. Scale bar is 10 μm. (B) Schematic of laser ablation experiment to determine if cortical resistance limits the rate of contractile ring closure. Contractile ring sizes were measured from z-stacks acquired before and 13 s after a cut was made across the cortex with a laser. (C) Graph plots the rates of ring closure derived from before and after ring size measurements for uncut controls (n = 19 embryos) and embryos with cuts parallel to the division plane (n = 14 embryos). Black symbols are single embryo measurements with measurement errors. Red symbols are the means; error bars are the SEM. The purple line marks expected closure rate if cortical tension is a major source of resistance.

DOI: https://doi.org/10.7554/eLife.36073.008

The following figure supplement is available for figure 2:

**Figure supplement 1.** Arp2/3 depletion does not alter ring constriction kinetics.

DOI: https://doi.org/10.7554/eLife.36073.009

more cortex flowed in from the posterior side than from the anterior side, likely due to distinct mechanical properties downstream of the polarity machinery. Prior work showed that Arp2/3 inhibition impairs the recruitment of PAR-2 to the posterior cortex and makes myosin and actin dynamics on the posterior cortex more similar to those in embryo anterior (*Xiong et al., 2011*). Inhibiting the Arp2/3 complex by depleting ARX-2 abolished the difference between the two sides but did not change the difference between the total amount of cortex entering the division plane and the area of the plane (*Figure 3—figure supplement 1*; *Video 3*). This result suggests that cortical surface area compression along the axis perpendicular to the ring persists throughout constriction.

Next, we probed the relationship between the rate of cortical surface area compression along the perpendicular-to-the-ring axis and the levels of two contractile ring components, myosin, which is required for ring constriction and cortical surface compression (*Reymann et al., 2016*; *Shelton et al., 1999*), and anillin, a filament cross-linker that localizes to the ring but is not essential for constriction or compression (*Maddox et al., 2005*; *Maddox et al., 2007*; *Reymann et al., 2016*). To do this, we monitored *in situ*-tagged myosin::GFP (*Dickinson et al., 2013*) (*Figure 3C*) and GFP:: anillin (*Figure 3—figure supplement 2*) in end-on reconstructions of the division plane. Both ring components exhibited similar behavior. Because overall measurements of ring component levels and constriction/compression rates scale with ring perimeter, all of our analyses consider measurements

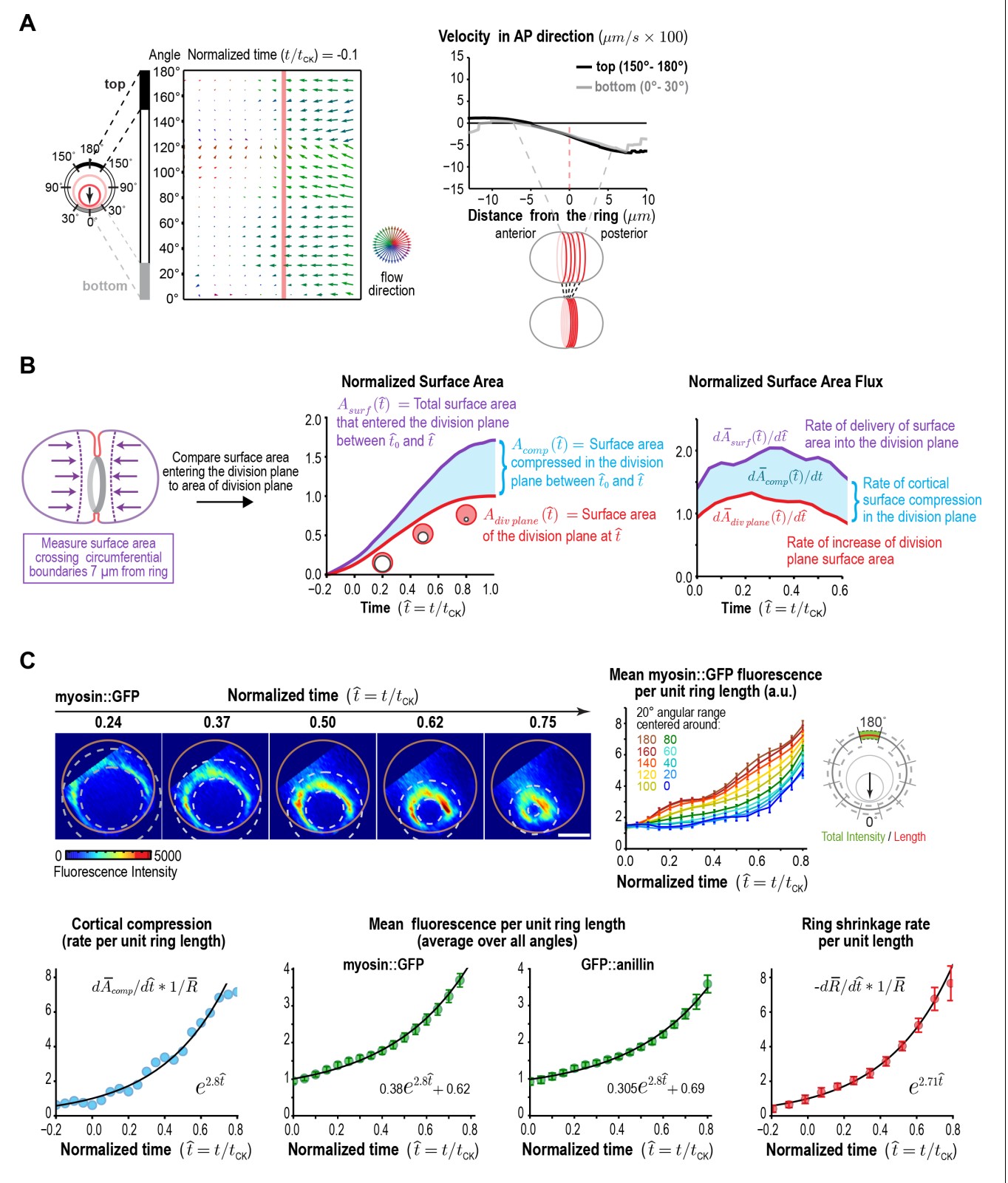

**Figure 3.** Ring myosin compresses cortical surface along the axis perpendicular to the ring, pulling in new cortical surface at a rate proportional to the amount of ring myosin. (**A**) The equatorial cortex is compressed during contractile ring assembly. Following the onset of spindle-based RhoA signaling, the initial recruitment of contractile ring proteins leads to uniform compression of cortical surface along the axis perpendicular to the forming ring across a 10 μm wide region spanning the cell equator. (*left*) Average flow map at (t/t_CK = −0.1) immediately after the onset of spindle-based signaling

*Figure 3 continued on next page*

*Figure 3 continued*

(n = 93 embryos). (*middle*) The surface velocity profile reveals a linear velocity gradient that spans the cell equator (−5 to +5 μm), indicating a uniform zone of cortical compression. (B) Cortical compression within the ring continues during constriction. (*left graph*) Plot comparing the area of the forming division plane (*red*) with the total cortical surface area that entered the division plane from the start of cytokinesis (*purple*; calculated as indicated in the schematic). (*right graph*) Plot comparing the rate of delivery of cortical surface into the division plane (*purple*) with the rate of growth of the division plane (*red*). The difference between the two is the rate of cortical surface compression (rate of reduction of cortical surface area; *cyan*). (C) The per-unit-length amount of ring myosin and the rate of cortical compression increase with the same exponential kinetics, suggesting that the rate of cortical compression may be controlled by the amount of myosin in the contractile ring/Rho zone. (*top left*) Representative images of the division plane in embryos expressing myosin::GFP reconstructed from 40-plane z-stacks. Gold circles mark the embryo boundary and dashed circles mark the boundaries used for ring intensity measurements. Scale bar is 10 μm. (*top right*) Graph plots per-unit-length myosin::GFP fluorescence for the indicated angular ranges (n = 36 embryos). (*bottom left*) Graph plots the rate of cortical surface compression per unit ring length (n = 93 embryos). (*bottom middle*) Graphs plot mean per-unit-length myosin::GFP (n = 36 embryos) and GFP::anillin (n = 26 embryos) fluorescence (n = 36 embryos) in the ring. (*bottom right*) Graph plots the per-unit-length rate of ring closure. Black lines are fitted single exponentials. Error bars are the SEM.

DOI: https://doi.org/10.7554/eLife.36073.010

The following figure supplements are available for figure 3:

**Figure supplement 1.** Arp2/3 inhibition abolishes the asymmetry in the amount of cortex entering the division plane from the anterior and posterior sides.

DOI: https://doi.org/10.7554/eLife.36073.011

**Figure supplement 2.** GFP::anillin fluorescence in the ring increases exponentially during constriction.

DOI: https://doi.org/10.7554/eLife.36073.012

**Figure supplement 3.** Correcting for signal attenuation with sample depth.

DOI: https://doi.org/10.7554/eLife.36073.013

**Figure supplement 4.** Ring component dynamics at the four-cell stage are consistent with exponential accumulation.

DOI: https://doi.org/10.7554/eLife.36073.014

per unit of ring length, which capture the evolution of the material properties of the ring independent of its perimeter. Quantification of mean per-unit-length fluorescence around the ring (after attenuation correction; *Figure 3—figure supplement 3*) revealed a steady increase for both markers as constriction proceeded. The increase in the per-unit-length amounts of myosin and anillin began on the top of the ring, which ingresses first, and initiated later on the bottom, which ingresses after the constriction midpoint (*Figure 3C*, *Figure 3—figure supplement 2*). Comparing the per-unit-length rate of cortical compression along the perpendicular-to-the-ring axis to the per-unit-length amounts of myosin and anillin revealed that both increased with the same exponential kinetics during constriction (*Figure 3C*). Thus, new cortical surface is pulled into the ring due to cortical compression at a rate proportional to the amount of ring myosin. Like the rate of cortical compression along the perpendicular-to-the-ring axis, the per-unit-length constriction rate also increased in proportion to the per-unit-length amount of myosin (*Figure 3D*). The exponential increase in the per-unit-length constriction rate explains the observed ability of the contractile ring to close at a relatively constant rate despite its progressively decreasing perimeter (*Bourdages et al., 2014*; *Carvalho et al., 2009*; *Zumdieck et al., 2007*). A relatively constant overall rate of ring

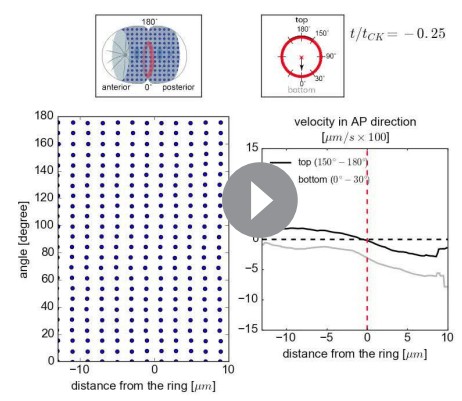

**Video 3.** Average cortical flow map calculated from time lapse imaging of the cell surface in 68 *arx-2(RNAi)* embryos expressing Myosin::GFP. (*top, left*) Schematic illustrates the location of the cylindrical surface covered by the map. (*top, right*) Dynamic schematic illustrates ring size and position for each value of t/t_ck. (*bottom, left*) The movement of each blue dot corresponds to surface movement at its location. The y-axis is the angular position relative to the initial ingression axis. The x-axis is the distance from the division plane along the anterior-posterior axis. (*bottom, right*) Dynamic graph plots the magnitude of the component of surface velocity aligned along the anterior-posterior axis for the top (150–180°; black) and bottom (0–30°; grey) regions of the cortex.

DOI: https://doi.org/10.7554/eLife.36073.015

closure is observed over a significant portion of constriction (*Figure 1A*; $t = 50\text{-}200s$) because the exponential increase in the per-unit-length constriction rate compensates for the decrease in ring perimeter.

In our prior work in four-cell stage *C. elegans* embryos (*Carvalho et al., 2009*), we showed that the per-unit-length levels of myosin, anillin and septins in the ring increase ~1.3 fold as ring perimeter decreases 2-fold (from 50 to 25 µm; see *Figure 3—figure supplement 4C*). At the time, we interpreted this data as suggesting that the per-unit-length amounts of ring components remain constant during ring closure. However, distinguishing between constant and exponential dynamics at the four-cell stage is challenging because the range of ring perimeters between furrow formation (perimeter ~50 µm) and contact with the midzone, which alters ring properties (perimeter ~23 µm; *Carvalho et al., 2009*), is too limited (*Figure 3—figure supplement 4A,B*). To determine if the data in four-cell stage embryos is consistent with an exponential increase in component levels, we analyzed per-unit-length myosin levels in the same strain expressing *in situ*-tagged myosin that we used for the one-cell experiment (right graph in *Figure 3—figure supplement 4C*), which yielded results very similar to our prior work employing a strain expressing myosin::GFP from an integrated transgene (left graph in *Figure 3—figure supplement 4C*; *Carvalho et al., 2009*). Although not sufficient to demonstrate exponential accumulation on their own, the four-cell data are well fit by the same exponential equation that describes myosin and anillin accumulation at the one-cell stage (*Figure 3—figure supplement 4C*, *Figure 3C*), consistent with the idea that contractile ring components accumulate via the same mechanism during both divisions.

## An analytical mathematical model for the positive feedback-mediated evolution of the contractile ring

Based on the above analysis, we conclude that, per-unit-length of the ring, the amount of ring myosin and anillin, the rate of cortical compression, and the rate of ring constriction all increase with near-identical exponential kinetics as the ring closes. The fact that ring components accumulate with exponential kinetics suggests control by positive feedback. Our results further suggest that the relevant feedback driving the accumulation of ring components could be between the amount of ring myosin and the rate of cortical surface compression within the ring, since: (1) the ring compresses cortical surface along the perpendicular-to-the-ring axis at a per-unit-length rate proportional to the amount of ring myosin and, (2) the per-unit-length amount of ring myosin increases at a rate proportional to the rate at which cortical surface is compressed within the ring. To explore this idea, we developed an analytical mathematical formulation that builds on our prior work at the four-cell stage. The natural coordinate system for contractile ring dynamics has two axes, an axis parallel to ring constriction (*Figure 4A*, *around-the-ring axis*) and an axis perpendicular to the ring between the relaxing poles (*Figure 4A*, *perpendicular-to-the-ring axis*). Our prior work indicated that shortening of the ring in the around-the-ring direction is coupled to the disassembly of ring components and does not change their per-unit-length levels (Constriction-Coupled Disassembly, *Figure 4A*; *Carvalho et al., 2009*). Our model, which we call the Constriction-Coupled Disassembly with Compression Feedback model, adds compression feedback along the axis perpendicular to the ring into this framework (*Figure 4A,B*). Our model consists of three equations with three model parameters and can recapitulate the experimentally observed dynamics of ring components, cortical flow into the division plane, and ring constriction.

After anaphase onset, spindle-based signaling activates RhoA on the cortex in an equatorial region termed the Rho zone (*Bement et al., 2006*; *Green et al., 2012*; *Jordan and Canman, 2012*; *Piekny et al., 2005*). In our model, the Rho zone and contractile ring are interchangeable. The Rho zone occupies the central region of the perpendicular-to-the-ring axis between the poles. Within this zone, RhoA-based signaling leads to the recruitment of an ensemble of interacting contractile ring proteins, including formin-nucleated actin, myosin, anillin and the septins, to the cortex after anaphase onset (*Jenkins et al., 2006*; *Maddox et al., 2005*, *2007*; *Mangal et al., 2018*; *Motegi and Sugimoto, 2006*; *Schonegg et al., 2007*; *Tse et al., 2012*; *Werner et al., 2007*). As our data indicate (*Figure 3A*), this leads to uniform compression of the cortex across this central 10 µm wide zone. We propose that, due to polar relaxation, compression of cortex in the Rho zone pulls naive cortex, defined as cortex outside the Rho zone (naive to RhoA-based signaling) into the Rho zone (*Figure 4A*). The new cortex that flows into the Rho zone as a result of compression is also loaded with contractile ring components and initiates compression. Thus, *along the perpendicular-to-the-*

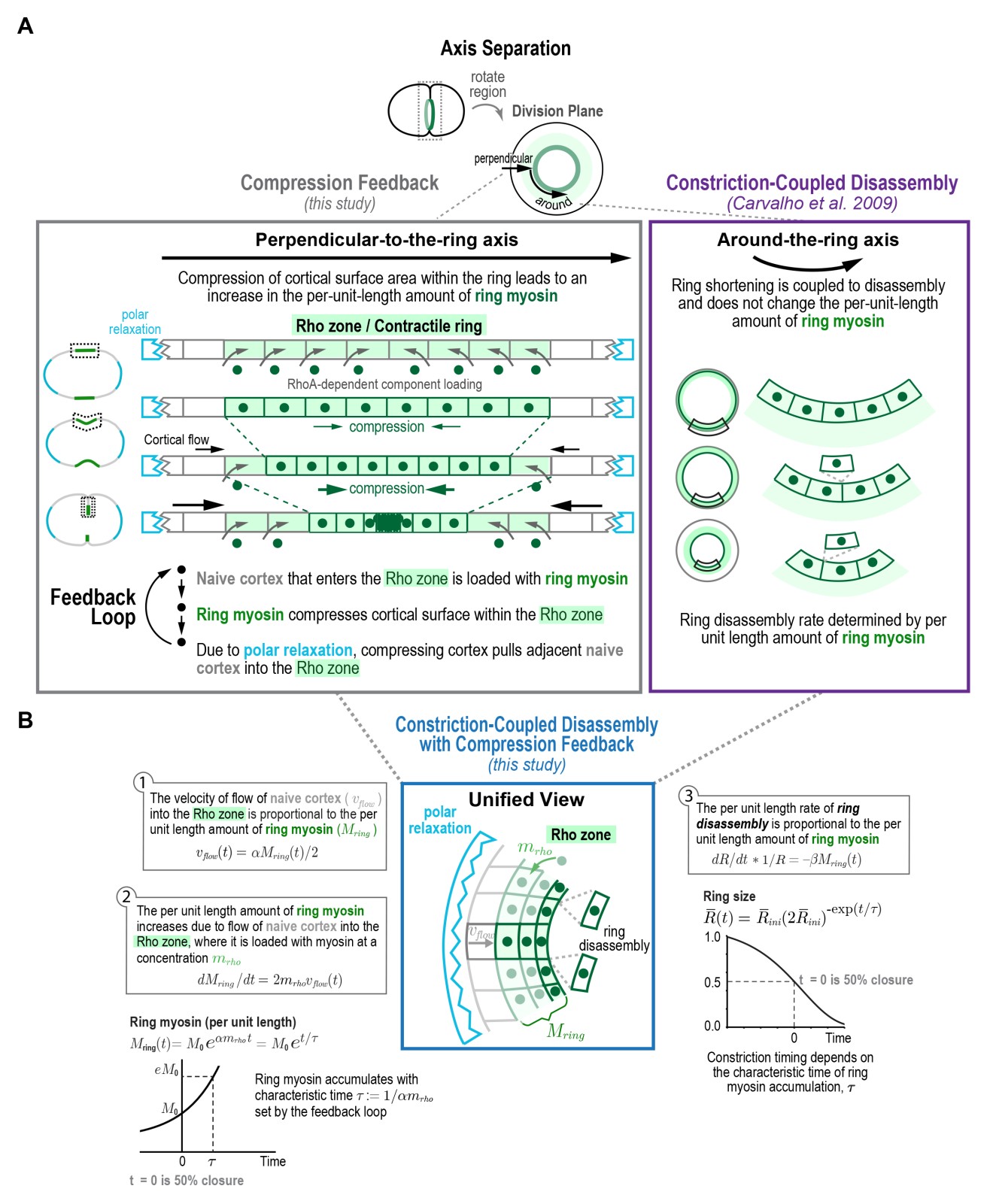

**Figure 4.** Constriction-coupled disassembly with compression feedback model. (**A**) The natural coordinate system for contractile ring dynamics has two axes, an axis parallel to ring constriction (*around-the-ring axis*) and an axis perpendicular to the ring (*perpendicular-to-the-ring axis*). Polar relaxation and filament alignment in the around-the-ring direction lead to anisotropy in behavior along the two axes, which are illustrated separately here. Our prior work analyzing component dynamics at the 4-cell stage in the *C. elegans* embryo has suggested that in the around-the-ring direction constriction

*Figure 4 continued*

is coupled to component disassembly (Constriction-Coupled Disassembly, *right*), so that ring constriction does not change the per-unit-length amount of ring components (*Carvalho et al., 2009*). Here, we propose that this is coupled to a feedback loop between ring myosin and compression-driven cortical flow (Compression Feedback, *left*) that operates along the axis perpendicular to the ring. Compression feedback would lead to an exponential increase in the per-unit-length amount of ring myosin and the per-unit-length constriction rate as the ring closes. (B) Constriction-Coupled Disassembly with Compression Feedback can be formulated as an analytical mathematical model consisting of three equations and three model parameters. (*left*) Equations (1) and (2) describe the feedback loop between the amount of ring myosin and the velocity of compression-driven flow of cortical surface into the ring. Solving these equations gives the expression for the per-unit-length amount of ring myosin, which accumulates exponentially as shown in the graph. (*right*) The feedback loop operating perpendicular to the ring controls the per-unit-length amount of ring myosin, which in turn controls the per-unit-length rate of ring constriction as described in Equation (3). Graph plots the equation for ring size resulting from solving the model equations in the time reference where $t = 0$ is the halfway point of ring closure.

DOI: https://doi.org/10.7554/eLife.36073.016

*ring axis* a feedback loop operates in which myosin in the Rho zone compresses cortical surface, which pulls naive surface into the Rho zone that is then loaded with myosin and other contractile ring components, leading to the exponential accumulation of ring components (*Figure 4A*, left panel). *In the around-the-ring direction*, ring shortening is coupled to the disassembly of ring components as we described previously (Constriction-Coupled Disassembly; *Carvalho et al., 2009*) and does not change per-unit-length component levels (*Figure 4A*, right panel).

In the mathematical formulation (*Figure 4B*), naive cortex flows into the Rho zone at a velocity ($v_{flow}(t)$) proportional to the per-unit-length amount of ring myosin ($M_{ring}(t)$; Equation (1) in *Figure 4B*), with $\alpha$ being the proportionality constant that relates the two. Ring myosin, in turn, increases at a rate proportional to this flow and the concentration of myosin that is loaded onto the cortex when it enters the rho zone ($m_{rho}$; Equation (2) in *Figure 4B*). As a result of the positive feedback between ring myosin and compression-driven flow, ring myosin increases exponentially with a characteristic time $\tau := 1/\alpha m_{rho}$ (time required for ring myosin to increase ~2.7 fold; *Figure 4B*, *left graph*). The per-unit-length rate of ring constriction ($\frac{dR}{dt} * \frac{1}{R}$) is proportional to the per-unit-length amount of ring myosin, related by the proportionality constant $\beta$ (Equation (3) in *Figure 4B*). To avoid the difficulty of accurately assigning the exact point when cytokinesis starts, we solved these equations in the time reference where $t = 0$ is the halfway point of ring closure ($\bar{R}(t = 0) = \frac{1}{2}$). In this time reference, the equation for ring size is:

$$\bar{R}(\bar{t}) = \bar{R}_{ini}(2\bar{R}_{ini})^{-\exp(\bar{t})}, \tag{4}$$

where $\bar{t} := t/\tau$ and $\bar{R}_{ini}$ is the dimensionless characteristic ring size (held fixed at a value of 1.1; see Materials and methods; *Figure 4B*, *right graph*). Other components, like anillin, that localize to the cell cortex will be delivered to the contractile ring via the same process as myosin, and would accumulate in a similar fashion, with

$$C_{ring}(\bar{t}) - C_{ring,base} = \frac{\alpha c_{rho}}{\beta}\ln(2\bar{R}_{ini})e^{\bar{t}}, \tag{5}$$

$$C_{ring,base} := C_{0,ring} - \ln(2\bar{R}_{ini})\frac{\alpha c_{rho}}{\beta}, \tag{6}$$

where $C_{0,ring}$ is the per-unit-length amount of the component at the half-way point of ring closure, $C_{ring,base}$ is the baseline amount of the ring component that does not increase exponentially, and $c_{rho}$ ($m_{rho}$ for myosin) is the concentration of the component loaded onto naive cortex when it enters the rho zone. The velocity of cortical flow and the constriction rate are

$$v_{flow}(\bar{t}) = \frac{\alpha}{\beta}\ln(2\bar{R}_{ini})\,e^{\bar{t}}, \tag{7}$$

$$-\frac{1}{\bar{R}}\frac{d\bar{R}}{d\bar{t}} = \ln(2\bar{R}_{ini})\,e^{\bar{t}}. \tag{8}$$

Thus, the per-unit-length constriction rate, velocity of cortical flow, and ring component amounts

would all increase exponentially, as we have observed experimentally (*Figure 3C*), with the characteristic time of ring myosin accumulation ($\tau = 1/\alpha m_{rho}$) set by the feedback loop between ring myosin and cortical flow. We conclude that an analytical mathematical formulation that combines Compression Feedback along the perpendicular-to-the-ring axis with Constriction-Coupled Disassembly along the around-the-ring axis can recapitulate the experimentally observed dynamics of ring components, cortical flow into the division plane, and ring constriction.

## Fluorescence recovery after photobleaching of the division plane is consistent with constriction-coupled disassembly with compression feedback

The Constriction-Coupled Disassembly with Compression Feedback model is characterized by anisotropy in the behavior of the cortex in the perpendicular-to-the-ring and around-the-ring directions that results from polar relaxation (*Figure 5A*, *left*). In the perpendicular-to-the-ring direction, cortical compression within the ring pulls in cortical surface, which is loaded with ring components and increases their per unit length amount. In contrast, in the around-the-ring direction, constriction is coupled to disassembly and does not affect the per-unit-length amount of ring components. An alternative model that could also explain the increase in the per-unit-length amount of ring components, which we will refer to as the 'Retention' model, is that the per-unit-length constriction rate accelerates due to retention of myosin and/or other ring components during ring shortening (*Figure 5A*, *right*). In the Retention model, ring components are not lost due to disassembly as ring perimeter decreases. Instead, their total amounts in the ring remain constant during constriction, resulting in an increase in per-unit-length amounts in inverse proportion to the reduction in ring perimeter (levels would increase as $\frac{1}{R}$). In the perpendicular-to-the-ring direction, compression would still pull cortical surface into the ring, as we have shown occurs experimentally, but the Retention model assumes that this flow would not deliver myosin into the ring, either because levels of myosin on the delivered cortex are insignificant relative to the amount of myosin in the ring or because the delivered myosin is lost due to disassembly. Comparison of the fits to the data for the per-unit-length amounts of ring myosin and anillin and the rates of ring shrinkage and cortical compression for the two models (*Figure 5B*, *Figure 5—figure supplement 1*) suggested that, whereas the Retention model could approximate the data, the Constriction-Coupled Disassembly with Compression Feedback model fit the data significantly better.

Since both models could approximate the data, we designed a photobleaching experiment to definitively distinguish between them (*Figure 6A*). We photobleached myosin::GFP in the division plane at ~30% closure, and then monitored the per-unit-length amount of fluorescent myosin in the bleached rings compared to control unbleached rings during constriction. The difference between the two curves is the per-unit-length amount of bleached myosin in the ring (*Figure 6C*). This experiment can be conceptually thought of as a 'pulse-chase' experiment in which we generate a population of bleached subunits in the ring and then follow their fate as the ring constricts. In the Retention model, all ring components are recruited as the ring assembles; as the ring constricts, these components are not lost, but instead become progressively more concentrated. In our experiment, we bleached all the myosin in the ring at the point when the ring was 30% closed. If we successfully bleached all the fluorescent myosin in the ring, the Retention model would predict that there would be no recruitment of additional components and therefore no new fluorescent myosin would appear as the ring decreased in size. However, we also need to consider the likely possibility that our bleach might not be perfect, and some fluorescent subunits, initially below the level of detection, remain in the ring. In this case, the per unit length amount of both the bleached and the unbleached subunits would be expected to increase in inverse proportion to the reduction in ring perimeter (as $\frac{1}{R}$, *Figure 6B*, *top panels*). This is not what we observed. Instead, the per-unit-length amount of bleached subunits (subunits present in the ring when it was 30% closed) did not increase as $\frac{1}{R}$, but instead remained constant (*Figure 6C*, *black curves*). From this, we conclude that the bleached subunits are not retained but are lost in proportion to the reduction in ring perimeter as the ring constricts as expected for the Constriction-Coupled Disassembly with Compression Feedback model (*Figure 6B*, *lower panels*).

Our 'pulse chase' experiment indicated that the myosin in the ring when it was 30% closed is subsequently lost in proportion to the reduction in ring perimeter as the ring constricts. In other words,

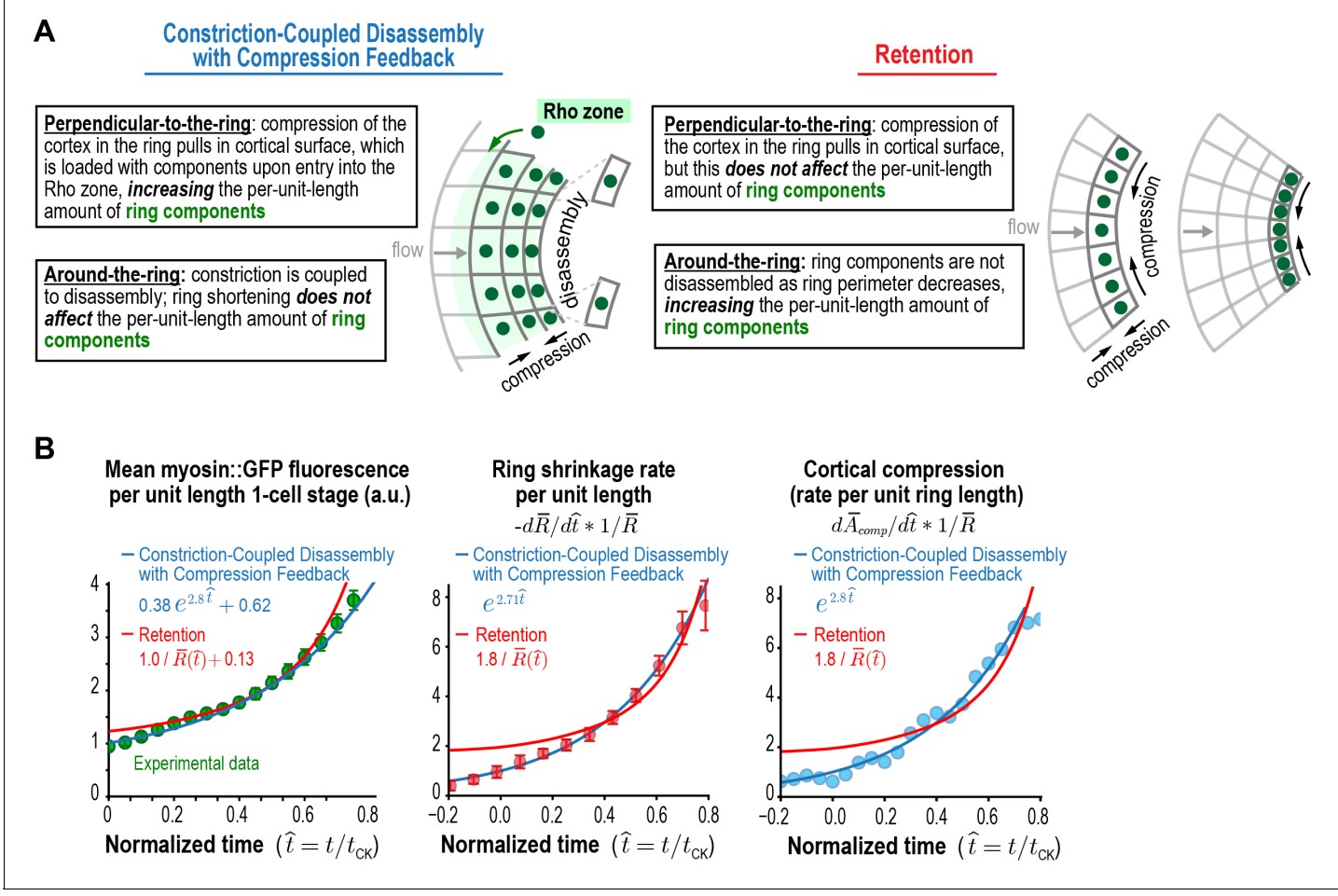

**Figure 5.** Myosin accumulation and the rates of ring constriction and cortical compression can be approximated with a retention model but are fit better by the Constriction-Coupled Disassembly with Compression Feedback model. (**A**) Two models could explain the acceleration in the per-unit-length constriction rate during constriction. In the Constriction-Coupled Disassembly with Compression Feedback model, the increase in per-unit-length component amounts arises from a feedback loop between ring myosin and compression-driven cortical flow along the direction perpendicular to the ring. In the Retention model, the increase in per-unit-length component amounts arises because components are retained rather than lost due to disassembly as ring perimeter decreases. (**B**) Graphs show mean per-unit-length myosin::GFP fluorescence in the ring along with the per-unit-length constriction and cortical compression rates. Myosin fluorescence data is reproduced from *Figure 3C* to allow comparison of the best fits for the Constriction-Coupled Disassembly with Compression Feedback (*blue lines*) and Retention (*red lines*) models.

DOI: https://doi.org/10.7554/eLife.36073.017

The following figure supplement is available for figure 5:

**Figure supplement 1.** Total myosin::GFP and GFP::anillin in the ring.

DOI: https://doi.org/10.7554/eLife.36073.018

the contribution of the population of myosin that was in the ring at the 30% closed stage to the total per-unit-length amount of ring myosin neither increases nor decreases as the ring constricts further, but instead remains constant. Thus, since the per-unit-length amount of myosin in the ring increases exponentially as the ring closes, this exponential increase must be due to addition of new myosin to the ring. Our data also revealed that the per-unit-length amount of fluorescent myosin in the bleached rings was equivalent to the per-unit-length amount of myosin in control rings minus the per-unit-length amount of myosin in the ring at the time when it was bleached. The fact that fluorescent myosin is added to bleached rings at the same rate as it is added to control rings suggests that there is very little exchange of the bleached myosin subunits with fluorescent myosin from the cytoplasm. If there was significant exchange, we would expect the rate of increase of the per unit length amount of fluorescent myosin to be higher in the bleached rings than in the controls (equal to new

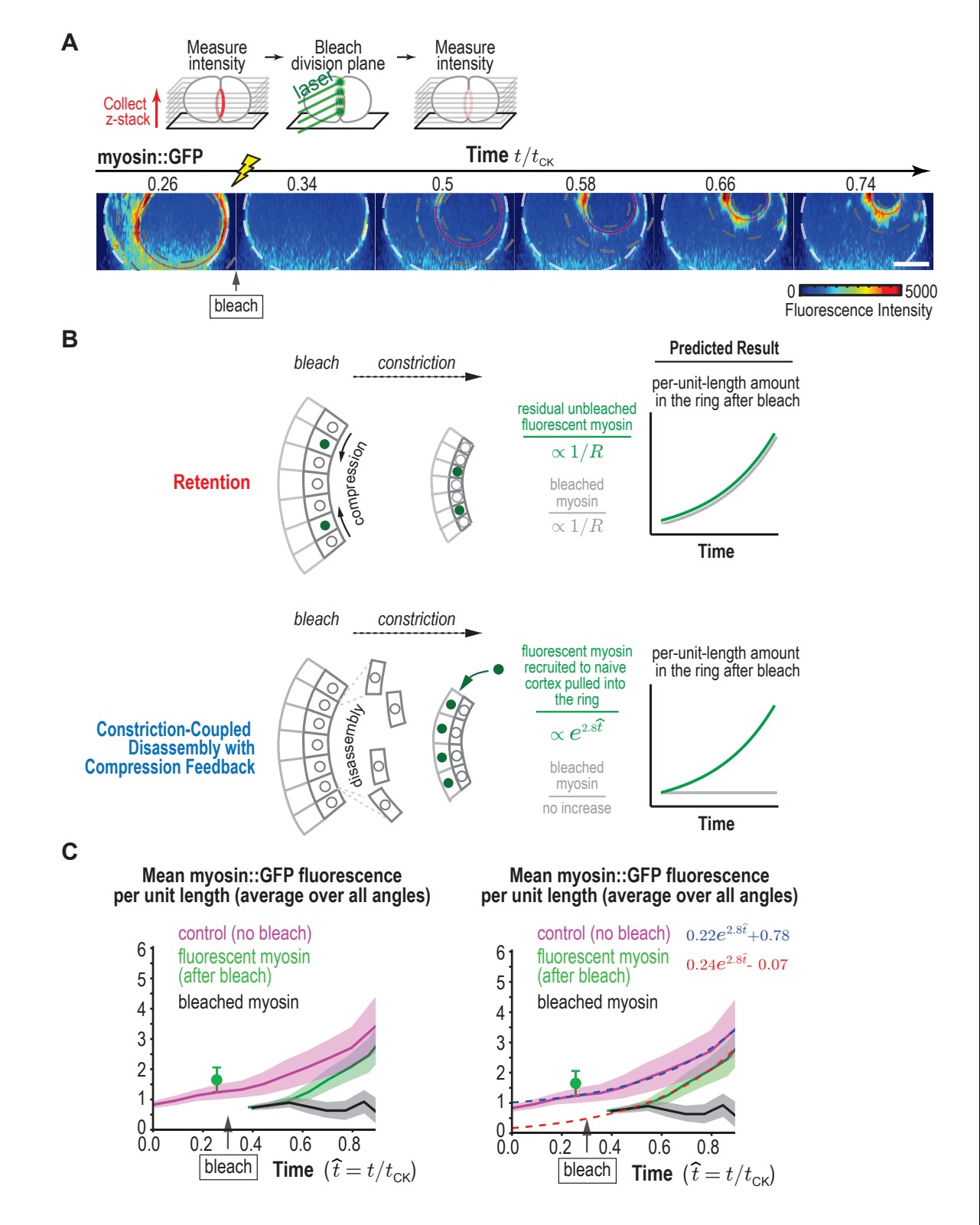

**Figure 6.** Fluorescence recovery after photobleaching of the division plane at the one-cell stage rules out Retention and is consistent with Constriction-Coupled Disassembly with Compression Feedback. (**A**) (*top*) Schematic of the photobleaching experiment. (*bottom*) Images of the division plane reconstructed from 30 × 1 µm z-stacks of an embryo expressing myosin::GFP whose division plane was bleached at $t/t_{CK}$ ~0.3. Red circle marks the contractile ring and dashed circles mark the boundaries used for ring intensity measurements. Image series is representative of eight imaged embryos.

*Figure 6 continued on next page*

*Figure 6 continued*

(B) Schematics illustrate the expected results predicted by the Retention and Constriction-Coupled Disassembly with Compression Feedback models. (C) (*left*) Graph plotting the mean per-unit-length amounts of fluorescent myosin::GFP in the ring for control embryos (*pink*, n = 24 embryos) and embryos in which the division plane was bleached at the indicated time (*green*, n = 8 embryos). The amount of bleached myosin::GFP in the ring (*black*) was calculated as the difference between the control and after bleach curves. Solid continuous lines are the average curves with errors shown as shaded regions. (*right*) Same as the graph on the left with dashed lines, representing exponential fits to the data, added. Errors for the control and after bleach data are SD and errors for the difference are SEM. Scale bar is 10 μm.

DOI: https://doi.org/10.7554/eLife.36073.019

addition plus exchange). In summary, our division plane photobleaching experiment at the one-cell stage is consistent with the Constriction-Coupled Disassembly with Compression Feedback model.

Finally, we wanted to assess whether the results of our division plane bleaching experiment at the one-cell stage are consistent with our prior work bleaching a spot on the contractile ring in embryos expressing fluorescent myosin at the four-cell stage. In this prior work, kymographs of a fixed length region along the constricting arc showed that the bleached spot narrowed as the ring constricted, yielding a tornado shape (*Figure 7A*; *Carvalho et al., 2009*). These experiments suggested that myosin in the ring disassembles in a constriction-coupled fashion, which is in agreement with our findings in the division plane bleach experiments at the one-cell stage (black curves in *Figure 6C*). In addition to Constriction-Coupled Disassembly, which does not alter the per-unit-length amount of ring myosin, the division plane bleaching experiments, along with our cortical flow analysis and modeling at the one-cell stage, suggested that new myosin is delivered into the ring by compression feedback along the axis perpendicular to the ring (illustrated in *Figure 7B*). We were interested in whether compression feedback could also drive component accumulation and constriction rate acceleration at the four-cell stage. We note that in our spot bleaching experiments, we had noticed that rather than closing ~50% over 95 s as predicted by Constriction-Coupled Disassembly alone, the tornados thinned more rapidly than expected (*Carvalho et al., 2009*), which is a prediction of the Constriction-Coupled Disassembly with Compression Feedback model (*Figure 7C*). As a more definitive test of whether myosin is delivered into the ring during constriction at the four-cell stage, like it is at the one-cell stage, we bleached the entire visible arc in cells at the four-cell stage and then monitored fluorescent myosin as they constricted. New myosin appeared in the bleached four-cell stage arcs in a fashion very similar to the bleached rings at the one-cell stage, suggesting that a similar mechanism delivers new myosin into the ring to drive the increase in the per-unit-length constriction rate at the one- and four-cell stages.

## Discussion

The near-constant overall rate of cytokinetic ring closure, despite the decreasing size of the ring, ensures that the partitioning of the contents of a mother cell occurs in a timely manner. Here, we provide evidence for a mechanism that explains this conserved property of the cytokinetic contractile ring. Our proposal arises from the near-identical exponential kinetics with which the amount of ring components, constriction rate, and rate of cortical compression (all measured per unit length of the ring) increase as the ring closes. This similarity led us to propose and quantitatively model positive feedback between ring myosin and compression-driven flow of cortex into the ring as the basis for the constant closure of the ring. Below we discuss this proposal in light of prior observations and models for cytokinetic ring closure.

### The constriction rate increase during ring closure is accompanied by an increase in the amount of myosin and other contractile ring components

In prior work (*Carvalho et al., 2009*), we found that once components are incorporated into the contractile ring, they did not exchange with subunits in the cytoplasm, but were instead lost via constriction-coupled disassembly. Our new experiments monitoring component levels at the one-cell stage and division plane photobleaching experiments at the one- and four-cell stages, support these conclusions. However, they also suggest the existence of a new feature that was not appreciated in the prior study due to the technical limitation that, at the four-cell stage, ring component levels could

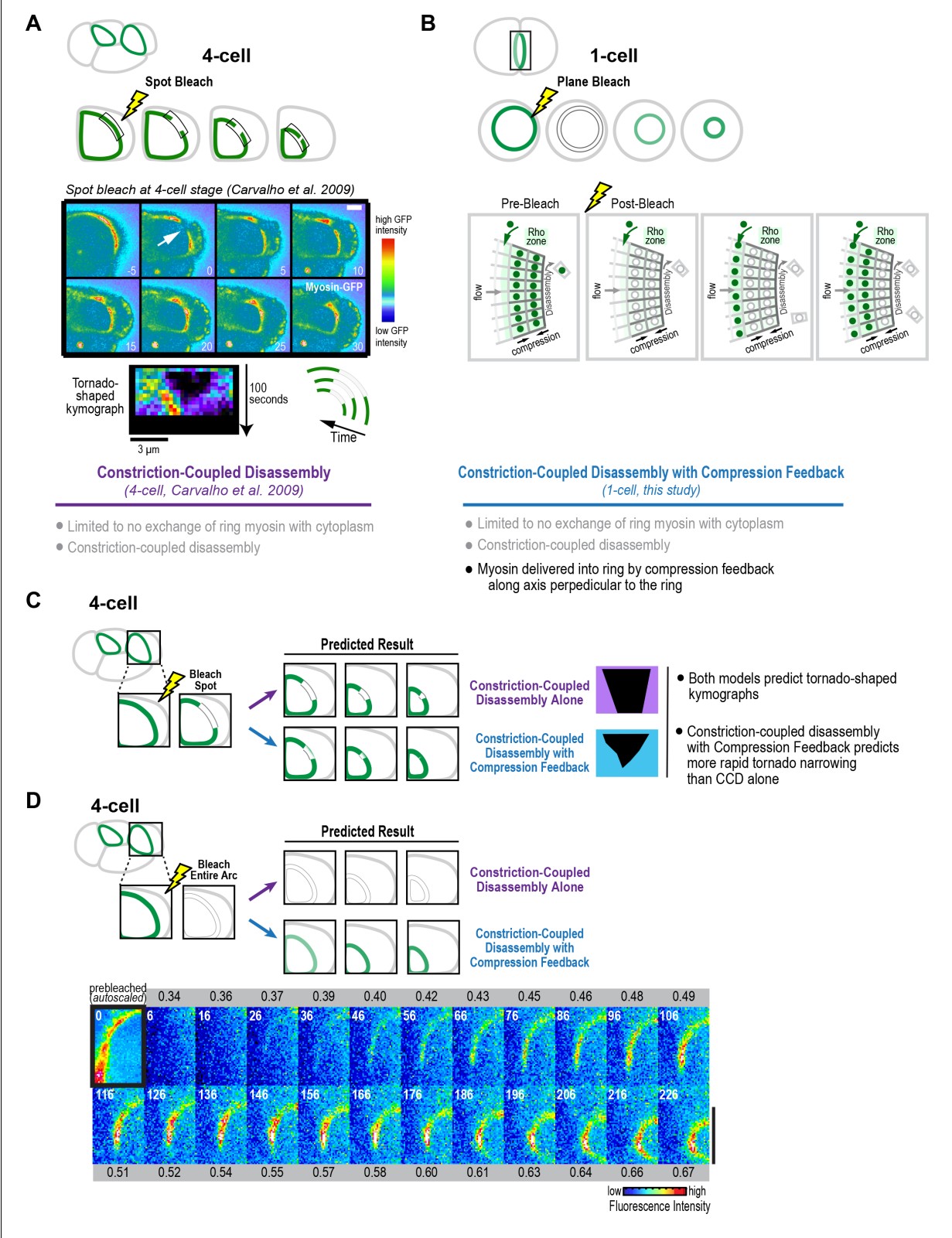

**Figure 7.** Fluorescence recovery after photobleaching of the division plane at the four-cell stage is consistent with Constriction-Coupled Disassembly with Compression Feedback. (**A**) (*top*) Schematics of an experiment that we performed previously in which a spot was bleached in contractile rings at the four-cell stage (***Carvalho et al., 2009***). In kymographs of a fixed length region of the arc, the bleached spot narrowed as the ring constricted, yielding a tornado shape. (*middle*) Image panels and kymograph reproduced from Figure 5 of ***Carvalho et al. (2009)***. The top set of panels show a

*Figure 7 continued on next page*

*Figure 7 continued*

four-cell stage spot bleaching experiment. The region indicated by the arrow was bleached. Time is in seconds after photobleaching. Stills are in pseudocolor with hot-to-cold colors representing high-to-low GFP intensity. Scale bar, 5 µm. Below the images is a representative tornado-shaped kymograph. The time interval between each row of pixels is 5 s. The top row corresponds to the time point before photobleaching. Pixel dimensions are 0.27 × 0.27 µm. These experiments suggested that after its incorporation into the ring, ring myosin does not exchange with myosin in the cytoplasm, and that the ring disassembles in constriction-coupled fashion. (B) In addition to Constriction-Coupled Disassembly, which does not alter the per-unit-length amount of ring myosin, the division plane bleaching experiments that we performed at the one-cell stage suggest that new myosin is delivered into the ring by compression feedback along the axis perpendicular to the ring. (C, D) Schematics in C and D show the predictions of Constriction-Coupled Disassembly Alone and Constriction-Coupled Disassembly with Compression Feedback for spot (C) and division plane (D) bleaching experiments at the four-cell stage. Rather than closing by 50% over 95 s as predicted by Constriction-Coupled Disassembly Alone, the tornados thinned more rapidly, consistent with the prediction of the Constriction-Coupled Disassembly with Compression Feedback model (*Carvalho et al., 2009*). (D) To test whether Compression Feedback delivers components to the ring at the four-cell stage as well as at the one-cell stage, we monitored recovery after photobleaching the entire contractile arc. Images show a representative bleached embryo (n = 10). The observed recovery pattern was similar to what we observed at the one-cell stage, consistent with the idea that compression feedback contributes to ring component accumulation at both the one- and four-cell stages. Scale bar is 10 µm.

DOI: https://doi.org/10.7554/eLife.36073.020

only be analyzed over a limited range of ring perimeters. This new feature is a feedback-based mechanism that operates along the axis perpendicular to the ring to drive the exponential accumulation of contractile ring proteins. We refer to the model that incorporates this feedback mechanism as *Constriction-Coupled Disassembly with Compression Feedback* (*Figure 8*).

The presence of compression feedback has important implications for understanding the mechanism that underlies the acceleration of the per-unit-length constriction rate during ring closure. In contrast to the prior view that the per-unit-length amount of ring components remains constant as ring perimeter decreases (*Carvalho et al., 2009*), our one-cell embryo data show that the per-unit-length amount of ring components increases exponentially in the same manner as the per-unit-length constriction rate. This finding suggests that the per-unit-length increase in the concentration of myosin and other contractile ring components underlies the acceleration of the constriction rate, and argues against the contractile unit model, which we had previously proposed to explain how the per-unit-length constriction rate might increase in the absence of a proportional increase in component levels (*Carvalho et al., 2009*).

## Constriction-Coupled Disassembly with Compression Feedback: a new explanation for acceleration of the per-unit-length constriction rate during cytokinesis

Our data suggest that the constriction rate acceleration is due to an increase in the per-unit-length amount of myosin and other contractile ring components. Our division plane photobleaching data rule out the idea that the increase in the per-unit-length amount of ring components is due to component retention and suggest instead that it is due to the action of a feedback loop that delivers myosin and other components into the ring. Our cortical flow analysis and modeling suggest that the feedback could be between ring myosin and compression-driven cortical flow in the perpendicular-to-the-ring direction (*Figure 8*). In our model, polar relaxation allows ring myosin to compress cortical surface along the pole-to-pole axis perpendicular to the ring, thereby pulling new cortex into the contractile ring/Rho zone that is loaded with myosin and other contractile ring components. An increase in the per-unit-length amount of ring myosin, in turn, would lead to increased cortical compression, resulting in a feedback loop that drives an exponential increase in the per-unit-length amount of ring components. In this model, the overall amounts of myosin, anillin (and presumably other components) in the ring would remain relatively constant as the ring constricts, as is experimentally observed (*Figure 5—figure supplement 1*), due to a balance between loss resulting from disassembly-coupled ring shortening and accumulation due to the feedback loop in the perpendicular-to-the-ring direction. Thus, the relatively constant overall levels of ring components would mask a dramatic restructuring of the ring that is occurring during closure.

We note that the Compression Feedback component of the model we propose here is reminiscent of early conceptual models of cytokinesis, which hypothesized that polar relaxation coupled to

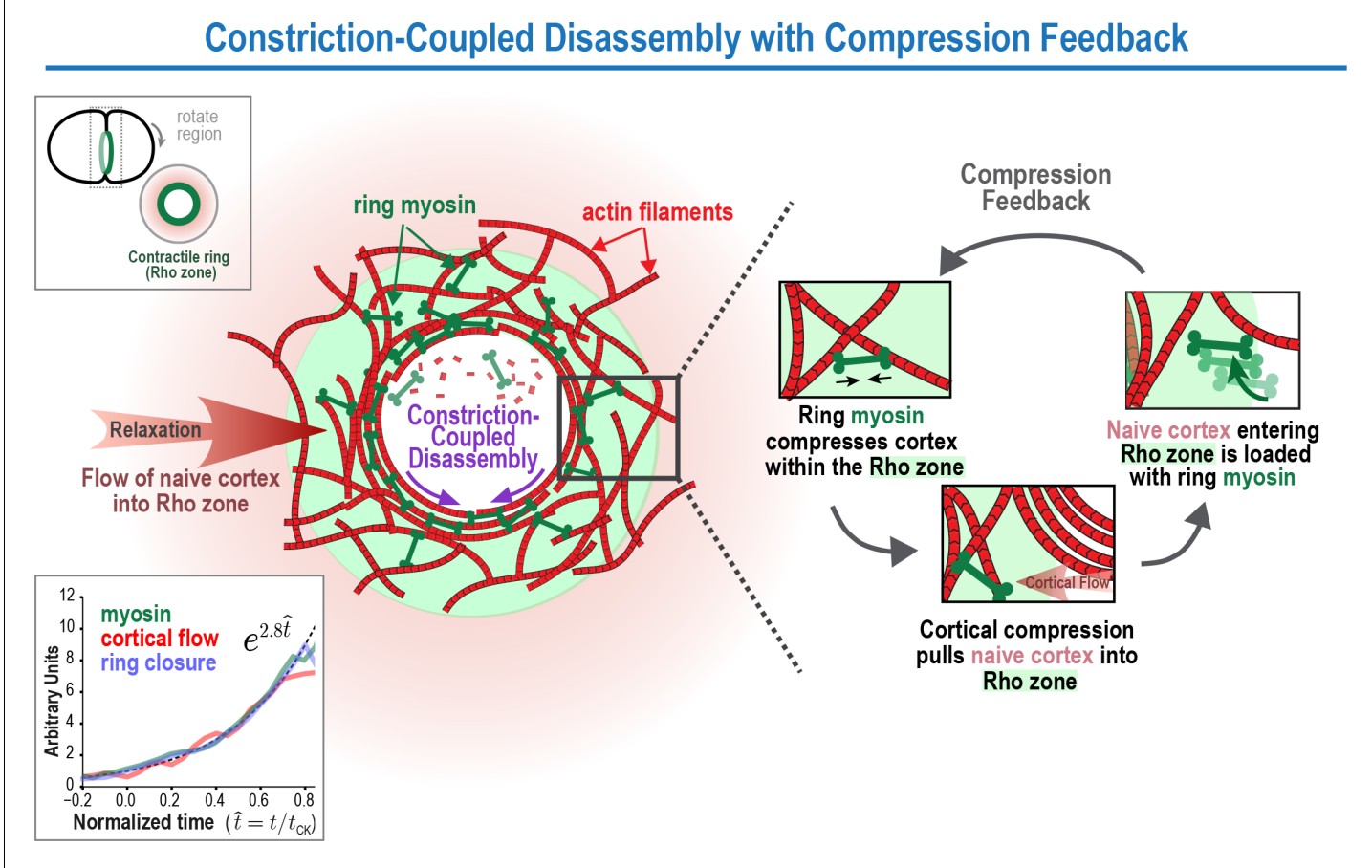

**Figure 8.** The Constriction-Coupled Disassembly with Compression Feedback model for contractile ring dynamics during cytokinesis. Polar relaxation allows ring myosin to compress cortical surface along the axis perpendicular to the ring. Compression pulls naive cortex not previously exposed to RhoA-based signaling into the contractile ring/Rho zone, where it is loaded with myosin and other contractile ring components. Feedback between ring myosin and compression-driven cortical flow leads to an exponential increase in the per-unit-length amount of ring myosin that maintains the high overall closure rate as ring perimeter decreases.

DOI: https://doi.org/10.7554/eLife.36073.021

a global upregulation of surface tension could trigger a flow of tension-generating elements toward the equator that would compress into a circular band and initiate a feedback loop (*Greenspan, 1978*; *Swann and Mitchison, 1958*; *Taber, 1995*; *White and Borisy, 1983*; *Wolpert, 1960*; *Zinemanas and Nir, 1987*, *1988*). However, in contrast to these early models, which proposed that polar relaxation would be sufficient for ring assembly, the feedback loop that we propose requires both polar relaxation and continuous loading of contractile ring components onto the cortex pulled into the equatorial Rho zone by compression. Thus, although polar relaxation allows the cortex within the ring to compress along the axis perpendicular to the ring, the ring is locally fed by RhoA-based signaling acting on the equatorial cortex. We note that this mechanism would also explain why the furrow would slide or regress and reform following repositioning of the spindle (*Rappaport, 1985*). Displacing the spindle to one side of the ring would reposition the Rho zone, causing components to be loaded onto the cortex entering the ring on one side but not the other, generating a force imbalance that would slide the ring over to align it with the spindle.

In addition to maintaining a high overall constriction rate to facilitate timely cell content partitioning, a second advantage of the feedback-based mechanism that we propose is that it would render the ring robust to internal or external mechanical challenges, such as cell-cell contacts, obstacles in the crowded cell interior, or defects in the cytokinesis machinery. If the ring stalled due to any of these causes, the feedback loop between ring myosin and compression-based flow along the

direction perpendicular to constriction would lead to the progressive build-up of contractile ring components until they reached a level where the obstacle could be overcome and constriction would again be able to proceed. Concentrating components via retention in the around-the-ring direction would not have this property, since successful constriction would be required for an increase in component levels. We note that ring-directed cortical flows similar to the ones we document here have also been observed in the context of wound healing (*Mandato and Bement, 2003*), where they could potentially serve a similar function in allowing the cell to ramp up contractile force and achieve wound closure.

The experimental basis for our model is our analysis of cortical dynamics, which indicates that the compression of cortical surface that initiates during contractile ring assembly (*Figure 3*; *Reymann et al., 2016*), persists throughout constriction, resulting in a continuous flow of cortical surface into the ring. A second key finding is that the per-unit-length amount of ring myosin and anillin and the per-unit-length rates of cortical compression and ring constriction increase with the same exponential kinetics, suggesting control by positive feedback. We note that it remains possible that there is a distinct source of positive feedback (other than between ring myosin and cortical compression as we propose) that controls myosin recruitment, and that myosin levels in turn control the rates of constriction and cortical compression. However, since our data indicate that cortical surface is compressed within the ring, such a model would need to invoke an as yet uncharacterized process to explain why compression of the cortex within the ring does not increase the concentration of ring components, as well as detail the nature of the alternative positive feedback loop that controls ring component accumulation. We note that compression within the ring along the direction perpendicular to the ring is also consistent with work in *S. pombe*, which has shown that contractile ring assembly occurs via a similar acto-myosin based compression of an equatorial band of nodes into a compact ring along the long axis of the cell (*Vavylonis et al., 2008*; *Wu et al., 2006*). However, in contrast to *S. pombe* where ring assembly and constriction occur in distinct phases, our model predicts that in animal cells, the accumulation of ring components due to compression along the direction perpendicular to the ring is ongoing and serves to accelerate the per-unit-length constriction rate as the ring closes.

## Polar relaxation enables cortical compression within the ring along the axis perpendicular to the ring

Monitoring cortical dynamics in combination with laser ablation experiments indicates that the polar cortex is distinct from the cortex in the intervening region between the contractile ring and the poles. The polar cortex expands in response to tension generated by the constricting ring, whereas the intervening cortex flows toward the ring without expanding. One possibility is that the polar cortex is less stiff than the rest of the cortex, causing it to stretch and thin in response to ring constriction-induced tension. Alternatively, the polar cortex could turnover more rapidly, leading to a higher rate of surface renewal after stretching. A third possibility is that the polar cortex is more prone to rupture, repair of which would locally increase cortical surface. Consistent with this last idea, blebs have been reported at the cell poles in cultured vertebrate and *Drosophila* cells, where they have been proposed to release tension at the poles (*Hickson et al., 2006*; *Sedzinski et al., 2011*). The distinct mechanical properties of the polar cortex suggest that its composition could be different from that of the adjacent cortex. This idea is consistent with both older work suggesting the existence of mechanisms that clear contractile ring proteins from the poles (*Bement et al., 2005*; *Chen et al., 2008*; *Foe and von Dassow, 2008*; *Murthy and Wadsworth, 2008*; *von Dassow, 2009*; *Werner et al., 2007*; *Zanin et al., 2013*) and recent studies that have begun to uncover molecular mechanisms that may drive clearing. Work in *C. elegans* has demonstrated the existence of a mechanism in which Aurora A, localized to astral microtubules by association with its activator TPXL-1, actively clears contractile ring proteins from the polar cortex (*Mangal et al., 2018*). A reduction in f-actin intensity at the cell poles due to delivery of a phosphatase by segregating chromosomes has also been reported in *Drosophila* cells (*Rodrigues et al., 2015*). Understanding how the polar cortex is different in molecular and mechanical terms, and the mechanisms that generate these differences are important goals for future work.

Cleaving sea urchin embryos exhibit constriction kinetics essentially identical to those during the first division of the *C. elegans* embryo (*Mabuchi, 1994*). Pioneering work measuring the distance between surface-adhered particles and the behavior of pigmented cortex-associated granules

(*Dan, 1954*; *Dan and Dan, 1940*; *Dan et al., 1938*), indicated that sea urchin embryos also exhibit a similar pattern of cortical expansion during ring constriction, in this case, a wave of cortical expansion that initiates at the poles and propagates through to the region adjacent to the furrow (*Dan et al., 1938*; *Dan and Ono, 1954*; *Dan et al., 1937*; *Gudejko et al., 2012*; *Swann and Mitchison, 1958*). Cortical compression and expansion have not been mapped in vertebrate cells; however, monitoring of fluorescent latex spheres adhered to cell surface proteins (*Fishkind et al., 1996*; *Wang et al., 1994*), injected stabilized fluorescent actin filaments (*Cao and Wang, 1990*), and fluorescently labeled myosin II (*DeBiasio et al., 1996*) all revealed concerted cortical flow towards the division plane in the equatorial region of the cell that contrasted with random surface movements at the cell poles. These observations suggest that feedback in which relaxation enables compression-driven cortical flow may be a conserved feature of animal cell cytokinesis.

It is worth noting that the model we propose suggests a resolution to a long-standing debate as to whether contractile ring components are delivered into the ring via cortical flow (*Cao and Wang, 1990*; *DeBiasio et al., 1996*; *Fishkind et al., 1996*; *Wang et al., 1994*) or recruited de novo from the cytoplasm downstream of RhoA-based signaling (*Vale et al., 2009*; *Yumura, 2001*; *Zhou and Wang, 2008*). In our model, following anaphase onset contractile ring components are initially recruited to the equatorial cortex de novo, as has been observed (*Vale et al., 2009*; *Yumura, 2001*; *Zhou and Wang, 2008*), but then component levels are amplified by a feedback loop that involves flow and compression of cortical surface as well as de novo component loading. Thus, in our model, both de novo loading of components by Rho-based signaling and compression-driven flow contribute to the change in component levels in the ring during constriction.

## The Constriction-Coupled Disassembly with Compression Feedback model as a tool to describe the evolution of the contractile ring

To quantitatively explore the idea that a feedback loop between the amount of ring myosin and compression-driven flow of cortical surface into the ring drives component accumulation during constriction, we developed an analytical mathematical framework. The model consists of three equations with three model parameters that describe this feedback and can recapitulate our experimental results. In addition to describing the processes underlying the evolution of the contractile ring, the Constriction-Coupled Disassembly with Compression Feedback model provides a simple framework that can be used to analyze the consequences of molecular perturbations. An additional interesting future direction will be to use parameter changes derived from the model as input for a finite-element model (similar to *Turlier et al., 2014*) in order to predict the evolution of cell shape given an *a priori* knowledge of cortical and contractile ring dynamics.

## Materials and methods

**Key resources table**

| Strain name | Genotype | Reference |
|---|---|---|
| OD821 | *ltSi200[pOD1997; Pnmy-2::nmy-2::gfp; cb-unc-119(+)] II; unc-119(ed3) III* | This study |
| OD857 | *ltSi200[pOD1997; Pnmy-2::nmy-2::gfp; cb-unc-119(+)] II; unc-119(ed3); ruIs32[pAZ132; pie-1/GFP::histone H2B] III* | This study |
| OD858 | *ltSi803[pOD1998; Parx-7::GFP::arx-7; cb-unc-119(+)] II; unc-119(ed3) III;* | This study |
| LP162 | *nmy-2(cp13[nmy-2::gfp + LoxP]) I* | *Dickinson et al., 2013* |
| OD95 | *unc-119(ed3) III; ltIs37 [pAA64; Ppie-1::mCherry::his-58; unc-119(+)] IV; ltIs38 [pAA1; Ppie-1::GFP::PH(PLC1delta1); unc-119 (+)]* | *Essex et al., 2009* |
| OD3011 | *ltSi1123[pSG017; Pani-1::GFP::ani-1 RE-encoded-exon5::ani-1 3'-UTR; cb unc-119(+)]II;unc-119(ed3)III* | This study |
| GOU2047 | *cas607[arx-2::gfp knock-in] V* | *Zhu et al., 2016* |

### *C. elegans* strains used in this study

The *C. elegans* strains listed in the table were maintained at 20°C using standard methods. OD821, OD3011, and OD858, expressing NMY-2::GFP, GFP::anillin, and GFP::ARX-7, respectively, were generated using a transposon-based strategy (MosSCI; [*Frøkjaer-Jensen et al., 2008*]).

Genomic regions encoding *nmy-2* (including 2079 bp and 1317 bp up and downstream of the stop codon, respectively), *ani-1* (including 2015 bp and 1215 bp up and downstream of the stop codon), and *arx-7* (including 3056 bp and 634 bp up and downstream of the stop codon) were cloned into pCFJ151 and sequences encoding GFP were inserted either just before (*nmy-2*) or after (*arx-7* and *ani-1*) the start codon. The single copy *nmy-2* transgene was generated by injecting a mixture of repairing plasmid (pOD1997, 50 ng/µL), transposase plasmid (pJL43.1, Pglh-2::Mos2 transposase, 50 ng/µL), and fluorescence selection markers (pGH8, Prab-3::mCherry neuronal, 10 ng/µL; pCFJ90, Pmyo-2::mCherry pharyngeal, 2.5 ng/µL; pCFJ104, Pmyo-3::mCherry body wall, 5 ng/µL) into EG6429 (ttTi5605, Chr II). Single copy *ani-1* and *arx-7* transgenes were generated by injecting a mixture of repairing plasmid (pSG017 (*ani-1*) or pOD1998 (*arx-7*), 50 ng/µL), transposase plasmid (CFJ601, Peft-3::Mos1 transposase, 50 ng/µL), selection markers (same as for *nmy-2* strain) and an additional negative selection marker (pMA122; Phsp-16.41::peel-1, 10 ng/µL) into EG6429 (ttTi5605, Chr II). After 1 week, progeny of injected worms were heat-shocked at 34°C for 2–4 hr to induce PEEL-1 expression and kill extra chromosomal array containing worms (*Seidel et al., 2011*). Moving worms without fluorescent markers were identified and transgene integration was confirmed in their progeny by PCR spanning both homology regions in all strains.

## *C. elegans* RNA-mediated interference

Double stranded RNA (dsRNA) targeting *arx-2* (K07C5.1) at a concentration of 1.7 mg/ml was generated by synthesizing single-stranded RNAs in 50 µL T3 and T7 reactions (MEGAscript, Invitrogen, Carlsbad, CA) using cleaned DNA template generated by PCR from N2 DNA using the oligos (TAA TACGACTCACTATAGGTCAGCTTCGTCAAATGCTTG and AATTAACCCTCACTAAAGGTGCAA TACGCGATCCAAATA). Reactions were cleaned using the MEGAclear kit (Invitrogen, Carlsbad, CA), and the 50 µL T3 and T7 reactions were mixed with 50 µL of 3 × soaking buffer (32.7 mM Na$_2$HPO$_4$, 16.5 mM KH$_2$PO$_4$, 6.3 mM NaCl, 14.1 mM NH$_4$Cl), denatured at 68°C for 10 min, and then annealed at 37°C for 30 min to generate dsRNA. L4 hermaphrodite worms were injected with dsRNA and allowed to recover at 16°C for 44–50 hr prior to imaging.

## Generating a 4D map of cortical flow

Cortical flow was monitored in embryos expressing myosin::GFP obtained from adult hermaphrodites by dissection. Embryos were mounted followed by sealing with a coverslip on double thick (1 mm) low percentage agarose (0.5%) pads to prevent compression that biases the initial angle of furrow ingression (*Figure 1—figure supplement 1B*). Images were acquired on an inverted microscope (Axio Observer.Z1; Carl Zeiss) equipped with a spinning-disk confocal head (CSU-X1; Yokogawa) and a 63 × 1.40 NA Plan Apochromat lens (Zeiss) using a Hamamatsu Orca-ER digital camera (Model C4742-95-12ERG, Hamamatsu photonics). Images were collected using custom software, written in Python, that utilizes the Micro-Manager (open source software, *Edelstein et al., 2014*) microscope control library. A 3 × 0.75 µm z-series was collected (400 ms exposure, 10–20% laser power) every 2 s. After 15 time points, a 15 × 1 µm z-stack, offset by 3 µm from the cortical surface, was imaged to monitor the position of the closing contractile ring. The entire imaging series was repeated every 36 s until the end of cytokinesis. Cortical flow was measured in maximum intensity projections of the 3 × 0.75 µm z-stacks of the cortical surface, after orientation of the images to place the embryo anterior at the top and the posterior at the bottom, by correlating myosin fluorescence between consecutive images using Gunnar Farnebäck's algorithm (*Farnebäck, 2003*) implemented within the openCV library with a 30-pixel window size. The threshold was calculated for every image by maximizing the ratio of total intensity inside a 200 × 350 pixel box positioned in the center of the embryo to the total intensity outside that box.

## Measurement of contractile ring position and size

Automated methods were employed to identify the edges of the embryo, determine the position of the contractile ring, and reconstruct the rings for each time point in an end-on view to determine the initial ingression axis (*Figure 1—figure supplement 2*). Ring size and position were determined using custom Python software that: (1) identifies the orientation of the anterior-posterior (AP) axis and rotates the embryo to place the embryo anterior at the top and the embryo posterior at the bottom, (2) finds the embryo center in different x-z planes along the AP axis and calculates embryo

radius, and (3) calculates the radius of the contractile ring and determines its position within the division plane. Details of each step are outlined below.

## Orienting embryos with their anterior end to the top

Acquired z-plane images were convolved with a 10-pixel Gaussian kernel to reduce noise. An optimal signal threshold that partitioned the embryo interior from exterior was identified by finding a local minimum in the intensity histogram that produced a binary mask with the expected area ($\sim$120,000 $\pm$ 50,000 pixel$^2$). The orientation of the AP axis was identified by fitting an ellipse to the thresholded area in the middle plane of the z-stack. The anterior side was identified by higher cortical myosin fluorescence and all images were rotated to place the embryo anterior at the top of the image and the embryo posterior at the bottom.

## Defining the central axis of embryo and determining embryo width

The central axis of the embryo was defined by drawing a horizontal line across the oriented embryo at the midpoint between its anterior and posterior ends and identifying the first and last points along this line with signal above the threshold for each z-plane. The identified pixels were virtually projected in an end-on (x-z) view and fit to a circle by minimizing residuals. To account for fluctuations in the embryo boundary due to noise and fluorescence variation, the procedure was repeated nine more times after shifting the position of the horizontal line toward the anterior pole by 10 pixels, covering approximately 1/5 of the embryo length (500 pixels). The position of the AP axis and the radius of the embryo were determined by averaging the 10 measurements.

## Measuring contractile ring size and position

As illustrated for the central plane images shown in *Figure 1—figure supplement 2*, the position of the contractile ring was determined by identifying pairs of points with the highest myosin fluorescence intensity on the opposite edges of the embryo in each z-plane that were not more than 20 pixels apart in the horizontal direction and were located at a y-axis position near the embryo middle. Contractile ring radius and position were determined by projecting the points to generate an end-on (x-z) view and fitting the data with a circle. The ring fit was iteratively improved by calculating predicted positions of myosin fluorescence at the ring in each z-plane using initially fitted parameters. Intensity maxima within 5 pixels of the predicted location were identified and the ring was refit. The initial guesses for the contractile ring size and position at the next time point were estimated from the previously calculated ring values. The algorithm restricted ring position fluctuations to 20 pixels along anterior-posterior axis and the size was estimated assuming constant rate of ring constriction. The automatic ring measurements were manually confirmed for each embryo. The initial ingression axis was determined as illustrated (*Figure 1—figure supplement 2*) by fitting a line through the centers of the rings with a normalized ring size ($\bar{R} := R/R_{emb} > 0.3$).

## Embryo time alignment for averaging

Sequences from individual embryos were time aligned by defining zero time ($t_0$) and the total time of cytokinesis ($t_{CK}$) for each embryo, and normalizing time by $t_{CK}$ prior to averaging, $\hat{t} := \frac{t-t_0}{t_{CK}}$. An initial determination of $t_0$ and $t_{CK}$ was made by fitting a line to the plot of normalized ring size ($\bar{R}(t) := R/R_{emb}$) versus time between 30% and 80% closure for each embryo as outlined in Figure 1A. Extrapolation of this line for each embryo defined $t_0$ as the time where the fitted line intersects 1, and the time of cytokinesis, $t_{CK}$ as the time where the fitted line intersects 0. Due to the small number of measurements from each embryo available for fitting (3-5 values where $0.8 > \bar{R} > 0.3$), the values of $t_0$ and $t_{CK}$ were refined by fitting $\bar{R}(\hat{t})$ for each embryo to the average dimensionless ring size, $< \bar{R} > (\hat{t})$. Calculation of the average dimensionless ring size was performed in iterative manner. The time for each embryo was aligned by $t_0$ and normalized by $t_{CK}$ using estimates from the fitted line in the first iteration. The average dimensionless ring size ($< \bar{R} > (\hat{t})$) was calculated by averaging normalized ring sizes of all embryos at corresponding normalized time. Contractile ring size was approximated for intermediate time points by linear interpolation. In further iterations, $t_0$ and $t_{CK}$ were refined for every embryo by minimizing the residuals between its normalized ring size, $\bar{R}(\hat{t})$, and the average dimensionless ring size, $< \bar{R} > (\hat{t})$, throughout the entire timecourse of cytokinesis, thus increasing the number of time points available for fitting $t_0$ and $t_{CK}$ (6-10 values per embryo). After

refining time alignment and normalization for each embryo, average dimensionless ring size was re-calculated and $t_0$ and $t_{CK}$ were refined for each embryo again. The refinement process was repeated until changes in average dimensionless ring size, $< \bar{R} > (\hat{t})$, were smaller than 0.001 on average (achieved within a few iterations). The collective fitting of all $t_0$ and $t_{CK}$ at every iteration was performed under restriction that the line fit through $< \bar{R} > (\hat{t})$ between 0.8 and 0.3 intercepted 0 at $\hat{t} = 0$ and 1 at $\hat{t} = 1$. This restriction ensured that $t_0$ and $t_{CK}$ determined from fits of individual embryos to the average ring size would be consistent with their original definition. The dimensional ring kinetics, $< R > (t)$, can be recovered using the following equation

$$<R>(t) = <R_{emb}><\bar{R}>(\hat{t}<t_{CK}>), \tag{9}$$

where $< R_{emb} > = 14.7 \pm 0.7\ \mu m$ and $< t_{CK} > = 200 \pm 30\ s$ are average embryo radius and time of cytokinesis accordingly.

## Cortical flow averaging

Cortical flow averaging was performed after spatial and temporal alignment of data collected in different embryos (n=93 embryos from 93 worms filmed over the course of 5 days for control, *Video 2*; n=68 embryos from 68 worms filmed over the course of 4 days for *arx-2(RNAi)*, *Video 3*). The number of embryos was chosen to achieve at least 10-fold coverage for all areas of the cortical map for controls and 5-fold coverage for *arx-2(RNAi)*. Linear interpolation was used to approximate the flow between consecutive time points. Because our imaging regime required periodic z-stack acquisition to determine the trajectory of ring closure, no flow approximation was done during those time periods (~6 s gap every 30 s). The flow data for each time point was represented as a set of vectors with direction and magnitude corresponding to the direction and magnitude of the cortical flow at the base of the vector. The base of each vector had two spatial coordinates: $x$, the position along the anterior-posterior axis (where the position of the contractile ring was defined as 0), and $\theta$, the angular position relative to the initial ingression axis (defined as described in *Figure 1A* and *Figure 1—figure supplement 2*). We note that mitotic exit is accompanied by a brief (~50-60s) period of rotational flow ([*Naganathan et al., 2014*; *Schonegg et al., 2014*]; see *Video 1*), which dissipates soon after initiation of cytokinesis (~$\hat{t}$=0.2-0.3). As this rotational contribution is not relevant here, we removed it by averaging the data from the right and left halves of the embryo (in an end-on view), allowing us to focus on rotation-independent flows. Thus, the flow with angular positions greater than 180 degrees was mirrored in angular direction

$$f_\theta(\hat{t},\ x,\ \theta > 180) \rightarrow -f_\theta(\hat{t}, x,\ 360 - \theta), \tag{10}$$

$f_\theta$ is the angular component of the flow vector $\vec{f}$. The flows were normalized by the embryo size and cytokinesis rate $\vec{\bar{f}}(\hat{t}, x,\ \theta) := \frac{t_{CK}}{R_{emb}} \vec{f}(\hat{t}, x,\ \theta)$ and averaged according to its position and time

$$<\vec{\bar{f}}> (t, x,\ \theta) = \frac{\sum_{emb} \vec{\bar{f}}(\hat{t}, x,\ \theta)}{N_{emb}}. \tag{11}$$

Calculation of expected cortical surface flow profiles

To aid in the interpretation of experimental results, expected profiles for cortical surface movement were calculated for defined patterns of cortical surface increase and plotted (*Figure 1B* and *Figure 1—figure supplement 3*). The general form of surface movement velocity is given by the following equation

$$v(x) = \int_0^x g\left(x'\right) dx' + u, \tag{12}$$

where $g(x)$ is the amount of cortical surface gain and $u$ is the velocity of asymmetric ring movement, which could be positive or negative, depending on whether the ring is moving towards or away from the surface. From equation (12) we obtain the following predictions

Uniform surface increase: $v(x) = Cx + u$;
Polar surface increase: $v(x) = C + u$;

Behind the ring surface increase: $v(x) = u$ (if the asymmetry of cytokinetic furrowing arises due to global surface movement) or $v(x) = 0$ (if the asymmetry in surface increase is related to the asymmetric furrowing).

## Cortical laser ablation

Cortical laser ablations, presented in *Figure 2*, were performed using a robotic laser microscope system (RoboLase; *Botvinick and Berns, 2005*). Embryos expressing myosin::GFP were mounted using standard procedures. A cortical cut, approximately 10 µm long, was made on the anterior side of the embryo when the ring was at ~50% closure (7 µm radius). The cut was confirmed by comparison of cortical fluorescence images before and after the cut and was considered successful if the foci moved away from the cut area (~3.5 µm distance), indicating cortical tension release. Contractile ring closure rate was calculated by measuring the difference in ring sizes before and after the cut, assessed from two 4 × 2 µm z-stacks acquired immediately before the cut and 13 s later. Errors in measuring the radius at the two timepoints were determined from the procedure used to fit the data to a circle and were propagated to determine the errors in the constriction rate measurements for individual embryos; mean errors are S.E.M. The cortical opening after ablation was approximately 35 µm$^2$ and, at the time of the cut, the rings were approximately 14 µm in diameter. If the ring size is determined by the available cell surface, then this gain in surface area should result in additional area increase of the division plane. Assuming that the surface area gain resulting from the lesion is distributed between the two cells it would result in ~18 µm$^2$ increase in the division plane, which corresponds to an additional ~0.4 µm reduction in ring radius. This reduction in size would lead to an increase in constriction rate of ~0.03 µm/s, during the 13 s interval between image acquisitions. The experiment was repeated 19 times for no cut condition, 14 times for parallel cut, and 15 times for perpendicular cut. All imaging was performed over the course of 5 days. The number of embryos was chosen to achieve sufficient accuracy in the determination of mean ring closure rates to assess whether it was altered by the cuts.

## Calculation of the surface area flowing into the division plane

We calculated the amount of surface area flowing into the division plane from flow measurements made 7 µm away from the position of the furrow on the anterior and posterior sides (as illustrated in *Figure 3B*). The rate of the surface flow is

$$\frac{dA_{surf}}{d\hat{t}}(\hat{t}) = 2R_{emb}\int_0^\pi <\vec{f}>(\hat{t}, x_0, \theta)d\theta, \tag{13}$$

where $x_0$ is -7 µm and 7 µm for the rate of flow from the anterior or the posterior sides, respectively. The total amount of surface area that entered the division plane from any time $\hat{t}_0$ to $\hat{t}$ is obtained by integrating equation (13) over time

$$A_{surf}(\hat{t}) = \int_{\hat{t}_0}^{\hat{t}} \frac{dA_{surf}}{d\hat{t}}\bigg|_{ant} + \frac{dA_{surf}}{d\hat{t}}\bigg|_{post}(t')dt'. \tag{14}$$

The increase in area of the division plane was calculated as following

$$A_{div\,plane}(\hat{t}) = 2\pi(<R>^2(\hat{t}_0) - <R>^2(\hat{t})). \tag{15}$$

In *Figure 3B*, we used $\hat{t}_0 = -0.2$. The cortical surface area compressed in the ring can be inferred from the difference between the surface area entering the division plane and the area of the division plane

$$A_{comp}(\hat{t}) := A_{surf}(\hat{t}) - A_{div\,plane}(\hat{t}). \tag{16}$$

## Division plane imaging

For quantification of myosin::GFP and GFP::anillin amounts in the contractile ring, adult worm dissection and one-cell stage embryos imaging was performed in a custom microdevice (*Carvalho et al., 2011*). The device was mounted on an inverted microscope (Axio Observer.Z1; Carl Zeiss) and embryos were imaged with a 63 × 1.4 NA Plan Apochromat objective using an electron-multiplying charge-coupled device camera (QuantEM:512SC, Photometrics; 100 ms exposure, EM gain set to

500, 10% laser power). Division planes were reconstructed from $40 \times 0.5$ µm z-stacks collected every 30 s after background subtraction and attenuation correction. All imaging was done at 20°C.

### Contractile ring photo-bleaching and imaging

One-cell stage embryos were mounted in microdevices as for division plane imaging and four-cell stage embryos were mounted on slides with 2% agarose pads. Embryos were imaged on a Nikon TE2000-E inverted microscope equipped with a $60 \times 1.40$ NA objective, an EM-CCD camera (iXon; Andor Technology; EM-Gain = 220, Exposure = 100 ms), and a krypton-argon 2.5 W water-cooled laser. For one-cell stage embryos, division planes were reconstructed from $30 \times 1$ µm stacks acquired every 20 s with 20% laser power and photo-bleaching was performed by 2 sweeps of a 488 nm laser with 100% power and 500 µs dwell time. For four-cell stage embryos, division planes were reconstructed from $16 \times 1$ µm stacks acquired every 10 s with 50% laser power and photo-bleaching was performed by 2 sweeps of a 488 nm laser with 100% power and 100 µs dwell time. For four-cell stage embryos, the time between the prebleached and first postbleached images was 6 s.

### Estimation of depth attenuation

To estimate depth attenuation within the division plane, we quantified the intensity of the division plane in two cell embryos expressing a GFP-tagged probe expected to be uniformly present on the plasma membrane. From each image, we subtracted a background intensity calculated as the average value inside two $11 \times 11$ µm rectangles positioned 2 µm away from the division plane inside the anterior and posterior cells (*Figure 3—figure supplement 3*). The division plane intensity profile was obtained by performing a 30-pixel maximum intensity projection along the AP axis, with the division plane positioned approximately in the middle (*Figure 3—figure supplement 3*). The intensity profiles in z from 13 embryos were fitted to an exponential using the same characteristic attenuation depth for all embryos

$$I = I_0 e^{-z/z_{att}}, \tag{17}$$

which yielded a characteristic depth of attenuation, $z_{att}$, of 15 µm.

### Quantification of myosin and anillin intensity in the contractile ring and on the cortex

For embryos at the one-cell stage, myosin::GFP and GFP::anillin intensities in the contractile ring and on the cortex were quantified in 40 x 0.5 µm z-stacks containing the ring after correction for depth attenuation and subtraction of background fluorescence. Average intensity along the ring was calculated across a set of embryos in 30 degree arcs (for myosin::GFP, n=36 embryos from 18 worms filmed over 5 days; for anillin::GFP, n= 26 embryos from 14 worms filmed over 4 days). The number of embryos was chosen to determine mean fluorescence with sufficient accuracy to derive appropriate conclusions. Positions along the ring were referenced based on the angle between the line from the position on the ring to the ring center and the initial ingression axis. Linear interpolation in time was used for every embryo to estimate intensity in the intermediate time points to perform averaging. Measured intensities were divided by arc length and averaged between different embryos to obtain mean GFP fluorescence per-unit-length for different angular ranges and the average for all angles. Total ring GFP fluorescence was calculated by integrating over ring perimeter. Cortical intensities were quantified by choosing the time point with the ring size closest to $\bar{R} = 0.8$ and measuring total fluorescence in the 15$^{th}$ plane after correction for depth attenuation and subtraction of background fluorescence.

Measurements of myosin::GFP fluorescence in the ring at the four-cell stage were performed as described in *Carvalho et al. (2009)*. However, background fluorescence was determined as the mean fluorescence within a variable size circle at least 10 pixels in diameter, instead of fixed at 10 pixels, to improve measurement quality.

### Derivation of the constriction-coupled disassembly with compression feedback model for cytokinesis

The Constriction-Coupled Disassembly with Compression Feedback model formalizes the following conceptual view of cytokinesis: After anaphase onset, spindle-based signaling patterns the cortex,

generating an equatorial zone where RhoA promotes the recruitment of contractile ring components (the Rho zone). Within the Rho zone, myosin engages with actin to exert an isotropic force that compresses the cortical surface, resulting in uniform compression across this region, as is observed experimentally (*Figure 3A*). Due to polar relaxation, the compressing cortex pulls naive cortex (defined as cortex outside the Rho zone) into the Rho zone. We propose that the new cortical surface that flows into the Rho zone as a result of compression is also loaded with contractile ring components. Thus, a feedback loop is established along the direction perpendicular to the ring, in which myosin in the ring compresses cortical surface, which pulls more surface that is loaded with myosin into the ring. Consistent with our division plane photobleaching experiment at the one-cell stage (*Figure 6C*) and our prior work (*Carvalho et al., 2009*), we propose that in the around-the-ring direction constriction-coupled disassembly leads to a reduction in ring components in proportion to the reduction in length. Thus, constriction does not alter the per-unit-length amount of ring components. Changes in myosin levels are therefore determined solely by the rate of flow of naive cortex into the Rho zone along the direction perpendicular to the ring, which can be solved as a one-dimensional problem. We assume that the rate of compression of cortical surface (between $x$ and $x+dx$) is proportional to local myosin concentration, $m(x,t)$, which exerts stress onto the actin network resulting in

$$\frac{\delta\varepsilon}{\delta t}(x,t) = -\alpha m(x,t), \tag{18}$$

where $\varepsilon$ is the cortical strain (i.e. change in length of cortical surface per-unit-length) and $\alpha$ is a proportionality constant that reflects the ability of the cortex to be compressed by ring myosin. The velocity of cortical surface movement is obtained from the following relationship (see also equation (12)).

$$v(x,t) = \int_0^x \frac{\delta\varepsilon}{\delta t}\left(x',t\right)dx'. \tag{19}$$

The conservation of mass for myosin flow results in the following

$$\frac{\partial m}{\partial t}(x,t) = -\frac{\partial}{\partial x}(m(x,t)v(x,t)) = \frac{\partial}{\partial x}\left(m(x,t)\int_0^x \alpha m\left(x',t\right)dx'\right). \tag{20}$$

If we integrate *equation (20)* over *x* on (-*w, w*) domain we obtain

$$dM_{ring}(t)/dt = \alpha m_{rho}M_{ring}(t), \tag{21}$$

where $M_{ring}(t) := \int_{-w}^{w} m(x,t)dx$ is the total per-unit-length amount of ring myosin engaged in compression, $2w$ is the width of the contractile ring/Rho zone where myosin is engaged and compressing cortex and $m_{rho} := m(w,t)$ is the concentraton of myosin loaded onto the cortex when it enters the rho zone. The velocity of flow of naive cortex into the rho zone is

$$v_{flow}(t) = \alpha M_{ring}(t)/2, \tag{22}$$

The one half is included to account for the fact that flow comes in from both sides. The solution of *Equation (21)* is

$$M_{ring}(t) = M_{0\,ring}\,e^{t/\tau}, \tag{23}$$

where we define the characteristic time of myosin accumulation, $\tau$, as $\frac{1}{\alpha m_{rho}}$. Note that the total amount of myosin in the ring will be the amount of engaged ring myosin plus an added baseline that would include any myosin not involved in compression (see *Equation 5*). We assume the per-unit-length rate of ring shrinkage is proportional to the amount of ring myosin, as observed in our data,

$$\frac{1}{R}\frac{dR}{dt} = -\beta M_{ring}(t), \tag{24}$$

where $\beta$ is a proportionality coefficient that reflects the ability of the ring to be constricted by ring myosin. Using *Equations (23) and (24)*, we obtain the dynamics of contractile ring size over time

$$\bar{R}(t) = \bar{R}_{ini} e^{-\beta\tau M_{0ring}\exp(t/\tau)}, \tag{25}$$

where $\bar{R}_{ini}$ is the dimensionless characteristic size of the ring; essentially the radius at minus infinity if the same exponential process controlling contractile ring assembly extended back in time infinitely. Instead, in vivo cytokinesis initiates when spindle-based signaling activates RhoA on the equatorial cortex leading to the abrupt recruitment of contractile ring components. If the time frame of reference is chosen so that $t = 0$ is cytokinesis onset immediately following the initial patterning of the cortex by RhoA, $M_{0\,ring}$ is the amount of ring myosin immediately following this event and the initial size of the ring is

$$\bar{R}_0(t) = \bar{R}_{ini} e^{-\beta\tau M_{0\,ring}}. \tag{26}$$

To facilitate future use of our model for analysis of contractile ring closure data, we use the time frame of reference where $t = 0$ is the point of 50% closure (i.e. $\bar{R}(t = 0) = \frac{1}{2}$), an easily identifiable time point that does not rely on exact assessment of the precise onset of cytokinesis. In this reference, $M_{0\,ring} = \frac{\ln(\bar{R}_{ini})}{\beta\tau}$, and by defining dimensionless velocity as $\bar{v} := \tau v$, we obtain equations (4-8). Note that equation (4) can be rewritten in the following way

$$\bar{R}(\bar{t}) = \bar{R}_{ini} e^{-\frac{1}{R}\frac{dR}{d\bar{t}}}, \tag{27}$$

where $\bar{t} := t/\tau$. This relationship implies that in this dimensionless time, where $\bar{R}(\bar{t} = 0) = \frac{1}{2}$, any two rings of the same size have the same dimensionless constriction rate.

## Data availability

Key source data is available from the Dryad repository.

## Code availability

The custom computer code used in this study is freely available from GitHub (*Khaliullin, 2018*; copy archived at https://github.com/elifesciences-publications/cytokinesis).

## Acknowledgements

This work was supported by a fellowship from the Jane Coffin Childs Memorial Fund to RNK and grants to MWB from AFOSR (FA9550-08-1-0284) and the Beckman Laser Institute Foundation. JSG-C was supported by the University of California, San Diego Cancer Cell Biology Training Program (T32 CA067754). AD and KO receive salary and other support from the Ludwig Institute for Cancer Research. We would also like to thank Michael Glotzer for discussions that helped us align our model with current thinking about the Rho zone.

## Additional information

### Funding

| Funder | Grant reference number | Author |
| --- | --- | --- |
| Ludwig Institute for Cancer Research | | Arshad Desai<br>Karen Oegema |
| Beckman Laser Institute and Medical Clinic | | Michael W Berns |
| Air Force Office of Scientific Research | FA9550-08-1-0284 | Michael W Berns |
| Jane Coffin Childs Memorial Fund for Medical Research | | Renat N Khaliullin |
| National Institutes of Health | T32 CA067754 | J Sebastian Gomez-Cavazo |

The funders had no role in study design, data collection and interpretation, or the decision to submit the work for publication.

## Author contributions
Renat N Khaliullin, Conceptualization, Data curation, Software, Formal analysis, Investigation, Visualization, Methodology, Writing—original draft, Writing—review and editing; Rebecca A Green, Visualization, Writing—original draft, Writing—review and editing; Linda Z Shi, Investigation, Methodology; J Sebastian Gomez-Cavazos, Investigation, Constructed the C. elegans strain expressing GFP::Anillin that was used to monitor accumulation of the contractile ring component anillin in this study; Michael W Berns, Conceptualization, Resources, Supervision, Funding acquisition, Methodology; Arshad Desai, Conceptualization, Supervision, Writing—original draft, Writing—review and editing; Karen Oegema, Conceptualization, Resources, Supervision, Funding acquisition, Writing— original draft, Project administration, Writing—review and editing

## Author ORCIDs
Arshad Desai (iD) http://orcid.org/0000-0002-5410-1830
Karen Oegema (iD) http://orcid.org/0000-0001-8515-7514

## Decision letter and Author response
Decision letter https://doi.org/10.7554/eLife.36073.030
Author response https://doi.org/10.7554/eLife.36073.031

## Additional files

### Supplementary files
• Transparent reporting form
DOI: https://doi.org/10.7554/eLife.36073.022

### Data availability
All data generated during this study are included in the manuscript and supporting files.

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
