## [Decision Letter]

[Editors’ note: a previous version of this study was rejected after peer review, but the authors submitted for reconsideration. The first decision letter after peer review is shown below.]

Thank you for submitting your work entitled "Positive feedback between contractile ring myosin and ring-directed cortical flow drives cytokinesis" for consideration by *eLife*. Your article has been reviewed by three peer reviewers, and the evaluation has been overseen by a Reviewing Editor and a Senior Editor. The following individual involved in review of your submission has agreed to reveal their identity: Michael Glotzer (Reviewer #2).

Our decision has been reached after consultation between the reviewers. Based on these discussions and the individual reviews below, we regret to inform you that your work will not be considered further for publication in *eLife* at present.

The reviewers appreciated the novel imaging and computational approaches you are using to understand mechanisms of cytokinetic force generation. The reviewers also found the premise that cortical flow of myosin together with existing myosin in the furrow regulates the rate of furrow contraction interesting. However, a number of issues were raised. These include 1) inconsistencies with what is known in the literature (some experiments with Rho pathway and Nop1 mutants can address these), 2) issues with the theoretical framework and terminologies used (raised by reviewers 1 and 2, which may need experiments and rewriting), 3) aspects of the data fitting and conclusions regarding cortical flow (reviewer 1; may require reanalysis / reconciliation), and 4) referencing (reviewers 2 and 3; should be easily fixed).

If you feel you can address the issues raised with experiments and rewriting / analysis, we will be happy to consider a revised version or as a new submission. In both cases, we will send the paper to the same reviewers.

*Reviewer #1:*

Cleavage furrow ingression during animal cytokinesis is driven by constriction of the actomyosin contractile ring. While major components of the contractile ring and key regulators for its assembly and constriction have been identified, our mechanistic understanding on how it generates force is limited. In their previous publication, based on the observation that the rate of the ring contraction is largely constant and proportional to the initial size of the ring, the Oegema group proposed a 'contractile unit' model, which assumes a presence of a contractile unit with a fixed initial length, which retains myosin while it shortens.

In this manuscript, the Oegema group studied further details of the mechanism of the ring constriction by precisely measuring the flow and total amount of cortical myosin as well as the myosin in the ring during the ring constriction. Based on these analyses as well as the results of laser micro-surgery and genetic manipulations, they propose a mathematical model with three parameters, which was demonstrated to be useful to explain the effects of a perturbation (depletion of Rho-kinase). The quality of image analysis, especially the 4D mapping of the cortical flow, is extremely high with striking number of video data analyzed. However, there are major problems to be addressed before publication.

1) Logic to choose the feedback model instead of the retention model.

The authors' logic for throwing away the retention model is not clear. The similarity between the time courses of the ring-directed cortical flow and the mean ring myosin or anillin per unit length, and the behavior of the ring myosin after FRAP are the observations on which they were based. Good fitting to exponential curves with a common time constant is interesting and is consistent with the cortical flow feedback model. However, the data are not strong enough to tell exponential from hyperbolic curves. The retention model predicts that the ring myosin per unit ring length is proportional to the inverse of the ring radius (∝1/R) (Figure 4C). Under the constant rate of ring constriction, this means that the ring myosin per unit ring length is proportional to 1/(R_unit_ – v∙t) (t: time, v: rate of ring constriction measured by radius), a hyperbola. With the value range used for Figure 3 and 4 (3 to 4-fold increase), it is almost impossible to distinguish an exponential curve from a hyperbolic one. Indeed, as a simplest example, a set of (x, y) calculated by y=1/(1-x) can be fitted with an exponential curve very well (as we can see by an R script below). The authors should explain why the data support exponential increase better than hyperbolic one.

What data tell us are the largely fixed total amount of ring myosin (with limited exchange with flanking cortex or cytoplasm) and the constant overall cortical flow into the division plane. These two are balanced by some events in the division plane or in the ring (this remains a big black box mainly due to missing direct observation of the cortical flow in the division plane). I appreciate that it was possible to make a mathematical model based on the positive feedback and that it could be fitted to two different conditions (control vs. rho kinase depletion). However, considering that the data don't indicate increase of the total amount of ring myosin, it is difficult for me to understand the necessity of the positive feedback.

2) Logic to disregard accumulation in the division plane, loss by disassembly and turnover with the cytoplasmic pool.

As far as I understand, the only reasoning for disregarding "accumulation in the division plane" and "loss due to disassembly" (Figure 3C) is the similarity between the time course of the ring-directed cortical flow and the time course of ring myosin and anillin per unit length. However, this argument assumes that there is no turnover of myosin with the cytoplasmic pool at the division plane and the ring and that the flow within the division plane is uniform and constant (no accumulation). The authors claim that the result of FRAP excludes the exchange of the ring myosin with the cytoplasmic pool. However, their argument "If ring myosin were turning over due to exchange with cytoplasmic myosin, we would expect the FRAP curve to approach the control curve and the difference between the FRAP and control curves to disappear" is true only if all the ring myosin is exchangeable with the cytoplasmic pool and the exchange occurs rapid enough. This assumption is too strong (a kind of straw man argument). The absence of complete recovery simply tells us that there is a non-exchangeable population in the ring myosin. Moreover, indeed, the difference is getting smaller. The basis for the constant cortical flow in the division plane is unclear.

3) Laser dissection.

The authors performed laser dissection experiments (Figure 2) to assess the influence of the cortical resistance on the rate of ring closure. However, it doesn't seem to be sufficient/complete to draw any firm conclusion. It is not clear whether they are comparing the parallel and perpendicular cuts on their effects on the ring constriction. Only the absence of the effect of the parallel cut was mentioned. The cortical flow is not spatially uniform (Figure 1), implying that the cortical tension is also non-uniform. Then, don't we need to consider the relative positioning (angle and distance) between the ring and the cut? Is the lesion big enough? Even after the cut the ring can still be connected to the polar cortex via the unaffected zone of cortex. What would happen if a whole polar cortex is completely separated from the other part of the cortex and the ring? The rationale for the purple line in the Figure 2C "Expected rate if […]" described in the Materials and methods "The cortical opening after ablation was approximately 35μm^2^; this translates into an additional reduction in ring radius by ~0.8μm, if the cortical surface tension dominates the ring closure rate." needs more detailed explanation.

(R script to demonstrate the difficulty in distinguishing exponential curve and hyperbolic curve)x <- (0:16)/20y <- 1/(1-x)model <- nls(y~a+b*exp(c*x), start=list(a=1, b=1, c=1))xx <- (0:80)/100z <- predict(model, newdata=list(x=xx))plot(x, y, ylim=c(0,5))lines(xx,z)coef <- summary(model)$coefficients[,1]fit = sprintf("y=%.4f+%.4f*exp(%.4f*x)", coef[1], coef[2], coef[3])legend("topleft", legend=c("y=1/(1-x)", fit), pch=1, lty=c(0,1), pt.cex=c(1,0))

*Reviewer #2:*

This manuscript describes the kinematic analysis and mathematical modeling of the behavior of contractile ring components during cytokinesis. The basic assertion is that cortical myosin flows into the furrow region during ingression and these flows lead to increased accumulation which drives more rapid ingression as furrowing proceeds. Within its technical limitations, this work is carefully performed. It makes its underlying assumptions clear, and is supported by a model that appears consistent with these assumptions. However, there are fundamental problems with these assumptions and significant technical limitations. Ultimately, I do not find the central point of the manuscript convincingly demonstrated.

Concerning the technical weakness: The majority of the data in the manuscript is based on measurements of cortical components on the flatter parts of the embryo. "Cortical flow could not be monitored in the division plane or at the cell poles, due to their high curvature." Thus the part of the cleavage furrow where contractile ring components accumulate to the greatest extent are not detected with the same spatial or temporal resolution. Thus there is an intrinsic limit to the authors ability to track flow of cortical components from the surface and account for its accumulation at the furrow tip. This could be addressed by analyzing cell division of blastomeres that furrow in the imaging plane.

The manuscript extensively discusses a concept they call cortical surface area. This is a mixed metaphor that appears to include aspects of the cortex and the plasma membrane. The authors state, "New cortical surface could be gained uniformly, immediately behind the contractile ring, or at the cell poles" and cite a number of publications that assess the behavior of the plasma membrane during cytokinesis. However, this concept is flawed. As cytokinesis proceeds, the membrane surface area must expand in order for the cell to retain integrity. While the plasma membrane is by definition continuous and uninterrupted, the cell cortex is different, it need not cover the entire membrane to a constant "depth". Furthermore, fluorescence imaging reveals large inhomogeneities in the cortex and, unlike membrane lipids, cortical components can associate and dissociate.

Another major, central weakness is that the manuscript primarily considers two models for accumulation of myosin in the furrow: retention and ring-directed flow. While these mechanisms likely contribute, they represent an incomplete view of the mechanisms by which components accumulate in the furrow. The authors appear to assume that contractile ring components are recruited at a specific time and then reorganized on the cortex. Indeed, the text states, "cytokinesis initiates when spindle-based signaling activates RhoA on the equatorial cortex leading to the abrupt recruitment of contractile ring components." Presumably, they also imagine that additional components are recruited at poles to replace the material that flows toward the furrow. However, this view is inconsistent with extensive analysis of the mechanism of Rho-dependent assembly of the contractile ring. RhoA is active throughout cytokinesis, as indicated by continual association of a RhoA biosensor, which is most concentrated at the furrow. Unfortunately, this key region largely falls outside of the part of the embryo the authors image at high spatial and temporal resolution. Given what is known about RhoA and its effectors, there is no reason to posit that during the course of furrow ingression, active RhoA does not continually activate its effectors and induce recruitment of contractile ring components throughout the progression of cytokinesis. Indeed, the observed increases in ring components may follow the increase in the concentration of active RhoA at the furrow.

The authors suggest that their data excludes recruitment of myosin from the cytoplasm, but the evidence is unconvincing (Figure 4C, subsection “Component levels and fluorescence recovery after photobleaching of the division plane support constriction rate acceleration due to ring-directed flow versus component retention”, second paragraph). For example, the FRAP data is consistent with the furrow tip containing a pool of myosin that has lower mobility. The "recovery" after bleaching could reflect de novo recruitment on top of the slowly exchanging bleached myosin. In addition, there is little evidence of flow of unbleached myosin into the ring, which would be predicted from the author's model.

The authors find a correlation between the rate accumulation of myosin in the ring, and the rate of constriction. There is no evidence that this correlation reflects a causative relationship.

Consistent with the previous point, evidence in the literature contradicts the authors explanation that cortical flows of myosin from regions flanking the furrow are required for the proposed exponential increase in contractile ring myosin that speeds up the rate of ingression. Specifically, *C. elegans* embryos deficient in NOP-1 are significantly depleted of cortical accumulation of contractile ring components outside the of the equatorial/furrow region. However, these furrows ingress with near wild-type kinetics, indicating that efficient furrow ingression does not require these major flows of contractile ring components. Rather, it suggests the existence of an alternative mechanism that provides for a time-dependent increase in contractile ring components.

Conversely, embryos that are defective in centralspindlin-directed RhoA activation, do contain cortical myosin that appears to flow in the proposed manner, yet such embryos ingress partially and slowly, suggesting that flow-mediated concentration of contractile ring components is insufficient to generate the proposed behavior of the ring.

The authors state "The broad conservation of this property, which allows cytokinesis to complete in a temporally restricted cell cycle window, suggests that feedback between contractile ring myosin and ring-directed cortical flow will be a broadly conserved property of contractile rings in animal cells." This would imply that cells that lack ring-directed cortical flow will exhibit aberrant timing of ring closure. However as mentioned above, analysis of NOP-1 deficient embryos violates this conjecture.

The authors discuss the concept of astral relaxation: "This differential response of the polar cortex to ring-generated tension, which results in a flow of myosin and other cortical components towards the cell equator, is consistent with the idea of polar relaxation hypothesized in early conceptual models of cytokinesis." They fail to mention or cite that astral relaxation has been experimentally documented in the early *C. elegans* embryo. Indeed it has been demonstrated that a posterior directed spindle directs anterior-directed flow of cortical components that self organize into a furrow (PMID 17669650). Importantly, this anterior furrow is entirely dependent upon the protein NOP-1 (PMID 22918944).

*Reviewer #3:*

The manuscript from Khaliulin et al. investigated the contribution of cortical flow in maintaining constant rate of ring constriction during cytokinesis in worm embryos. Both de novo actin/myosin assembly at the division site and cortical flow of components to the cleavage furrow are involved in cytokinesis. However, it remains controversial about the relative importance of each pathway. But I think the paper still need some minor revisions to be acceptable for publication.

1. Due to no cell cortex in yeasts but the ring constricts at a constant rate (Pelham and Chang, Nature, 2002), the references of Wu and Pollard 2005 paper in the Introduction and Discussion are somehow misleading. In fission yeast, myosin-IIs mostly retain in the ring during its constriction, they are also highly dynamic by exchanging between the ring and cytoplasm. The similarity and difference between Wu and Pollard, Carvalho et al, and the current work should be discussed.

2. In Zhou, M., & Wang, Y. L. (2008), "Distinct pathways for the early recruitment of myosin II and actin to the cytokinetic furrow" (Mol Biol Cell, 19(1), 318-326), it is found that myosin-II is recruited to the furrow mainly by de novo assembly, but not cortical flow, during early cytokinesis in mammalian cells. This and other similar works on cortical flow and de novo assembly should be cited and discussed.

3. A figure supplement showing the cortical flow map at cell poles before and during ring constriction will be useful.

4. The terms "cortex surface" and "surface gain" are confusing. Because the plasma membrane was not directly tracked, it should be make clear what you mean here. Otherwise, casual readers may think the plasma membrane is inserted at cell pole, which is likely, but not tested here.

5. The Materials and methods section is not clear regarding how background fluorescence was subtracted. Which region was used as the background?

6. The reference citations are not consistent, sometimes multiple authors from the same paper are listed.

[Editors’ note: what now follows is the decision letter after the authors submitted for further consideration.]

Thank you for resubmitting your work entitled "A positive feedback-based mechanism for constriction rate acceleration during cytokinesis in *C. elegans*" for further consideration at *eLife*. Your revised article has been favorably evaluated by Anna Akhmanova as the Senior Editor, Mohan Balasubramanian as the Reviewing Editor, and three reviewers (including Michael Glotzer).

The manuscript has been improved but there are several remaining issues that need to be addressed before acceptance, as outlined below.

The referees have returned their comments and I have discussed the comments with the Senior Editor and we have compiled this decision letter with all substantive points raised by the three referees.

In particular, I have taken the step of providing the full list of comments so that the counter arguments to your model for mechanism of acceleration of cytokinetic ring contraction and the role of myosin II accumulation via cortical flow are fully captured.

In light of the fact that all three referees have concurred that the imaging data are among the best in the field of worm cytokinesis and that I believe you are proposing a striking and provocative new model for aspects of cytokinesis, we are interested in publishing your work, and invite you to submit a revision. In the discussions between referees, it was highlighted that the paper was worthy of publication, but that the models need to be considered more critically.

We would like you to rewrite the paper significantly, based on these comments, as well as perform a straight-forward experiment.

One of the referees has raised the unsatisfactory resolution of the current study vis-à-vis the previous work from your laboratory (Carvalho et al., 2009) (Points 10 and importantly 11). The Senior Editor and I concur with the points the referee has raised. We believe a clear statement of your current position is consolidating the Carvalho paper and the current study will be very valuable for the field and will prevent any confusion in the field.

Also, I believe one experiment, of treating 1 and 4 cell embryos with LatA, as a test of your model is required (point 11).

Point 9 below also requires special attention. The exponential vs. hyperbolic accumulation of components in the ring cannot be easily distinguished and the referee has detailed their concern. Please clarify the limitations of your analysis.

Please also pay attention to the use of terminology, which have been raised by all referees (cortical surface area, cortical area, naive cortex), raised by the referees throughout the comments.

Below, I mention what needs to be done for each point.

Although I think I can decide on any submitted revision, I might call upon one of the referees, if required.

Please find below the consolidated comments of the three referees and my recommendations.

1) The revisions to the manuscript have clarified their model so that readers can better understand what the authors claim to demonstrate. I remain unconvinced of the authors model. My concerns are due to the fact that the manuscript is largely based on an *inference* of cortical compression.

I want to first re-state that the data shown are carefully obtained. The measurements of cortical flow are of interest. The authors provide evidence that the rate of furrow formation is limited by internal viscosity in the contractile ring, which is a novel insight and an important point.

Overall, I remain unconvinced by the author's interpretation of their results. There is some value in their quantitative model, though (1) it is not well constrained, there is no reason why the flow has to be the source of positive feedback and (2) it has not been extensively tested experimentally. However, if the authors want to publish their interpretation, I have no strong objection, particularly in *eLife* where readers can readily access the reviews that indicate that experts in the field do not subscribe to their interpretation of their results. Notably, the authors and I communicated following the first version. This communication has lead to a more clear description of their model. Yet during that process I communicated the gist of the comments below, yet they remain unresolved in this version. Specifically, in the previous version, the manuscript gave the impression that cortical flows delivered myosin into the furrow region, thereby accelerating ingression, flows, and myosin accumulation through a form of positive feedback. Now, the authors have clarified their interpretation as follows:

a) The amount of cortical surface area that flows into the furrow region exceeds the surface area of the division plane.b) As a consequence of (1) and an unstated assumption that cortical surface area is not disassembled, the authors infer that cortical surface area is compressed. There is no direct evidence for cortical compression except at the very earliest stages of furrow formation. Interestingly the rate of flow at the stage where compression is observed is 2-3x lower than that during furrow ingression (Figure 1B vs. 3A).c) The total cortical surface area is inferred to increase exponentially and it parallels the increase in ring components.d) Yet, the myosin that flows into the furrow region is not a major contributor to ring myosin, rather it largely disassembles. Indeed, in en face views of the division plane, there is no detectable flow of myosin from the "exposed" cortex to the ring.e) However the flow of cortex is proposed to provide additional, initially "naive", cortical surface area that is then patterned by RhoA (Figure 7), and it is this exponentially increasing cortical surface area that leads to exponentially increasing levels of ring myosin. This begs the question, what is "naive" cortex?

(Editors’ recommendation: rewrite significantly for clarity)

2) At the core of the issue with this model is inference of cortical surface area compression. First, the authors claim that cortical compression can be readily inferred from the difference between the amount of cortical flow into the division plane and the area of the division plane. Yet, the actomyosin cortex is dynamic, in addition to compressing and expanding, it can assemble and disassemble. Indeed, the manuscript shows that at the poles, cortical surface area is created as the cortex flows into the furrow region. And, myosin – a key component of the cortex – is largely assumed to be lost as the furrow flows into the division plane (see point 3). Thus, while cortical compression is possible, cortical disassembly is another possibility, which is not given sufficient consideration. In fact it is a strong possibility given that there is loss of a key component of the cortex, myosin.

(Editors’ recommendation: rewrite significantly for clarity)

3) Furthermore, the authors have not explained why "naive cortex" would be required for the zone of active RhoA to drive an increase in myosin accumulation?

(Editors’ recommendation: rewrite significantly for clarity)

4) Given that ARP-2/3 nucleated actin is likely a nucleator of some of the actin in the cell cortex, it is notable that its depletion does not dramatically affect the rate or extent of furrow ingression in otherwise WT embryos, as has been shown previously (PMID 22226748). This raises the follow-up question: what is "naive" cortex in ARP-2/3 depleted embryos?

(Editors’ recommendation: rewrite significantly for clarity)

5) In the author's rebuttal letter (reviewer 2, fourth response) the authors state, "The reviewer would propose that there could be another source of positive feedback that controls myosin accumulation (for example some type of ring intrinsic feedback loop involving Rho-based signaling), and that exponential accumulation of myosin arising from this as yet un-described feedback loop could, in turn, control the constriction rate and the rate of cortical compression. We do not disagree that this could be the case." Positive feedback in RhoA signaling during cytokinetic processes has been demonstrated, (PMID 26479320), and there is evidence for a mechanism in which RhoA might generate positive feedback through the recruitment of centralspindlin and its activation of the RhoGEF ECT-2 (PMID 26252513).

(Editors’ recommendation: rewrite significantly for clarity)

6) Given the topic of this manuscript, it is surprising that the authors do not mention that local RhoA activation is sufficient to induce furrow formation (PMID 27298323) and all of the literature concerning the mechanism of RhoA activation during cytokinesis.

(Editors’ recommendation: consider discussing this paper)

7) The authors state, "We propose that, due to polar relaxation, the compressing cortex pulls naive cortex not patterned by the initial round of RhoA signaling, into the Rho zone." Here the authors are generating confusion between terms that have a different historical meaning. Polar relaxation was used to describe a mechanism by which astral microtubules might induce a net increase in equatorial contractility by the local inhibition of contractility at the poles (polar relaxation). Here, they are discussing how existing equatorial contractility induces flow of cortex away from the poles. These terms are already sufficiently confused in the literature, it would be better to avoid adding to it.

(Editors’ recommendation: rewrite significantly for clarity)

8) One paper published during their revision is quite relevant: PMID: 29146911. DOI: 10.1038/s41467-017-01231-x. I suggest that the authors cite and briefly discuss the paper in their final manuscript.

(Editors’ recommendation: consider discussing this paper)

9) Exponential/hyperbolic accumulation.

In Figure 5, the authors compare exponential curves and hyperbolas for fitting with the experimental data and conclude that exponential curves fit better. However, it is unclear whether their comparison is fair. For fitting with the data of mean fluorescence per unit length with an exponential curve (Compression feedback), three parameters, i.e., the amplitude, the time constant and baseline can be adjusted. On the other hand, for fitting with a hyperbola (Retention model), it is unclear what the formula for R(t) looks like and what degree of freedom was allowed.

As I pointed out in the previous reviewer comments, clear distinction between the exponential curve and hyperbola is not trivial. The authors' own data and interpretation demonstrate this difficulty. First, in Figure 3C, "Cortical compression (rate per unit ring length)" is fitted with an exponential function. However, this quantity dAcompdt1R should follow a hyperbolic increase in time since the first term, dAcompdt, is largely constant (Figure 3B 'Normalized Surface Area Flux') and the second term, 1/R, is an inverse of a linearly decreasing function of time (Figure 1A). Second, in the same panel, "Ring shrinkage rate per unit length (-dRdt1R)" is also fitted with an exponential curve. However, this quantity should also be hyperbolic for the same reason (-dRdtis constant most of the time during furrow ingression Figure 1A). These examples nicely demonstrate the difficulty in distinguishing between exponential and hyperbolic changes by curve fitting with a set of data that are not really suitable. The authors' approach doesn't have sufficient power to discriminate between possible theories.

(Editors’ recommendation: rewrite significantly for clarity as well as discuss the limitation of the curve fitting approaches you have taken)

10) Exchange of ring myosin with cytoplasmic pool.

I am really confused with what the authors are actually thinking about the exchange of myosin with the cytoplasmic and cortical pools. Based on the whole division plane FRAP experiment in Figure 6, they exclude the exchange of the ring myosin with cytoplasmic myosin. On the other hand, in the schematic in Figure 4, the recruitment of cytoplasmic myosin to the Rho zone is depicted as the major source of the increase of the cortical myosin in the Rho zone. Although it was not explicitly stated in mathematical modeling, myosin on the cortical flow within the Rho zone and myosin accumulated at the contractile ring behave differently as to new recruitment and removal by disassembly. In the FRAP in Figure 6, both of these myosins, as well as myosin on the 'naive cortex' in the division plane, were photobleached. Recovery seems to have started at the contractile ring instead of the flowing cortex outside of the ring. The simplest explanation would be that there is an exchange of myosin at the ring.

A constant level of per-unit-length bleached myosin is a basis for their compression feedback model. However, bleached myosin calculated by the two formulas made by exponential fitting is not constant. Contrary to their description (subsection “Fluorescence recovery after photobleaching of the division plane is consistent with the Compression Feedback model”, last paragraph), the two curves are getting closer (see graphs that can be generated by running an R script at the bottom). This point was clearer in Figure 4C of the original submission. If we apply the same logic as later in the aforementioned paragraph, the data indicate that the recovery is at least partially due to the exchange with cytoplasmic myosin. I don't understand why they could assert "We also note that, consistent with our prior observations at the 4-cell stage (Carvalho et al., 2009) we did not observe evidence of turnover of ring myosin due to exchange with myosin in the cytoplasm."

(Editors’ recommendation: rewrite significantly for clarity and explain limitations)

11) Consistency with Carvalho 2009.

The authors' argument in Figure 6B is valid to exclude the retention model without any exchange at the ring. However, the same logic also strongly argues against the model proposed by Carvalho (2009), which excludes both the exchange of the ring myosin with cytoplasmic myosin and with the nearby cortex. The tornado-shaped non-recovery zones in the kymographs were explained by the closure of the ring and slow exchange within the ring in the absence of the exchange with the cytoplasm nor delivery by flow from the flanking cortexes. However, if the current model is correct, the flow from the flanking cortex should cause a gradual recovery in the tornado-shaped zones in the kymographs. In other words, the current model is not consistent with the data by Carvalho (2009).

In Figure 3—figure supplement 4, the authors quantified the per-unit-length amount of myosin in 4-cell stage division. This should be essentially equivalent to Figure 4D in Carvalho (2009), from which they had concluded that the per-unit-length amount of myosin in the ring is constant (note: this is reproduced as Author Response Image 2 in their reviewer response, hiding the latter half of the time scale where the drop was observed before the sudden 1.3-fold increase at 10 µm perimeter. In addition "In contrast, we observed a ~1.3 fold increase for all three components." is also misleading as they mentioned the 1.3-fold increase only in the last few µm, sticking to the constant per-unit-length level. Additionally, the point of transition is 18 µm in Figure 4D of Carvalho (2009) but 25 µm in their rebuttal. It is not clear why at a glance they look different (or the author could conclude differently). Are they based on the same set of image data? Or, was the recording newly performed? How do they look if they are plotted with the same x-axis (the perimeter of the ring or fraction of ring closure)?

Simply speaking, the major conclusions in Carvalho (2009) are inconsistent with those in current manuscript. There are 4 possibilities:a) Divisions in 1-cell stage and in 4-cell stage are different.b) There is no such difference. The data or interpretation in Carvalho (2009) was wrong. The current model is correct.c) There is no such difference. The current model is wrong. The old model was correct.d) There is no such difference. Both the old and new models are wrong

The authors should clarify which is the case. If b) is the case, detailed point-by-point explanations will be essential as to which data/interpretation in the previous paper still stand or don't stand anymore.

To distinguish between these possibilities, repeating latrunculin A treatment during furrow constriction at 1-cell and 4-cell stages using eggshell permeabilization by perm-1(RNAi) would be highly informative. In Carvalho (2009), insensitivity of the 4-cell division to latrunculin A added during furrow ingression provided a support for disregarding the importance of turnover.

(Editors’’ recommendation: Perform the LatA treatment experiment as well as clearly state what exactly your position is in terms of this manuscript vs. the Carvalho manuscript)

12) Inconsistency between the scheme in Figure 4 and their own observation (Naive cortex?).

The model doesn't match with what was observed by the authors. It is ambiguous what the 'naive cortex' is. In the scheme in Figure 4A left box, it is placed between the equatorial Rho zone and the polar cortexes and treated as empty boxes without myosin. The only route of recruitment of myosin to the cortex is the direct recruitment of myosin II from the cytoplasm in the Rho zone. However, this picture is not consistent with their own observations.

In Figures 1 and 3, they observed the flow of myosin II in the cortical regions at the surface of the embryos, which largely correspond to the regions marked in grey in Figure 4 (and the green Rho zone before furrow ingression starts). Myosin DOES exist in these regions of "naive cortex". The authors may argue that myosin in these regions is inactive. However, at the beginning of furrow ingression (Figure 3, t/tt_ck_ -0.1 and Figure 1t/t_ck_ 0.2 'bottom'), the gradients are observed in regions wider than the ~10 µm wide 'Rho zone'. In later stages, the authors attribute the velocity gradients in the regions flanking the furrow to a projection artifact (dotted segments in Figure1) although it is not very clear how convincing this interpretation is. In the regions where the flow is represented by a solid line, the velocity is largely constant. However, an absence of compression doesn't necessarily mean that myosin is inactive. The tension generated by myosin might just be balanced. Indeed, laser ablation caused outward flows in all the directions (not limited to the direction towards the furrow), indicating that the 'naive cortex' is under active tension although it would be lower than that in the Rho zone.

(Editors’ recommendation: rewrite significantly for clarity)

13) Inconsistency between the mathematical model and the scheme in Figure 4

In the schematic in Figure 4, recruitment of myosin from the cytoplasm is depicted as the major route of the increase in the total amount of myosin in the Rho zone. However, in the mathematical model, a key assumption is that the mass of cortical myosin is conserved while it flows (formula 20). There is no term that corresponds to the recruitment from the cytoplasm.

(Editors’ recommendation: rewrite significantly for clarity)

14) Lack of the effect of geometry change by furrow constriction, or the mechanism for coupling removal of myosin with furrow constriction.

Simply due to the geometry change, even without new recruitment of myosin either via cortical flow or from the cytoplasm, furrow constriction results in an increase of the per-length-amount of myosin if there is no removal of myosin coupled with the disassembly of the contractile units. However, this fundamental fact hasn't been properly incorporated in their mathematical model.

The consequence of this process is mentioned and depicted in the right box of the Figure 4A "Ring shortening is coupled to disassembly and does not change the per unit length amount of ring myosin". In Figure 4B, in box 3, under the lead "The per unit length rate of ring disassembly is proportional to the per unit length amount of ring myosin", the formula (24) is shown. However, this formula is about the relationship between the per unit length amount of ring myosin and the per unit length rate of ring constriction (instead of the per unit length rate of ring disassembly). The caption can be true only when the ring disassembly is proportional to the ring constriction. In their model, this is achieved by neglecting the effect of geometry change due to furrow constriction while they are discussing formula 18 to 23, and later on by using M_ring_ as equivalent to the per unit length amount of ring myosin without properly explaining that their model doesn't include a mechanism for coupling constriction and disassembly, which is not trivial and was a key discovery in Carvalho (2009).

(Editors’ recommendation: rewrite significantly for clarity)

15) The velocity of flow of naive cortex into the Rho zone.

The authors' theory predicts a linear relationship between v_flow_, the velocity of flow of naive cortex into the Rho zone, and M_ring_, the total per-unit-length amount of ring myosin (formula 22). Although, mathematically, v_flow_(t) is the speed of the flow at the boundary of the Rho zone, considering the continuity of the flow at the boundary of the Rho zone and largely uniform flow, it is reasonable to interpret v_flow_(t) as the velocity/speed of flow of naive cortex. In Figure 4, this is indicated by growing arrows labeled "Cortical flow" on the 'naive cortex'. However, the data in Figure 1 and 3 don't show such behavior. Instead, the flow seems to show rapid increase around t/t_ck_~0 and gradually slows down between t/t_ck_>0.2. It will be informative if the top and bottom speeds of cortical flow are plotted against time. Anyway, this pattern is inconsistent with the theoretical prediction. In this case, their favourite trick to convert a constant quantity into an exponentially increasing one by dividing by the ring radius wouldn't work well since the radius that can be used here is the radius at the boundary of the Rho zone, which only decreases towards 5 µm, instead of the ring radius, which decreases towards 0 µm.

(Editors’ recommendation: rewrite significantly for clarity)

16) 'Rho zone'.

It is not clear what exactly the 'Rho zone' is especially after the furrow has deepened (the distance from the embryo surface to the contractile ring is larger than 5 µm). In the mathematical model, they assume that w is a constant. However, this is not realistic. The actual width of the distribution of active Rho in the cell is likely to be broader in the beginning and become narrower. The word "ring" stands for the contractile ring in most places while in some places it refers to a broader zone used for quantifying myosin (e.g. Figure 6A, the zone between the two boundaries marked with dotted lines). In the latter case, the 'ring' largely overlaps with the 'Rho zone' but not in the former case.

(Editors’ recommendation: rewrite significantly for clarity)

17) Feedback?

The authors argue that exponential increase suggests a positive feedback. However, this is not necessarily true (even if their interpretation of exponential increase were true). For example, under an optimal condition, bacteria grow exponentially. Usually, this is not explained by a positive feedback loop. To confirm a feedback loop, an experiment to perturb a key step in the loop should be performed.

(Editors’ recommendation: rewrite significantly for clarity or perform an experiment)

18) "Cortical surface".

I support reviewer #2's original point about the terms "cortical surface" or "cortical surface area". The authors' rebuttal is not convincing. In 50's, the current concepts of "plasma membrane/cell membrane" and "cell cortex" were not established yet. The fluid mosaic model was established in 70's. The "membrane" in Swann and Mitchision (1958) refers to a combination of the lipid bilayer and the underlining cytoskeletal network.

The markers listed are all added from the medium and attached on the cell surface (except for Dan's pigmented granules). The relations between these surface markers and the cortical cytoskeletal network haven't been clarified and can be variable. The expansion (an increase of the distance between the markers) can be caused between the markers that are not anchored to the cortical cytoskeleton by insertion of new membrane lipid bilayer. If markers are somehow anchored to the cortical cytoskeleton, the expansion can also be caused by relaxation of the cytoskeletal network (or radial pull by the neighboring cytoskeletons).

The terms "cortical surface" and "cortical surface area" are confusing. The latter has been widely used to describe the geometry of brains. The usage of it in the context of cytokinesis seems to be a recent invention by the authors. At least, this word doesn't appear in Dan (1954) nor in Swann and Mitchision (1958) although they used "cortical layer", "cortical gel" etc.

In most of the cases in this manuscript, just "cell cortex" instead of "cortical surface" seems to be appropriate.

# an R script to plot unbleached myosin based on the fitted formulas in Figure 6Ct <- (0:100)/100y1 <- 0.22*exp(2.8*t)+0.78y2 <- 0.24*exp(2.8*t)-0.07y3 <- y1-y2quartz(width=4, height=4.5)plot(t, y3)plot(0,0, xlim=c(0,1), ylim=c(0,4.5), type="n", xlab="time", ylab="signal per length", xaxs="i", yaxs="i")lines(t, y1, col='magenta')lines(t, y2, col='green')lines(t, y3)

(Editors’ recommendation: rewrite significantly for clarity)

---

## [Author Response]

[Editors’ note: the author responses to the first round of peer review follow.]

Our decision has been reached after consultation between the reviewers. Based on these discussions and the individual reviews below, we regret to inform you that your work will not be considered further for publication in eLife at present.The reviewers appreciated the novel imaging and computational approaches you are using to understand mechanisms of cytokinetic force generation. The reviewers also found the premise that cortical flow of myosin together with existing myosin in the furrow regulates the rate of furrow contraction interesting. However, a number of issues were raised. These include 1) inconsistencies with what is known in the literature (some experiments with Rho pathway and Nop1 mutants can address these), 2) issues with the theoretical framework and terminologies used (raised by reviewers 1 and 2, which may need experiments and rewriting), 3) aspects of the data fitting and conclusions regarding cortical flow (reviewer 1; may require reanalysis / reconciliation), and 4) referencing (reviewers 2 and 3; should be easily fixed).If you feel you can address the issues raised with experiments and rewriting / analysis, we will be happy to consider a revised version or as a new submission. In both cases, we will send the paper to the same reviewers.

Summary of major changes in the revised manuscript:

The comments from reviewers on our initial submission focused on the following major requests:

1) Re-write the text to clarify and provide more context for the nature of our cortical dynamics measurements and the conclusions that can be drawn from them.

2) Revise our model to be integrated with the current concept of RhoA-based signaling.

Examine cortical flows in NOP-1-depleted embryos to exclude that *nop-1* inhibition presents a counter example to the model we propose.

4) Expand the presentation to clarify why our data support the exponential accumulation of contractile ring components due to positive feedback versus a hyperbolic increase (1/R) due to component retention.

5) Improve the presentation of our division plane photobleaching experiment.

6) Include a discussion of how our model relates to the long-running debate in the field as to whether contractile ring components are delivered into the ring by cortical flow or recruited de novo downstream of RhoA-based signaling.

To achieve these 6 goals, as well as address other reviewer comments, we decided to remove the analysis of Rho kinase depleted embryos (originally presented in Figures 6 and 7) from the manuscript, and replace them with 4 new figures that explain our revised model and the experiments we performed to validate its predictions in greater detail (new Figures 4-7 added to the revision). The text has also been largely re-written to address all of the reviewer-highlighted issues.

As requested by reviewer #2, we also performed an analysis of cortical flow in *nop-1(RNAi)* embryos and found that neither the amount of myosin in the cortical puncta that we use as fiduciary marks for monitoring cortical dynamics nor the rates of cortical flow during constriction are detectably altered by *NOP-1* depletion. Thus, cortical dynamics in *NOP-1* depleted embryos are not in conflict with our model.

Note that in our response we will address the concerns of reviewer #2 first, because the discussion of the revisions we made to our model in response to his comments provides useful context for discussing how we addressed the concerns of reviewers #1 and #3.

Reviewer #2:[…] The manuscript extensively discusses a concept they call cortical surface area. This is a mixed metaphor that appears to include aspects of the cortex and the plasma membrane. The authors state, "New cortical surface could be gained uniformly, immediately behind the contractile ring, or at the cell poles" and cite a number of publications that assess the behavior of the plasma membrane during cytokinesis. However, this concept is flawed. As cytokinesis proceeds, the membrane surface area must expand in order for the cell to retain integrity. While the plasma membrane is by definition continuous and uninterrupted, the cell cortex is different, it need not cover the entire membrane to a constant "depth". Furthermore, fluorescence imaging reveals large inhomogeneities in the cortex and, unlike membrane lipids, cortical components can associate and dissociate.

The property we are measuring, cortical surface area, is clearly defined, and we are very careful not to conflate cortical expansion with membrane expansion. Cortical surface area is not a concept that we invent here, it is the physical property that is measured by our method. Our experiments are an updated quantitative version of the classic experiments performed by Katsuma Dan, who tracked the movement of kaolin particles adhered to the surface of sea urchin embryos during cleavage (Dan et al., 1937; Dan et al., 1938; Dan and Dan, 1940; Dan, 1954; Dan and Ono, 1954; summarized by Swann and Mitchison, 1958). It is important to note that work in multiple systems has shown that the entire cell surface, from cortex-associated granules in the cytoplasm to cell surface receptors, moves in a coordinated fashion during cytokinesis (Cao and Wang, 1990; Dan, 1954; Dan and Dan, 1940; Dan, Dan, and Yanagita, 1938; DeBiasio, LaRocca, Post, and Taylor, 1996; Fishkind, Silverman, and Wang, 1996; Hird and White, 1993; Reymann et al., 2016; Swann and Mitchison, 1958; Wang, Silverman, and Cao, 1994). In the *C. elegans* embryo, myosin II exhibits a RhoA-dependent enrichment in the contractile ring. However, it is also present in small puncta, distributed over the entire cortex, that we use as fiduciary marks akin to the kaolin particles in the Dan experiment. Consistent with prior work, we also show the movement of the myosin foci that we track mirrors that of cortical actin filaments in our system (Figure 1—figure supplement 1C). Our method allows us to monitor movement of the cortical surface as well as cortical surface compression and expansion (which are detected as gradients in cortical velocity).

Our measurements do not have anything to do with the plasma membrane or how its deposition is patterned. Membrane deposition could be patterned in a similar way to cortical expansion or it could exhibit a distinct spatial distribution. We do not monitor membrane deposition and make no claims about this in the manuscript. The depth of the cortex whose flow we are monitoring and whether or not the cortex is homogeneous also do not impact our conclusions. We completely agree with the idea that the cortex is likely to be heterogeneous – for example, less dense at the poles than at the equator – and this is an interesting topic, but this does not bear on our conclusions.

With respect to discussion of prior work when we present our analysis of where cortical surface expansion occurs during cytokinesis to determine whether it: (i) is limited to the poles, ii) occurs uniformly over the cortex, or iii) occurs in the division plane behind the ingressing ring (illustrated in Figure 1—figure supplement 3), we cite references where people have observed or assumed each pattern. In *Xenopus*, it is usually stated that new differentially pigmented “membrane” is deposited behind the ingressing ring. Membrane may indeed be deposited here as localized exocytosis has been observed in this location (Danilchik et al., 2003). However, in this system, analysis of the movement of iron oxide (Bluemink and de Laat, 1973) and carbon particles (Byers and Armstrong, 1986; Danilchik et al., 2003) adhered to cell surface receptors integrated into the cortex, and of radiolabeled (Byers and Armstrong, 1986) or lectin-labeled cell surface proteins (Danilchik et al., 2003), was also used to monitor cortical surface expansion/compression, analogous to the original experiments by Dan. The results of these experiments indicated that cortical surface expanded in the region behind the furrow in addition to any membrane that may be deposited there. That cortical surface expansion can occur at the cell poles was similarly demonstrated by Dan, who measured the distance between surface-adhered particles and the distribution of pigmented cortex-associated granules (Dan et al., 1937; Dan et al., 1938; Dan and Dan, 1940; Dan, 1954; Dan and Ono, 1954; Swann and Mitchison, 1958). We also chose uniform cell surface expansion as one of our test patterns because this is an assumption made in modeling papers (the examples we cite are Turlier et al., 2014 and Zumdieck et al., 2007).

Because of the confusion in the literature that the reviewer highlights, and to clarify the history, nature, and significance of the measurement we are making, we have rewritten the section of the manuscript describing the generation and analysis of our cortical flow map. The first section of which is included below:

“During the first division of the *C. elegans* embryo, the surface area of the cell increases by ~40% to accommodate the shape change that generates the daughter cells. […] How deposition of plasma membrane, the fluid lipid layer that overlies the cortex, is controlled and where it occurs are distinct questions that we will not discuss here.”

Concerning the technical weakness: The majority of the data in the manuscript is based on measurements of cortical components on the flatter parts of the embryo. "Cortical flow could not be monitored in the division plane or at the cell poles, due to their high curvature." Thus the part of the cleavage furrow where contractile ring components accumulate to the greatest extent are not detected with the same spatial or temporal resolution. Thus there is an intrinsic limit to the authors ability to track flow of cortical components from the surface and account for its accumulation at the furrow tip. This could be addressed by analyzing cell division of blastomeres that furrow in the imaging plane.

In addressing the reviewer’s concern, we realized that the misconception that we are proposing “long distance transport” of myosin from the embryo surface into the ring was arising from the schematics (in Figure3G and 5B of the original submission) that we were using to illustrate our model, which we have removed and replaced with a completely reworked figure in the revised manuscript (new Figure 4). We describe these revisions in detail in response to the reviewer’s next point below. We note that, despite the turnover of its components, the cortical surface as a whole is a persistently interconnected entity. Thus, monitoring the movement of the cortex on the embryo surface allows us to accurately measure the area of the cortical surface flowing into the division plane, which we can compare to the area of the division plane to measure the amount of cortical surface compression.

The reviewer also asks whether we could monitor cortical flow in the division plane by imaging at the four-cell stage when this division provides an end-on view of the division plane. These divisions have a number of limitations (some of which are detailed in Figure 3—figure supplement 4 in the revision). However, the major limitation is that the division plane, while oriented end-on, is relatively deep within the embryo (~15 µm from the surface) and is not in a single focal plane. We visualized the division plane at the four-cell stage in maximum intensity projections of 16 z-planes collected at 1 µm intervals. The combination of the imaging depth and the number of z-planes required to capture the division plane make it impossible for us to monitor cortical flow with sufficient spatial and temporal resolution for detailed quantitative analysis.

Another major, central weakness is that the manuscript primarily considers two models for accumulation of myosin in the furrow: retention and ring-directed flow. While these mechanisms likely contribute, they represent an incomplete view of the mechanisms by which components accumulate in the furrow. The authors appear to assume that contractile ring components are recruited at a specific time and then reorganized on the cortex. Indeed, the text states, "cytokinesis initiates when spindle-based signaling activates RhoA on the equatorial cortex leading to the abrupt recruitment of contractile ring components." Presumably, they also imagine that additional components are recruited at poles to replace the material that flows toward the furrow. However, this view is inconsistent with extensive analysis of the mechanism of Rho-dependent assembly of the contractile ring. RhoA is active throughout cytokinesis, as indicated by continual association of a RhoA biosensor, which is most concentrated at the furrow. Unfortunately, this key region largely falls outside of the part of the embryo the authors image at high spatial and temporal resolution. Given what is known about RhoA and its effectors, there is no reason to posit that during the course of furrow ingression, active RhoA does not continually activate its effectors and induce recruitment of contractile ring components throughout the progression of cytokinesis. Indeed, the observed increases in ring components may follow the increase in the concentration of active RhoA at the furrow.

We are grateful for the useful feedback, which has helped us to significantly improve how we present our model. We would like to stress that, in contrast to the reviewer’s suggestion, we do not think that the cortex is patterned and then the initially recruited components are subsequently rearranged as the process proceeds, or that contractile ring components are recruited at the poles and then delivered to the ring. Our model is not that different than what is suggested by the Reviewer. In our revised model, we have made it clear that we think that RhoA-based signaling loads contractile ring proteins onto the cortex as it is pulled into the Rho zone by the compression of cortical surface within the ring. In the model in our original manuscript, this concept was embedded into the parameter m_cort_, which was defined as the concentration of myosin on the cortex at the point where it enters the Rho zone. Motivated by the reviewer’s comments we have renamed this constant m_rho_ and have clarified that this most likely corresponds the concentration of myosin that is loaded onto the cortex by RhoA-based signaling after the cortex is pulled into the Rho zone due to cortical compression within the ring. To clarify our vision for how the feedback loop that we propose between ring myosin and compression-driven cortical flow is integrated with RhoA-based signaling, we have completely reworked the presentation of our model (presented in anew Figure 4). The description of our revised model that appears in the text is:

“The natural coordinate system for contractile ring dynamics has two axes, an axis parallel to ring constriction (Figure 4A, around-the-ring axis) and an axis perpendicular to the ring (Figure 4A, perpendicular-to-the-ring axis). […] Thus, unlike the feedback loop operating along the perpendicular-to-the-ring axis, which would lead to an exponential increase in the per-unit-length levels of ring components, ring shortening would be coupled to disassembly and would not alter the per-unit-length amount of ring components.”

The authors suggest that their data excludes recruitment of myosin from the cytoplasm, but the evidence is unconvincing (Figure 4C, subsection “Component levels and fluorescence recovery after photobleaching of the division plane support constriction rate acceleration due to ring-directed flow versus component retention”, second paragraph). For example, the FRAP data is consistent with the furrow tip containing a pool of myosin that has lower mobility. The "recovery" after bleaching could reflect de novo recruitment on top of the slowly exchanging bleached myosin. In addition, there is little evidence of flow of unbleached myosin into the ring, which would be predicted from the author's model.

The model that we present in the revision for how unbleached myosin is delivered to the ring following bleaching of the division plane (new Figure 6B) is quite similar to what we would understand the reviewer’s proposed model to be. Both would presume, as the FRAP data suggests, that the bleached myosin in the ring is not significantly exchanging with myosin in the cytoplasm and is disassembled rather than retained in the ring as it constricts. Both our revised model and the reviewer’s model also propose that there is a mechanism that leads to the exponential de novo accumulation of unbleached myosin in the ring as indicated by our FRAP data. The fact that this accumulation is exponential suggests that it is controlled by a positive feedback loop. The difference between the model we propose and the reviewer’s model is in what this feedback loop might be. Our quantitative imaging data show that the per-unit-length amounts of myosin and the rate of cortical compression increase with the same exponential kinetics during ring closure. In the modeling portion of our paper, we therefore propose that the relevant feedback driving myosin accumulation could be between the amount of ring myosin and the rate of cortical surface compression within the ring. In our model myosin and other contractile ring components are loaded de novo on cortex that is pulled into the Rho zone by cortical compression. We develop an analytical mathematical formulation that describes this feedback, and show that it can recapitulate our experimental results (Figure 4). The reviewer would propose that there could be another source of positive feedback that controls myosin accumulation (for example some type of ring intrinsic feedback loop involving Rho-based signaling), and that exponential accumulation of myosin arising from this as yet undescribed feedback loop could, in turn, control the constriction rate and the rate of cortical compression. We do not disagree that this could be the case. However, since our experimental data indicate that cortical surface is compressed within the ring, such a model would also need to invoke a second as-yet-uncharacterized disassembly process to explain why compression of the cortex within the ring would not increase the concentration of ring components. In our view, it makes sense to propose the simpler model of feedback between ring myosin and compression driven flow because all of the necessary elements have been shown to occur, and the model can explain available data without needing to assume processes that have not yet been identified.

In our view, the reviewer’s criticism stems in part from the philosophical divide between the expectations of experimental science and modeling. In our view, models should be constructed based on reasonable assumptions and be able to explain available experimental data. However, to propose a model, it is not necessary to experimentally rule out all other possible models that *might* emerge in response to new data in the future.

However, to address the reviewer’s criticism since we hope that this paper will be read by both an experimental and modeling audience, we have re-written the manuscript to make the line between what we have shown experimentally and the model that we are proposing more clear. For example, see the transition in the paragraph below that leads into our modeling section in the revision.

“From our experimental work we conclude that: (1) the ring compresses cortical surface along the axis perpendicular to ring constriction throughout cytokinesis at a rate proportional to the amount of ring myosin and, (2) the amount of ring myosin and anillin increase at a rate proportional to the rate at which cortical surface is compressed into the ring, (3) the per-unit length amounts of ring myosin and anillin and the per-unit-length rates of cortical compression and ring constriction increase with the same exponential kinetics as the ring closes. […]To explore this idea, we developed an analytical mathematical formulation, which we call the Compression Feedback model, consisting of three equations with three model parameters, that describes this feedback and can recapitulate our experimental results (Figure 4A, B).”

We have also added a new paragraph to our Discussion that highlights this boundary, and the fact that the model proposed by the reviewer has not been ruled out.

“The experimental basis for our model is our analysis of cortical dynamics, which indicates that the compression of cortical surface within the ring along the axis between the relaxing poles that initiates during contractile ring assembly (Figure 3; (Reymann et al., 2016)), persists throughout constriction, resulting in a continuous flow of cortical surface into the ring. […] However, since our data indicate that cortical surface is compressed within the ring, such a model would need to invoke an as-yet-uncharacterized process to explain why compression of the cortex within the ring would not increase the concentration of ring components.”

The authors find a correlation between the rate accumulation of myosin in the ring, and the rate of constriction. There is no evidence that this correlation reflects a causative relationship.

Here we show that the per-unit-length rate of ring constriction is proportional to the amount of ring myosin, which, as the reviewer points out, is a correlation. In our mathematical model, we propose that myosin levels control constriction rate. We believe this is a reasonable proposal because it is consistent with prior experimental data from our lab, which showed that inhibition of Rho kinase, which reduces the concentration of cortical myosin, reduces the constriction rate (Piekny and Mains, 2002, Maddox, 2007), from the Canman lab analyzing the immediate effect on ring constriction of temperature upshift during cytokinesis of embryos expressing a temperature-sensitive myosin heavy chain mutant predicted to disrupt myosin dimerization and head-head coupling (Davies et al., 2014; PMID: 25073157), and from the Maddox lab showing that partial depletion of myosin reduces the constriction rate in the one cell stage *C. elegans* embryo (Descovich et al., 2017, PMID: 29282285).

Consistent with the previous point, evidence in the literature contradicts the authors explanation that cortical flows of myosin from regions flanking the furrow are required for the proposed exponential increase in contractile ring myosin that speeds up the rate of ingression. Specifically, C. elegans embryos deficient in NOP-1 are significantly depleted of cortical accumulation of contractile ring components outside the of the equatorial/furrow region. However, these furrows ingress with near wild-type kinetics, indicating that efficient furrow ingression does not require these major flows of contractile ring components. Rather, it suggests the existence of an alternative mechanism that provides for a time-dependent increase in contractile ring components.The authors state "The broad conservation of this property, which allows cytokinesis to complete in a temporally restricted cell cycle window, suggests that feedback between contractile ring myosin and ring-directed cortical flow will be a broadly conserved property of contractile rings in animal cells." would imply that cells that lack ring-directed cortical flow will exhibit aberrant timing of ring closure. However as mentioned above, analysis of NOP-1 deficient embryos violates this conjecture.

Cortical flow has not been previously measured during cytokinesis in NOP-1-depleted embryos. Thus, to address the reviewer’s comment, we imaged *nop-1(RNAi)* embryos expressing myosin::GFP and monitored cortical flow (n=16 embryos; Author response image 1). Prior work has shown that NOP-1-depleted embryos fail pseudocleavage, an incomplete cytokinesis-like event that occurs prior to mitosis during the establishment of embryonic polarity. NOP-1 depleted embryos are viable because pseudocleavage is not an essential event. To confirm that NOP-1 was depleted under our RNAi conditions, we filmed embryos by DIC microscopy prior to mitosis to ensure that pseudocleavage failed in the embryos in which we imaged cortical flow during cytokinesis (Author response image 1). The myosin foci that we use as fiduciary marks to monitor cortical dynamics appeared normal in NOP-1-depleted embryos (Author response image 1). Plotting cortical velocity in the AP direction versus distance from the ring at different times during cytokinesis revealed that cortical flows during cytokinesis were also not detectably altered by NOP-1 depletion (Author response image 1). We conclude that NOP-1-depleted embryos exhibit cortical flows similar to controls during cytokinesis, and thus NOP-1 depletion does not invalidate our model.

**Author response image 1. respfig1:** Cortical flows during cytokinesis are not altered in NOP-1-depleted embryos. (**A**) *nop-1(RNAi)* embryos were monitored prior to the first mitotic division to confirm that pseudocleavage failed indicating successful NOP-1 depletion. (**B**) Fluorescence confocal images showing that the myosin::GFP foci that we use as fiduciary marks to monitor cortical flow were not altered by NOP-1 depletion. (**C**) Graphs plotting cortical velocity in the AP direction versus distance from the ring at different times during cytokinesis. Two examples of control (black) and *nop-1(RNAi)* (red) embryos are shown.

Conversely, embryos that are defective in centralspindlin-directed RhoA activation, do contain cortical myosin that appears to flow in the proposed manner, yet such embryos ingress partially and slowly, suggesting that flow-mediated concentration of contractile ring components is insufficient to generate the proposed behavior of the ring.

Again, there is no data on cortical flows in centralspindlin-inhibited embryos during cytokinesis. Based on work to date on centralspindlin, we anticipate that the loading of components onto the cortex that is pulled into the Rho zone by cortical compression (Figure 4A) would be compromised, short-circuiting the feedback loop driving the accumulation of contractile ring components.

The authors discuss the concept of astral relaxation: "This differential response of the polar cortex to ring-generated tension, which results in a flow of myosin and other cortical components towards the cell equator, is consistent with the idea of polar relaxation hypothesized in early conceptual models of cytokinesis." They fail to mention or cite that astral relaxation has been experimentally documented in the early C. elegans embryo. Indeed it has been demonstrated that a posterior directed spindle directs anterior-directed flow of cortical components that self organize into a furrow (PMID 17669650). Importantly, this anterior furrow is entirely dependent upon the protein NOP-1 (PMID 22918944).

We now cite the indicated reference.

Reviewer #1:Cleavage furrow ingression during animal cytokinesis is driven by constriction of the actomyosin contractile ring. While major components of the contractile ring and key regulators for its assembly and constriction have been identified, our mechanistic understanding on how it generates force is limited. In their previous publication, based on the observation that the rate of the ring contraction is largely constant and proportional to the initial size of the ring, the Oegema group proposed a 'contractile unit' model, which assumes a presence of a contractile unit with a fixed initial length, which retains myosin while it shortens.

The reviewer is incorrect. In our 2009 paper, we argued *against* myosin retention. We had measured the per-unit-length amount of three contractile ring components, including myosin::GFP and GFP::anillin (Author response image 2, reproduced with permission from Figure 4D in Carvalho et al., 2009, PMID:19490897), as the ring decreased from 50 µm to 25 µm in perimeter at the four-cell stage. We also showed that when rings of all sizes reach a perimeter of 25 µm (radius ~3.5 µm), they undergo a transition due to contact with the midzone that alters the rate of ring constriction and component accumulation. The retention model predicts that component levels will double as contractile ring perimeter decreases 2-fold. In contrast, we observed an ~1.3 fold increase for all three components. This, combined with FRAP analysis we performed, suggested that components in the ring are not exchanging with components in the cytoplasm, which led us to propose that ring components are lost due to disassembly in proportion to the reduction in ring length during constriction. The “contractile unit model” that the reviewer refers to was a speculation that we made to explain how the per-unit-length constriction rate could increase during constriction despite the apparent lack of ring component retention. We hypothesized that if actin filaments shortened by depolymerizing from their ends during constriction, retention of the overall number of filament ends in the ring might explain how a high constriction rate is maintained. To make our prior model clearer, we have now included the following sentence in the Introduction.

“Alternatively, it has been proposed that the number of actin filaments could be retained. If actin filaments shorten from their ends during constriction, the overall amount of actin polymer could decrease in proportion to the reduction in perimeter while the number of filament ends remains constant, perhaps leading to observed increase in the per-unit-length constriction rate (Carvalho et al., 2009).”

**Author Response Image 2. respfig2:** Amounts of myosin::GFP and GFP:anillin per unit ring length measured at the 4-cell stage.

At the four-cell stage, the range of ring sizes for which it is possible to monitor component levels is limited (see Figure 3—figure supplement 4 of the revision) and the fact that cells are in contact on 2 sides with neighboring cells further complicates analysis, making it difficult to study the basis for constriction rate acceleration. These difficulties motivated us to develop methodologies for computational reconstruction of cortical flow and correction for attenuation with depth in 1-cell stage embryos. Our analysis and modeling lead us to propose that a feedback loop between ring myosin and compression-driven cortical flow along the axis perpendicular to the ring leads to an exponential increase in the per-unit-length amount of myosin and other contractile ring components that balances the decrease in ring size to allow the ring to close at a relatively constant overall rate during a significant portion of constriction. We missed this exponential accumulation in the prior work because it is difficult to appreciate at the four-cell stage due to the limited fraction of ingression over which component levels can be analyzed. In the revision, the relationship between the results we present here and our prior work at the four-cell stage are explained in Figure 3—figure supplement 4.

In this manuscript, the Oegema group studied further details of the mechanism of the ring constriction by precisely measuring the flow and total amount of cortical myosin as well as the myosin in the ring during the ring constriction. Based on these analyses as well as the results of laser micro-surgery and genetic manipulations, they propose a mathematical model with three parameters, which was demonstrated to be useful to explain the effects of a perturbation (depletion of Rho-kinase). The quality of image analysis, especially the 4D mapping of the cortical flow, is extremely high with striking number of video data analyzed. However, there are major problems to be addressed before publication.1) Logic to choose the feedback model instead of the retention model.The authors' logic for throwing away the retention model is not clear. The similarity between the time courses of the ring-directed cortical flow and the mean ring myosin or anillin per unit length, and the behavior of the ring myosin after FRAP are the observations on which they were based. Good fitting to exponential curves with a common time constant is interesting and is consistent with the cortical flow feedback model. However, the data are not strong enough to tell exponential from hyperbolic curves. The retention model predicts that the ring myosin per unit ring length is proportional to the inverse of the ring radius (∝1/R) (Figure 4C). Under the constant rate of ring constriction, this means that the ring myosin per unit ring length is proportional to 1/(R_unit_ – v∙t) (t: time, v: rate of ring constriction measured by radius), a hyperbola. With the value range used for Figure 3 and 4 (3 to 4-fold increase), it is almost impossible to distinguish an exponential curve from a hyperbolic one. Indeed, as a simplest example, a set of (x, y) calculated by y=1/(1-x) can be fitted with an exponential curve very well (as we can see by an R script below). The authors should explain why the data support exponential increase better than hyperbolic one.What data tell us are the largely fixed total amount of ring myosin (with limited exchange with flanking cortex or cytoplasm) and the constant overall cortical flow into the division plane. These two are balanced by some events in the division plane or in the ring (this remains a big black box mainly due to missing direct observation of the cortical flow in the division plane). I appreciate that it was possible to make a mathematical model based on the positive feedback and that it could be fitted to two different conditions (control vs. rho kinase depletion). However, considering that the data don't indicate increase of the total amount of ring myosin, it is difficult for me to understand the necessity of the positive feedback.

We have taken the reviewer’s concern to heart, and have addressed this point by restructuring the manuscript to include two new figures (new Figures 5 and 6) and a new accompanying text section that explains why our data support an exponential increase due to positive feedback rather than the hyperbolic increase (1/R) predicted by component Retention. Please note that the new figures also incorporate comments from the other reviewers, particularly reviewer 2 who indicated the need to integrate our observations with the established concept of the Rho zone during cytokinesis.

We agree with the reviewer that our measurements of per-unit-length myosin (and anillin) fluorescence do not distinguish between an exponential increase due to positive feedback between ring myosin and compression-driven cortical flow (which we refer to as the Compression Feedback model) and a hyperbolic increase (1/R) as would be predicted by component Retention (Retention model). We had indicated this in the original submission in Figure 4B (which showed total myosin levels with the fits for the Retention and Compression Feedback models), and used that to motivate the division plane photobleaching experiment that does rule out the Retention model. However, our presentation was not clear and we have now both better delineated the models and split the fitting analysis from the photobleaching.

In our new Figure 5, we show that the measurements of the per-unit-length rates of constriction and compression are significantly better fit by the Compression Feedback model than the Retention model (see new Figure 5). More importantly, the photobleaching analysis (now described in a new Figure 6) strongly argues against the Retention model and is consistent with the Compression Feedback model. In addition to this new figure structure, we have completely reworked the text section that describes these results and believe this has greatly improved this aspect of the manuscript.

2) Logic to disregard accumulation in the division plane, loss by disassembly and turnover with the cytoplasmic pool.

This criticism arose from the schematics and text in our original submission, which implied “long distance transport” of myosin foci from the cortex on the surface of the embryo into the ring. As reviewer 2 correctly pointed out, this presentation was misleading. In response, we have completely reworked the presentation of the model and significantly restructured the manuscript (see detailed third response to reviewer #2). In this reworked presentation, this criticism is no longer relevant. Please note that the experimental observations – generation of the cortical flow map, definition of the similar exponential behavior of rate of ring component accumulation, rate of cortical compression, and rate of ring constriction, and the exclusion of Retention by the photobleaching experiment – as well as the mathematical model built on them are unchanged in the revision.

As far as I understand, the only reasoning for disregarding "accumulation in the division plane" and "loss due to disassembly" (Figure 3C) is the similarity between the time course of the ring-directed cortical flow and the time course of ring myosin and anillin per unit length. However, this argument assumes that there is no turnover of myosin with the cytoplasmic pool at the division plane and the ring and that the flow within the division plane is uniform and constant (no accumulation). The authors claim that the result of FRAP excludes the exchange of the ring myosin with the cytoplasmic pool. However, their argument "If ring myosin were turning over due to exchange with cytoplasmic myosin, we would expect the FRAP curve to approach the control curve and the difference between the FRAP and control curves to disappear" is true only if all the ring myosin is exchangeable with the cytoplasmic pool and the exchange occurs rapid enough. This assumption is too strong (a kind of straw man argument). The absence of complete recovery simply tells us that there is a non-exchangeable population in the ring myosin. Moreover, indeed, the difference is getting smaller. The basis for the constant cortical flow in the division plane is unclear.

The reviewer asks whether the data from our photobleaching allows us to exclude the possibility that myosin that has already been incorporated into the ring can exchange with the cytoplasmic pool. The data in question is produced in Figure 6C. As we explain in the manuscript, if ring myosin turns over due to exchange with cytoplasmic myosin, we would expect the curve for fluorescence in the ring after the bleach to approach the control curve, and the difference between the two curves (black points in the graph in Figure 6C) to decrease exponentially. In contrast to this prediction, the difference between the two curves is about as constant as can be expected for experimental data. Although the experimental data do not completely rule out the possibility of *any* exchange, they suggest that exchange is minimal.

We are not sure what the reviewer means by “the basis for the constant cortical flow in the division plane is unclear”. We assume that the reviewer is referring to the right graph shown in Figure 3B in the revision which shows that the rate of surface area delivery into the division plane is roughly constant during the majority of constriction. We think that the reason for this is the same as why the overall constriction rate of the ring remains roughly constant despite its decreasing perimeter. As the ring closes, the exponential increase in the per-unit-length amount of ring myosin leads to a corresponding exponential increase in the per-unit-length rate of cortical surface area compression that balances the decrease in ring size leading to a roughly constant rate of cortical surface area delivery into the division plane throughout the majority of constriction.

3) Laser dissection.The authors performed laser dissection experiments (Figure 2) to assess the influence of the cortical resistance on the rate of ring closure. However, it doesn't seem to be sufficient/complete to draw any firm conclusion. It is not clear whether they are comparing the parallel and perpendicular cuts on their effects on the ring constriction. Only the absence of the effect of the parallel cut was mentioned.

We performed perpendicular cuts as a second control, which would have been useful if the parallel cuts had resulted in an effect on the constriction rate. To simplify the presentation, we have now removed the data for the perpendicular cuts from Figure 2C in the revision, and mention that perpendicular cuts also did not have an effect as “data not shown”.

The cortical flow is not spatially uniform (Figure 1), implying that the cortical tension is also non-uniform. Then, don't we need to consider the relative positioning (angle and distance) between the ring and the cut?

The reviewer is correct. Due to variable orientation of the embryos we expect variable amount of tension release. This variability would introduce variable degrees of acceleration, but in all cases the effect should have been positive, driving the mean in the same direction. In addition, the embryos presented in the figure all showed opening of the cuts to a similar degree. Those that did not open to a similar extent were not considered. This validation step additionally ensured that only the embryos with a high tension area facing the objective were analyzed. We also note that the embryos were under compression in our laser ablation setup and, as shown in Figure 1—figure supplement 1B, these embryos tend to be relatively uniformly oriented with their initial ingression axis 90 degrees to the objective axis.

Is the lesion big enough? Even after the cut the ring can still be connected to the polar cortex via the unaffected zone of cortex. What would happen if a whole polar cortex is completely separated from the other part of the cortex and the ring? The rationale for the purple line in the Figure 2C "Expected rate if […]" described in the Materials and methods "The cortical opening after ablation was approximately 35μm^2^; this translates into an additional reduction in ring radius by ~0.8μm, if the cortical surface tension dominates the ring closure rate." needs more detailed explanation.(R script to demonstrate the difficulty in distinguishing exponential curve and hyperbolic curve)x <- (0:16)/20y <- 1/(1-x)model <- nls(y~a+b*exp(c*x), start=list(a=1, b=1, c=1))xx <- (0:80)/100z <- predict(model, newdata=list(x=xx))plot(x, y, ylim=c(0,5))lines(xx,z)coef <- summary(model)$coefficients[,1]fit = sprintf("y=%.4f+%.4f*exp(%.4f*x)", coef[1], coef[2], coef[3])legend("topleft", legend=c("y=1/(1-x)", fit), pch=1, lty=c(0,1), pt.cex=c(1,0))

The lesions that were made with the laser across the cortex were as large as possible, spanning the entire visible area of cortex on the anterior side of the embryo (~10um in length). We believe the lesions are large enough to result in a detectable change in rate if cortical surface tension limits ring constriction. The cortical opening after ablation was approximately 35 µm^2^ and, at the time of the cut, the rings were approximately 14 µm in diameter. If the ring size is determined by the available cell surface, then this gain in the surface area should result in additional area increase of the division plane. Assuming that the surface area gain resulting from the lesion is distributed between the two cells it would result in ~18µm^2^ increase in the division plane, which corresponds to an additional ~0.4 µm reduction in ring radius. This reduction in size would lead to an increase in constriction rate of ~0.03µm/s, during the 13s interval between image acquisitions. This expected increase is marked with the purple line in Figure 2C. We have improved the clarity of this section in the revision and now refer the readers to the supplement where the detailed calculation is presented. The laser cutting result is also supported by our experiments inhibiting the Arp2/3 complex, which is expected to reduce effective cortical viscosity and thus cortical tension. This perturbation also did not alter the constriction rate, supporting the conclusion that cortical tension does not limit the rate of ring constriction.

While it would be interesting to see what would happen if the whole polar cortex could be completely separated from the other part of the cortex and the ring” this would be very technically challenging given the geometry and is beyond the scope of the current work.

Reviewer #3:[…] 1. Due to no cell cortex in yeasts but the ring constricts at a constant rate (Pelham and Chang, Nature, 2002), the references of Wu and Pollard 2005 paper in the Introduction and Discussion are somehow misleading. In fission yeast, myosin-IIs mostly retain in the ring during its constriction, they are also highly dynamic by exchanging between the ring and cytoplasm. The similarity and difference between Wu and Pollard, Carvalho et al, and the current work should be discussed.

We agree that a thorough comparison of our work with what is known about ring dynamics in fungi, particularly *S. pombe*, is of importance. However, in this paper we already have had a challenging time clarifying our measurements, the analytical model and placing the findings in the context of what is known about cortical and contractile ring dynamics in animal cell systems, We believe that a comparative discussion including fungi would much better suited to a collaborative review format, which we would be excited to work on in the coming months after publication of this work. We also agree with the point that the reference to the Wu and Pollard 2005 paper in the introduction and discussion is somewhat misleading because of these differences, so we have now removed this reference.

2. In Zhou, M., & Wang, Y. L. (2008), "Distinct pathways for the early recruitment of myosin II and actin to the cytokinetic furrow" (Mol Biol Cell, 19(1), 318-326), it is found that myosin-II is recruited to the furrow mainly by de novo assembly, but not cortical flow, during early cytokinesis in mammalian cells. This and other similar works on cortical flow and de novo assembly should be cited and discussed.

In this revision, reviewer 2 persuaded us to place our model in the context of Rho-based signaling, which we have done. Thinking hard about this made us realize that we are proposing a hybrid model that sits halfway between de novo recruitment downstream or Rho-based signaling and recruitment by cortical flow. As reviewer 2 pointed out, recent work, including our own (Mangal et al., 2018; PMID: 29311228), indicates that contractile ring proteins are recruited within the Rho zone and are cleared from the cortex outside the Rho zone. Incorporating this into our model, where the relevant concentration of components on the cortex is at the point where the cortex enters the rho zone/contractile ring, we propose that the compression of cortical surface in the Rho zone pulls new cortex into the Rho zone that is then loaded with contractile ring components triggering the initiation of compression. For a detailed description of the revised model see Figure 4 and the response to reviewer 2 (point 3) above. In the revised model, the flow of new cortex into the ring is required for the de novo loading of ring components. In other words, if our model is correct, both cortical flow and de novo loading contribute to the evolution of the component levels in the ring during constriction.

As requested by the reviewer, we have added a paragraph into the Discussion, that discusses the relationship between our model and this prior work. “It is worth noting that our proposed model represents an interesting twist on an ongoing debate in the cytokinesis field as to whether contractile ring components are recruited via cortical flow into the ring (Cao and Wang, 1990; DeBiasio et al., 1996; Fishkind et al., 1996; Wang et al., 1994) or de novo from the cytoplasm downstream of RhoA-based signaling (Vale et al., 2009; Yumura, 2001; Zhou and Wang, 2008). […] We would therefore propose that both the de novo loading of components by Rho-based signaling and compression-driven flow contribute to the evolution of the component levels in the ring during constriction.”

3. A figure supplement showing the cortical flow map at cell poles before and during ring constriction will be useful.

We have not been able to monitor cortical flow at the poles due to the high surface curvature in this region. Even in attempts we have made to image embryos “on end”, the observable surface area of the polar region is very small (see Author Response Image 3). The size of the imageable region is comparable to the size of the flow averaging window used for correlation between consecutive slides, which makes flow analysis uninterpretable. The area can be increased by increasing the number of z planes used in the maximum intensity projection (Author Response Image 3), however, each additional plane increases the area by a small amount due to high curvature and the plane would be collected with a larger time delay from the first plane decreasing our temporal resolution. Finally, the region between the reconstructed polar region and the flow map presented in this paper would prevent us from connecting it with the remainder of our flow map, limiting its utility. In conclusion, although we would like to be able directly observe cortical flow at the poles, due to the technical limitations outlined above, this is not currently a viable approach with our imaging setup.

**Author response image 3. respfig3:** Technical limitations prevent imaging of cortical flow at the cell poles.

4. The terms "cortex surface" and "surface gain" are confusing. Because the plasma membrane was not directly tracked, it should be make clear what you mean here. Otherwise, casual readers may think the plasma membrane is inserted at cell pole, which is likely, but not tested here.

Yes, this is indeed a problem. To eliminate these sources of possible confusion, we have included a new introductory section (below) that clarifies this issue. We have also edited the text to consistently use the terms cortical surface area, cortical surface compression, and cortical surface expansion, which are what we measure using our method.

“During the first division of the C. elegans embryo, the surface area of the cell increases by ~40% to accommodate the shape change that generates the daughter cells. […] How deposition of plasma membrane, the fluid lipid layer that overlies the cortex, is controlled and where it occurs are distinct questions that we will not discuss here.”

5. The Materials and methods section is not clear regarding how background fluorescence was subtracted. Which region was used as the background?

This was not clear because we were missing a reference to Figure 3—figure supplement 3 where we illustrate the region used for the background intensity measurements. The background was calculated as the average value inside two 11x11µm rectangles positioned 2µm away from the division plane inside the anterior and posterior cells as illustrated in this figure.

6. The reference citations are not consistent, sometimes multiple authors from the same paper are listed.

This problem has been corrected.

[Editors' note: the author responses to the re-review follow.]

*[…]* Below, I mention what needs to be done for each point. Although I think I can decide on any submitted revision, I might call upon one of the referees, if required.

1) The revisions to the manuscript have clarified their model so that readers can better understand what the authors claim to demonstrate. I remain unconvinced of the authors model. My concerns are due to the fact that the manuscript is largely based on an inference of cortical compression.I want to first re-state that the data shown are carefully obtained. The measurements of cortical flow are of interest. The authors provide evidence that the rate of furrow formation is limited by internal viscosity in the contractile ring, which is a novel insight and an important point.Notably, the authors and I communicated following the first version. This communication has lead to a more clear description of their model. Yet during that process I communicated the gist of the comments below, yet they remain unresolved in this version. Specifically, in the previous version, the manuscript gave the impression that cortical flows delivered myosin into the furrow region, thereby accelerating ingression, flows, and myosin accumulation through a form of positive feedback. Now, the authors have clarified their interpretation as follows:a) The amount of cortical surface area that flows into the furrow region exceeds the surface area of the division plane.b) As a consequence of (1) and an unstated assumption that cortical surface area is not disassembled, the authors infer that cortical surface area is compressed. There is no direct evidence for cortical compression except at the very earliest stages of furrow formation. Interestingly the rate of flow at the stage where compression is observed is 2-3x lower than that during furrow ingression (Figure 1B vs. 3A).

The reviewer states that our model is based on “inference” of cortical surface area compression, which is not the case. In fact, we are directly measuring this property. We suspect that the reviewer’s confusion stems from a misunderstanding with respect to the terminology we are using. In mechanics, “compression” describes a reduction in material size, thus the rate of “cortical surface area compression” is simply defined as the rate of reduction in cortical surface area. This reduction could be accompanied by (1) concentration of cortical components or (2) cortical component disassembly. We suspect the reviewer may be confusing the mechanics definition of “compression” with the lay use of the word, which often corresponds only to the first possibility. We have added text to clarify the definition of this term within the body of the text to avoid further confusion.

Our measurements allow us to conclude that cortical surface area is compressed throughout cytokinesis, both in the very early stages of furrowing when the ring is still on the surface (t/t_ck_=0.1, shown in Figure 3A) and after the ring has ingressed from the surface (Figures 3B and 3C). Measuring cortical surface area compression as a gradient in cortical velocity (as we do in Figure 3A) is equivalent to measuring it based on the difference between the flow of cortex into the division plane and the area of the division plane (as we do in Figure 3B and C).

Again, we emphasize that this conclusion is agnostic to the mechanism by which this reduction occurs. Cortical surface area compression could be accompanied by actin-myosin disassembly, in which case components would not accumulate as cortical surface area decreases. Alternatively, in the absence of a coupled mechanism for acto-myosin disassembly, a reduction in cortical surface area would lead to an increase in the concentration of cortical components.

Overall, I remain unconvinced by the author's interpretation of their results. There is some value in their quantitative model, though (i) it is not well constrained, there is no reason why the flow has to be the source of positive feedback and (2) it has not been extensively tested experimentally. However, if the authors want to publish their interpretation, I have no strong objection, particularly in eLife where readers can readily access the reviews that indicate that experts in the field do not subscribe to their interpretation of their results.

The reviewer does not take issue with our experimental findings. He agrees that our results show that there is a on ongoing reduction of cortical surface area (cortical surface area compression) within the division plane and that the accumulation of contractile ring components exhibits exponential kinetics suggesting positive feedback. Where the reviewer has difficulty is in how we translate these findings into the assumptions that underlie our analytical mathematical model for contractile ring dynamics.

Based on the fact that cortical compression within the ring and the accumulation of contractile ring components occur with parallel exponential kinetics, we propose that cortical compression is part of a feedback loop that drives component accumulation. This is a natural proposal since cortical compression would drive component accumulation unless there is a mechanism to disassemble components as the cortex is compressed that prevents this accumulation.

In contrast to our proposal, discussions with the reviewer on this point have revealed that he prefers the idea that there is a different positive feedback loop that drives contractile ring protein accumulation, whose mechanism is as yet unknown. In this model, the rates of cortical compression and ring constriction would follow, but not contribute to component accumulation. For the reviewer’s model to work, one needs to assume that cortical surface compression *does not* concentrate contractile ring components. Thus, this alternative model requires two new additional mechanisms, for which there is no experimental evidence. In contrast, for our model, all of the necessary elements have been demonstrated to occur.

We have tried to draw a clean line between our experimental results and the modeling portion of the paper and have strengthened this separation in the revision. Philosophically, we also strongly disagree with the reviewer about the purpose of modeling in the context of scientific exploration. We do not think that it is possible to prove a model (only to disprove one), or that a model should be viewed as an “interpretation” of experimental results. Modeling is a tool to quantitatively explore whether a proposal would/could work. Our model represents a simple, reasonable framework that can explain all of the available data. The point of putting forward a model is that it constitutes a concrete framework that can be challenged by future work advance our understanding of a process. To propose a model, it is not necessary to rule out every possible alternative model that could arise as new data emerges.

c) The total cortical surface area is inferred to increase exponentially and it parallels the increase in ring components.

To clarify, what we show is that there is an exponential increase in the amount of cortex that is compressed per unit length of the ring during constriction. This increase is offset by the decrease in ring perimeter during constriction so that the overall amount of cortex compressed by the ring and the overall rate of ring constriction remain relatively constant during a large portion of constriction.

d) Yet, the myosin that flows into the furrow region is not a major contributor to ring myosin, rather it largely disassembles. Indeed, in en face views of the division plane, there is no detectable flow of myosin from the "exposed" cortex to the ring.e) However the flow of cortex is proposed to provide additional, initially "naive", cortical surface area that is then patterned by RhoA (Figure 7), and it is this exponentially increasing cortical surface area that leads to exponentially increasing levels of ring myosin. This begs the question, what is "naive" cortex?(Editors’ recommendation: rewrite significantly for clarity)

Once cortex enters the Rho zone, RhoA-based signaling recruits an ensemble of proteins to generate a highly-cross linked matrix. In addition to formin-nucleated actin, anillin, the septins, and myosin are also concentrated in the Rho zone/contractile ring (which are equivalent in our model). Assembly of this cross-linked cortical meshwork likely explains why whereas myosin on the cortex turns over with a t_1/2_ of ~30s (PMID:20852613), whereas myosin in the contractile ring exchanges slowly or not at all. In our model, naive cortex is cortex outside the Rho zone (naive to RhoA-based signaling). Although this cortex is not loaded with high levels of contractile ring components, it is decorated with myosin puncta that we use as fiduciary marks to track cortical movements. One consequence of cortical compression of the cortex within the Rho zone/contractile ring is that it will, by definition, pull naive cortex into the Rho zone.

2) At the core of the issue with this model is inference of cortical surface area compression. First, the authors claim that cortical compression can be readily inferred from the difference between the amount of cortical flow into the division plane and the area of the division plane.

As discussed above, this issue arises from a misperception by the reviewer of the definition of cortical surface area “compression”. Cortical surface area compression is defined as a reduction in cortical surface area; this property is directly measured by our methods. The reviewer instead thinks that cortical compression only occurs when there is a reduction in surface area without loss or disassembly of cortical components. We have now clarified the definitions of cortical expansion and compression in the text to prevent confusion.

Yet, the actomyosin cortex is dynamic, in addition to compressing and expanding, it can assemble and disassemble. Indeed, the manuscript shows that at the poles, cortical surface area is created as the cortex flows into the furrow region.

We do not use the term “created” because it implies that new cortex is assembled. Our data show that cortical surface area increases (cortical surface area expansion) at the cell poles as the ring constricts. However, we do not know the mechanism underlying this expansion. In our Discussion, we outline three possibilities.

“One possibility is that the polar cortex is less stiff than the rest of the cortex, causing it to stretch and thin in response to ring constriction-induced tension. […] Consistent with this last idea, blebs have been reported at the cell poles in cultured vertebrate and *Drosophila* cells, where they have been proposed to release tension at the poles (Hickson et al., 2006; Sedzinski et al., 2011).”

Two of these possibilities (possibilities 2 and 3) would involve “creation” of new cortex as the reviewer suggests. However, it is also possible that cortical surface area increases without the addition of new cortical material because the polar cortex is stretched and becomes thinner (possibility 1)

And, myosin – a key component of the cortex – is largely assumed to be lost as the furrow flows into the division plane (see point 3). Thus, while cortical compression is possible, cortical disassembly is another possibility, which is not given sufficient consideration. In fact it is a strong possibility given that there is loss of a key component of the cortex, myosin.(Editors’ recommendation: rewrite significantly for clarity)

On the same theme as the points above, cortical compression and cortical disassembly are not mutually exclusive. Our analysis of cortical flow shows that cortical surface area compression is occurring. Mechanistically, compression of the cortex along the axis perpendicular to the ring could be accompanied by component disassembly or could lead to the concentration of contractile ring components.

Our mathematical model develops the idea that positive feedback between ring myosin and cortical surface compression along the axis perpendicular to the ring drives the exponential increase in the per-unit-length amount of ring myosin that maintains the high overall constriction rate during ring closure. In our model, we assume that compression of the cortex along the axis perpendicular to the ring does concentrate ring components. This is a reasonable assumption based on our experimental finding that the per-unit-length amounts of ring myosin and anillin increase at a rate proportional to the rate at which cortical surface is compressed into the ring. If we assume, as the reviewer proposes, that compression does not concentrate ring components, we would have to make two assumptions for which there is no experimental evidence. We would have to assume that compression is accompanied by an as yet unidentified disassembly mechanism that prevents the cortical components from increasing in concentration as the cortex contracts, and we would have to assume that there is a different as yet undescribed feedback loop that somehow drives the exponential accumulation of contractile ring components as the ring constricts. At the reviewer’s request, we have mentioned in the Discussion that our data do not rule out this possibility. However, as there is no evidence for an alternative positive feedback loop operating over the timescale of constriction, and we therefore have no details about how such a feedback loop would work, it is not reasonable to suggest that the mathematical model in our manuscript should center around this potential hypothetical situation.

3) Furthermore, the authors have not explained why "naive cortex" would be required for the zone of active RhoA to drive an increase in myosin accumulation?(Editors’ recommendation: rewrite significantly for clarity)

The reviewer’s question is about the idea we introduce in our mathematical model that there is naive cortex, which we define as cortex that was not originally in the spindle-defined Rho zone, that enters the Rho zone due to compression of the cortex within the ring. After this cortex enters the ring, it is loaded with contractile ring components (like the other cortex in the Rho zone) and starts to compress.

The assumption that this would occur was introduced into our model at the request of this reviewer, who asked that we integrate our model with the concept of the Rho zone. The Rho zone is an equatorial region where RhoA acts on the cortex to promote the accumulation of contractile ring proteins. The Rho zone is defined by the spindle; if the spindle is moved after the contractile ring forms, the Rho zone moves with the spindle and the contractile ring either slides, if the distance is short, or regresses and is reformed to be coincident with the new spindle positioned RhoA zone. In other work, we have shown that in the absence of cortical contractility, contractile ring components accumulate across a 10 µm wide zone that spans the cell equator (PMID: 29311228), which is in close agreement with where cortical compression is initially observed on the cell surface before the ring ingresses into the division plane (Figure 3A). Our data shows that the cortex within the Rho zone is compressed along the pole-to-pole axis. This causes a flow of cortex that was not originally in the spindle-defined Rho zone (“naive cortex”) to be pulled into the Rho zone. The only reasonable assumption to be made in this framework is that the new cortex that enters the Rho zone will also be subjected to RhoA-based signaling and will recruit contractile ring components and start to compress.

4) Given that ARP-2/3 nucleated actin is likely a nucleator of some of the actin in the cell cortex, it is notable that its depletion does not dramatically affect the rate or extent of furrow ingression in otherwise WT embryos, as has been shown previously (PMID 22226748). This raises the follow-up question: what is "naive" cortex in ARP-2/3 depleted embryos?(Editors’ recommendation: rewrite significantly for clarity)

The reviewer is really asking a general question for the field. The impacts of inhibition of the Arp2/3 complex on the integrity of the cortex are generally much milder than one might expect, suggesting that other sources of actin contribute to the cortex outside of the contractile ring. This is an important question, but not one that we should be expected to answer here. Our data further indicate that the Arp2/3 complex is not essential for loading contractile ring components onto the cortex in the Rho zone.

5) In the author's rebuttal letter (reviewer 2, fourth response) the authors state, "The reviewer would propose that there could be another source of positive feedback that controls myosin accumulation (for example some type of ring intrinsic feedback loop involving Rho-based signaling), and that exponential accumulation of myosin arising from this as yet un-described feedback loop could, in turn, control the constriction rate and the rate of cortical compression. We do not disagree that this could be the case." Positive feedback in RhoA signaling during cytokinetic processes has been demonstrated, (PMID 26479320), and there is evidence for a mechanism in which RhoA might generate positive feedback through the recruitment of centralspindlin and its activation of the RhoGEF ECT-2 (PMID 26252513).(Editors’ recommendation: rewrite significantly for clarity)

The reviewer is referring to work by Bill Bement and George von Dassow showing that coupling between Rho signaling and actin assembly can lead to the formation of waves of acto-myosin contractility that move across the cortical surface. The main finding of this work is actually that there is *negative feedback* in which RhoA driven F-actin assembly suppresses RhoA activation. The existence of the waves also suggests that at the front of the waves where actin has not yet assembled, active RhoA can promote RhoA activation. This positive feedback at the front of the waves is not likely to be relevant to the feedback we document during constriction. The contractile ring is not at the front of a cortical wave where actin has not yet assembled – it is actually more analogous to the back of the wave where actin has assembled and has become concentrated. Thus, this work would suggest that negative feedback could reduce RhoA signaling within the constricting ring. The other paper the reviewer cites includes a schematic model suggesting that the GAP domain of centralspindlin activates the GEF through a RhoA dependent interaction, which the reviewer is implying could be relevant to the positive feedback at the wave fronts in the Bement and von Dassow experiments. In addition to the fact that the contractile ring at the front of the furrow is unlikely to be analogous to the actin-free wave fronts in the Bement and von Dassow work, there is no experimental support for the idea that an interaction between the centralspindlin GAP domain and the ECT-2 GEF domain activates the GEF in the paper that the reviewer mentions. In the paper they say that “We assayed for activation of the ECT-2 GEF activity by the CYK-4 GAP domain in vitro. However, we have not yet been able to detect stimulation of GEF activity (data not shown).”

6) Given the topic of this manuscript, it is surprising that the authors do not mention that local RhoA activation is sufficient to induce furrow formation (PMID 27298323) and all of the literature concerning the mechanism of RhoA activation during cytokinesis.(Editors’ recommendation: consider discussing this paper)

It is well established by a lot of prior work that RhoA activation, due to spindle-based signaling leads to the recruitment of contractile ring proteins. As there is much literature on this topic, we cite reviews that summarize the field. The paper that the reviewer mentions shows that light mediated recruitment of a RhoGEF can replace spindle-based signaling, leading to the assembly of furrows that can partially ingress; this is not a paper that we think is essential to cite. The focus of our manuscript does not center on the mechanisms of RhoA activation. Thus, there is a lot of literature that is more relevant to the topics at hand that we do need to cite and discuss.

7) The authors state, "We propose that, due to polar relaxation, the compressing cortex pulls naive cortex not patterned by the initial round of RhoA signaling, into the Rho zone." Here the authors are generating confusion between terms that have a different historical meaning. Polar relaxation was used to describe a mechanism by which astral microtubules might induce a net increase in equatorial contractility by the local inhibition of contractility at the poles (polar relaxation). Here, they are discussing how existing equatorial contractility induces flow of cortex away from the poles. These terms are already sufficiently confused in the literature, it would be better to avoid adding to it.(Editors’ recommendation: rewrite significantly for clarity)

The quote the reviewer highlights is a central point of our manuscript, and we strongly disagree with the assertion that we are confusing historical terms. In fact, our work significantly clarifies the definition of polar relaxation, which has been murky in the literature, and shows how coupling of polar relaxation (expansion of the polar cortex) to compression (reduction in surface area) of the equatorial cortex generates the observed pattern of contractile ring dynamics.

Our analysis of cortical flow and our laser cutting experiments show that when the contractile ring pulls on the adjacent cortex, the cortex at the poles expands in response to the ring-generated tension without providing significant resistance that would affect the rate of ring closure. This is in contrast to the behavior of the intervening cortex between the ring and the pole; when subjected to the same ring-generated tension, this cortex flows towards the ring without expansion or compression. Thus, polar relaxation is the ability of the polar cortex to expand when put under tension. In the quote that the reviewer highlights, we explain how polar relaxation works together with spindle-based RhoA signaling that promotes the accumulation of contractile ring proteins that cause compression of the equatorial cortex (reduction in cortical surface area). Compression of the equatorial cortex (reduction in cortical surface area) is coupled to expansion (increase in cortical surface area) of the polar cortex. A main point of our manuscript is that the polar cortex expands not only to provide the cortical surface area need to cover the embryo, but also feeds extra cortical surface into the ring as it constricts. Polar relaxation therefore allows the compression driven flow of cortex into the contractile ring at the cell equator. We have revised the text to strengthen these points and avoid any lack of clarity with respect to historical terms.

8) One paper published during their revision is quite relevant: PMID: 29146911. DOI: 10.1038/s41467-017-01231-x. I suggest that the authors cite and briefly discuss the paper in their final manuscript.(Editors’ recommendation: consider discussing this paper)

The reviewer-suggested reference (to a paper by Andrew Wilde and colleagues entitled “Cytokinesis requires localized β-actin filament production by an actin isoform specific nucleator”) is a very interesting paper. However, it is not clear to us why the reviewer would like us to cite/discuss this in our manuscript. The reference suggests that in human cells anillin contributes to activation of the cytokinesis formin DIAPH3 and that this formin nucleates filaments assembled from β- rather than γ-actin. Other than assuming that actin is nucleated by the cytokinesis formin in the Rho zone in our model, our manuscript does not deal with the cytokinesis formin or the isoform content of the actin filaments in the ring.

9) Exponential/hyperbolic accumulation.In Figure 5, the authors compare exponential curves and hyperbolas for fitting with the experimental data and conclude that exponential curves fit better. However, it is unclear whether their comparison is fair. For fitting with the data of mean fluorescence per unit length with an exponential curve (Compression feedback), three parameters, i.e., the amplitude, the time constant and baseline can be adjusted. On the other hand, for fitting with a hyperbola (Retention model), it is unclear what the formula for R(t) looks like and what degree of freedom was allowed.

The retention model described by the equation C/C0=R0/R+b does not have a time constant component, so we could only use two parameters fitting the model R0 and b. After fitting the Retention model to our data in the range of time from 0.2 to 0.6 we obtained R0=1.0 and b=0.13. Since parameter b value did not affect the fit in a significant way (see Author response image 4), we removed the extra parameter to present the retention model in more favorable light, as a model with only one parameter. However, as the reviewer points out such a decision does raise the question of fairness of the fit, so we have now added the second parameter back into the model and show the complete fitted equation in Figure 5B (version in Author response image 4 rather than version in Author response image 4).

**Author response image 4. respfig4:** Comparison of the Retention model fit without (**A**) and with (**B**) inclusion of a baseline term.

*As I pointed out in the previous reviewer comments, clear distinction between the exponential curve and hyperbola is not trivial. The authors' own data and interpretation demonstrate this difficulty. First, in Figure 3C, "Cortical compression (rate per unit ring length)" is fitted with an exponential function. However, this quantity* dAcompdt1R *should follow a hyperbolic increase in time since the first term,* dAcompdt*, is largely constant (Figure 3B 'Normalized Surface Area Flux') and the second term, 1/R, is an inverse of a linearly decreasing function of time (Figure 1A). Second, in the same panel, "Ring shrinkage rate per unit length (*-dRdt1R*)" is also fitted with an exponential curve. However, this quantity should also be hyperbolic for the same reason (*-dRdtis constant most of the time during furrow ingression Figure 1A). These examples nicely demonstrate the difficulty in distinguishing between exponential and hyperbolic changes by curve fitting with a set of data that are not really suitable. The authors' approach doesn't have sufficient power to discriminate between possible theories.(Editors’ recommendation: rewrite significantly for clarity as well as discuss the limitation of the curve fitting approaches you have taken)

First, we would like to emphasize that we don’t disagree with the reviewer’s point that our curve fitting analysis does not definitively distinguish between hyperbolic accumulation (indicative of component retention) or exponential accumulation (consistent with a feedback-based model). Our main goal in comparing the ability of the two models to fit the data in Figure 5 is to motivate the photobleaching experiment in Figure 6 that we designed to distinguish between retention and feedback-based accumulation. In the revision, we have edited the text on this on point to make it clear in the text.

“Comparison of the fits to the data for the per-unit-length amounts of ring myosin and anillin and the rates of ring shrinkage and cortical compression for the two models (Figure 5B, Figure 5—figure supplement 1) suggested that, whereas the Retention model could approximate the data, the Constriction-Coupled Disassembly with Compression Feedback model fit the data significantly better.

Since both models could approximate the data, we designed a photobleaching experiment to definitively distinguish between them (Figure 6A).”

Although the curve fitting is not definitive, we do think that the comparison of the ability of the two models to fit the data does lend support for accumulation due to a feedback-based mechanism versus component retention. Approximation of the rate of surface area compression with a constant is fair only in the region between 0.2 to 0.6 (Figure 3B) and approximation of the rate of ring closure with a line is valid only between 0.2 and 0.8 (Figure 1A). Thus, one can expect an agreement between a hyperbola and the data for the rates of ring shrinkage and cortical compression only in the region between t/t_ck_= 0.2 to 0.6, possibly extending toward 0.7 (as we show in Figure 5B). In contrast to the crude approximation of this data with constants that led to the proposal of retention models, here we use the data to directly obtain rates per unit length throughout the entire process. For this analysis, we specifically collected data in the time regime prior to the linear dependence of the ring size with time (t/t_ck_=-0.2 to 0.2). This analysis revealed that the rates of ring shrinkage and cortical compression are better explained with a single exponential, which fits the data well for the entire process (t/t_ck_= -0.2 to 0.8). Our analysis suggests that the deviation of the data from the hyperbolic curves in the t/t_ck_=-0.2 to 0.2 time range is not a measurement artifact, but a real observation (Figure 5B).

10) Exchange of ring myosin with cytoplasmic pool.I am really confused with what the authors are actually thinking about the exchange of myosin with the cytoplasmic and cortical pools.

We appreciate the reviewer’s confusion, as the photobleaching experiments that we use to assess exchange are the most difficult data in the paper to understand. It is particularly challenging to analyze the consequences of photobleaching in the contractile ring because the structure itself is continuously changing (shrinking in perimeter while increasing in per unit length component levels). To address the reviewer’s concern, we have completely rewritten this section:

“Since both models could approximate the data, we designed a photobleaching experiment to definitively distinguish between them (Figure 6A). […] In summary, our division plane photobleaching experiment at the one-cell stage is consistent with the Constriction Coupled Disassembly with Compression Feedback model.”

Based on the whole division plane FRAP experiment in Figure 6, they exclude the exchange of the ring myosin with cytoplasmic myosin. On the other hand, in the schematic in Figure 4, the recruitment of cytoplasmic myosin to the Rho zone is depicted as the major source of the increase of the cortical myosin in the Rho zone. Although it was not explicitly stated in mathematical modeling, myosin on the cortical flow within the Rho zone and myosin accumulated at the contractile ring behave differently as to new recruitment and removal by disassembly. In the FRAP in Figure 6, both of these myosins, as well as myosin on the 'naive cortex' in the division plane, were photobleached. Recovery seems to have started at the contractile ring instead of the flowing cortex outside of the ring. The simplest explanation would be that there is an exchange of myosin at the ring.

As we discussed in response to the previous point, our division plane photobleaching data suggest the existing myosin in the ring is disassembled in proportion to reduction in ring perimeter during ring closure (Constriction-Coupled Disassembly). At the same time, our data suggest that new myosin is added to the ring via a feedback-based mechanism at a rate proportional to the amount of existing myosin.

The reviewer’s question relates to how we incorporate these findings into our mathematical model. First, it is important to note, and we have clarified this point in the revised text, that in our model the Rho zone and the contractile ring are the same thing. Myosin in the Rho zone is myosin in the contractile ring. We do think that myosin inside the contractile ring/Rho zone behaves differently from cortical myosin outside the contractile ring. Cortical myosin outside the ring, like the myosin in the foci on the embryo surface that we follow during our cortical flow analysis, has been shown to turn over with a t_1/2_ of ~30s (PMID: 20852613). In contrast our data suggest that once myosin is loaded onto the cortex in the contractile ring/Rho zone, it does not exchange with myosin in the cytoplasm and is instead disassembled in a constriction-coupled fashion. We think this difference arises because in the ring RhoA-based signaling recruits an ensemble of proteins (including formin-nucleated actin, anillin, the septins) in addition to myosin to generate a highly cross linked matrix.

Our cortical flow analysis indicates that the per-unit-length amount of myosin in the ring increases at a rate proportional to the rate at which cortical surface flows into the ring due to cortical surface compression within the ring. We incorporate this into our mathematical model by proposing that naive cortex, which we define as cortex outside the Rho zone (naive to RhoA-based signaling) is loaded with a fixed concentration of myosin (m_rho_), along with other contractile ring components, as it is pulled into the Rho zone due to cortical surface compression within the ring. Prior to being pulled into the ring it is decorated with myosin puncta that we use as fiduciary marks to track cortical movements, but since our imaging suggests that myosin levels are significantly lower outside of the ring, in our mathematical model, we propose that the amount of myosin on the naive cortex outside the ring is inconsequential compared to the amount of myosin loaded onto the cortex when it is pulled into the Rho zone. Thus, in our model we do not consider myosin on the cortex outside of the Rho zone/contractile ring, and all myosin within the Rho zone/contractile ring behaves in the same fashion.

A constant level of per-unit-length bleached myosin is a basis for their compression feedback model. However, bleached myosin calculated by the two formulas made by exponential fitting is not constant. Contrary to their description (subsection “Fluorescence recovery after photobleaching of the division plane is consistent with the Compression Feedback model”, last paragraph), the two curves are getting closer (see graphs that can be generated by running an R script at the bottom). This point was clearer in Figure 4C of the original submission. If we apply the same logic as later in the aforementioned paragraph, the data indicate that the recovery is at least partially due to the exchange with cytoplasmic myosin. I don't understand why they could assert "We also note that, consistent with our prior observations at the 4-cell stage (Carvalho et al., 2009) we did not observe evidence of turnover of ring myosin due to exchange with myosin in the cytoplasm."(Editors’ recommendation: rewrite significantly for clarity and explain limitations)

The reviewer’s concern is that the difference between the fitted lines has an exponential dependence due to the difference in the exponential prefactors. The prefactors in front of exponential terms were fitted to the data, thus it is reasonable that their values would not be identical and the difference of 5% should not be considered as the evidence of an additional biological process. Rather than calculating the difference between the two fitted curves, what the Reviewer should focus on is the data representing the difference between the two measurements. This data shows no significant trend during the ring closure (i.e. is about as constant as one can expect given that it is experimental data). Therefore, while myosin turnover within the ring is possible, our data shows that it happens on a much slower time scale than the ring closure.

We also note that whether or not there is a small amount of exchange does not impact our main conclusions. It would mean that in addition to being lost in proportion to the reduction in ring perimeter as the ring constricts, an additional small proportion of the myosin would be exchanged with myosin from the cytoplasm as the ring constricts. Thus, the per unit length amount of bleached myosin in the ring is either remaining constant or is slightly decreasing during constriction. In either case, the per unit length amount of myosin in the ring is clearly *not* increasing as 1/R, ruling out the possibility that the increase in the per unit length level of myosin in the ring is due to retention. Thus, since the per unit length amount of myosin in the ring increases exponentially as the ring closes, this exponential increase must be due to addition of new myosin to the ring.

11) Consistency with Carvalho 2009.The authors' argument in Figure 6B is valid to exclude the retention model without any exchange at the ring. However, the same logic also strongly argues against the model proposed by Carvalho (2009), which excludes both the exchange of the ring myosin with cytoplasmic myosin and with the nearby cortex. The tornado-shaped non-recovery zones in the kymographs were explained by the closure of the ring and slow exchange within the ring in the absence of the exchange with the cytoplasm nor delivery by flow from the flanking cortexes. However, if the current model is correct, the flow from the flanking cortex should cause a gradual recovery in the tornado-shaped zones in the kymographs. In other words, the current model is not consistent with the data by Carvalho (2009).

In contrast to the reviewer’s suggestion that our current one-cell stage data is inconsistent with our prior results at the four-cell stage, we think that our current model explains a key feature of our bleaching data at the four-cell stage that we did not understand at the time. In our prior work (Carvalho et al., 2009) we made kymographs along the contractile arc to follow the fate of a bleached spot in the arc as the ring constricted. The kymographs that we made in embryos expressing myosin::GFP exhibited a tornado-like shape generally consistent with the bleached zone shortening due to constriction-coupled disassembly (Carvalho et al. 2009). However, in our prior work we also noted that if the narrowing of the gap was due solely to ring constriction, we would have expected the gap to decrease by 50% in length over a 95s interval, whereas in reality we observed to the gaps to narrow more rapidly, typically having the appearance of being filled in to some extent by 60-90 s. The model that we propose here is that there is constriction-coupled disassembly – just like we proposed previously – however in addition, components also accumulate along the entire ring due to compression feedback along the axis perpendicular to the ring. The accumulation due to compression feedback, by adding to the fluorescence signal uniformly along the ring, is predicted to have precisely the effect that we observed in our prior work of causing the tornados to narrow more rapidly than expected for constriction-coupled disassembly alone.

To address the point of how the results of our prior spot bleaching experiment at the four-cell stage (Carvalho et al., 2009) relates to the division plane bleaching experiments that we perform at the one- and four-cell stage in this manuscript, we added a new figure to the revision (new Figure 7) and an accompanying text section:

“Finally, we wanted to assess whether the results of our division plane bleaching experiment at the one-cell stage are consistent with our prior work bleaching a spot on the contractile ring in embryos expressing fluorescent myosin at the four-cell stage. […] New myosin appeared in the bleached four-cell stage arcs in a fashion very similar to the bleached rings at the one-cell stage, suggesting that a similar mechanism delivers new myosin into the ring to drive the increase in the per-unit-length constriction rate at the one- and four-cell stages.”

In Figure 3—figure supplement 4, the authors quantified the per-unit-length amount of myosin in 4-cell stage division. This should be essentially equivalent to Figure 4D in Carvalho (2009), from which they had concluded that the per-unit-length amount of myosin in the ring is constant (note: this is reproduced as Author Response Image 2 in their reviewer response, hiding the latter half of the time scale where the drop was observed before the sudden 1.3-fold increase at 10 µm perimeter. In addition "In contrast, we observed a ~1.3 fold increase for all three components." is also misleading as they mentioned the 1.3-fold increase only in the last few µm, sticking to the constant per-unit-length level. Additionally, the point of transition is 18 µm in Figure 4D of Carvalho (2009) but 25 µm in their rebuttal. It is not clear why at a glance they look different (or the author could conclude differently). Are they based on the same set of image data? Or, was the recording newly performed? How do they look if they are plotted with the same x-axis (the perimeter of the ring or fraction of ring closure)?

First, we want to correct some misperceptions that are evident from the reviewer’s question. The 4-cell stage data in our manuscript is new data, not data reproduced from our older work (Carvalho et al., 2009). The 1.3-fold increase in component levels that we are talking about is not the tail at the end of constriction that the reviewer is referring to that occurs when the ring is less than 10 µm in perimeter – we do not analyze this region (or anything after a perimeter of 20 µm) because it represents the transition of the ring from constriction to assembly of the intercellular bridge and abscission. The 1.3-fold increase that we are referring to is the increase that occurs during the first 60% of closure prior to contact with the midzone – which is the region we showed in Author Response Image 2. Furthermore, in looking back, we realized that we had erroneously marked the four-cell stage transition point at 18 µm in Figure 5 from our 2009 paper, the correct value for the 4-cell stage (23 µm) is found in Figure 2 from this paper. We discuss these points in detail below.

**Author response image 5. respfig5:** (**A**) New 4-cell stage data. Images of the division plane in a representative dividing cell at the 4-cell stage reconstructed from 16x1μm z-stacks of an embryo expressing myosin::GFP from an in situ tagged trans-gene (n=16 embryos) imaged). (**B**) Quantification of the per unit length amount of myosin in the ring from the data in A.

When we monitored the constriction rate during the first four divisions of the *C. elegans* embryo in Carvalho et al. 2009, we noticed that the overall constriction rate (e.g. the four-cell division in Figure 2A of Carvalho et al., 2009; note that this is the overall rate of ring constriction, which is different from the per-unit-length constriction rate that we use in the current manuscript) initially appeared constant (dark green points) and then transitioned to a regime in which it progressively decreased (light green points). This transition could be altered in inhibitions that disrupt the microtubule bundles in the spindle midzone, suggesting that it results from contact with the midzone. In our prior work (Carvalho et al., 2009), we crudely defined the transition point by fitting lines to the constant and decreasing rate regions and taking the intersection (as shown for the four-cell stage data in Figure 2A of Carvalho et al., 2009), which was 23 µm at the four-cell stage, as the transition point. Using this definition, the transition point was slightly lower at the one-cell stage—about 18 µm (see Figure 2B of Carvalho et al., 2009). The transition point is the point after which the constriction rate of the ring is noticeably affected by the presence of the midzone. Thus, in the analysis presented here, which is focused on the inherent dynamics of the ring and the relationship of constriction rate to myosin levels, we avoid measurements after this transition.

In response to a prior reviewer request to analyze how our one-cell stage data analyzing per-unit length myosin levels compares with our prior data at the four-cell stage, we acquired new data in the same strain with in situ-tagged myosin::GFP that we used for our one-cell analysis (data in Author response image 5). We did this because the new in situ-tagged strain we are using is superior to the older strain used in our prior work in which myosin::GFP was expressed from a bombardment-inserted transgene under an exogenous promoter. Nevertheless, a side-by-side comparison revealed that our new data is essentially identical to our older data in the pre-transition zone region (compare data in Figure 3—figure supplement 4C). The green regions in the two graphs mark the data prior to the transition point. Note that our new data is normalized by having myosin::GFP fluorescence at 0 fraction of ring closure be 1 and our old data was normalized by the mean fluorescence across all time points being 1.

The 1.3-fold increase in component levels that we describe occurs between ring perimeters of 50 and 25 µm (fraction of closure ~1 to 0.5) prior to the transition point and not when the ring is less than 10 µm in perimeter as the reviewer suggests – we do not analyze the data in this tail region because it represents the transition to abscission, which is why we had not included this region in our prior author response figure. The data from the analyzable region at the four-cell stage (folding in of the ring to transition point; marked in green in Author response image 5 and Figure 3—figure supplement 4C) are well fit by the same exponential equation that describes our one-cell data, consistent with the idea that components may increase in a similar exponential fashion at the four-cell stage as we have shown that they do at the one-cell stage.

Simply speaking, the major conclusions in Carvalho (2009) are inconsistent with those in current manuscript. There are 4 possibilities:a) Divisions in 1-cell stage and in 4-cell stage are different.b) There is no such difference. The data or interpretation in Carvalho (2009) was wrong. The current model is correct.c) There is no such difference. The current model is wrong. The old model was correct.d) There is no such difference. Both the old and new models are wrongThe authors should clarify which is the case. If b) is the case, detailed point-by-point explanations will be essential as to which data/interpretation in the previous paper still stand or don't stand anymore.

The reviewer requests that we clarify the relationship between our current conclusions and model based on our one-cell stage data and the conclusions and model from our prior work (Carvalho et al., 2009). To address this point in the revision, we have added a new figure that relates the division plane photobleaching experiments at the one- and four-cell stages to the spot bleaching experiments at the four-cell stage in our prior work (new Figure 7) and have reworked three additional figures to illustrate how our current model builds on the findings from our prior work (Figure 4, Figure 6, and Figure 8). We have also added a new introductory section to our Discussion (reproduced below) to explain how our new findings affect our thinking on why the per unit-length constriction rate accelerates during ring closure.

“The constriction rate increase during ring closure is accompanied by an increase in the amount of myosin and other contractile ring components

In prior work (Carvalho et al., 2009), we found that once components are incorporated into the contractile ring, they did not exchange with subunits in the cytoplasm, but were instead lost via constriction-coupled disassembly. […] This finding suggests that the per-unit-length increase in the concentration of myosin and other contractile ring components underlies the acceleration of the constriction rate, and argues against the contractile unit model, which we had previously proposed to explain the per-unit-length constriction rate increase without a proportional increase in component levels (Carvalho et al., 2009).”

To distinguish between these possibilities, repeating latrunculin A treatment during furrow constriction at 1-cell and 4-cell stages using eggshell permeabilization by perm-1(RNAi) would be highly informative. In Carvalho (2009), insensitivity of the 4-cell division to latrunculin A added during furrow ingression provided a support for disregarding the importance of turnover.(Editors’ recommendation: Perform the LatA treatment experiment as well as clearly state what exactly your position is in terms of this manuscript vs. the Carvalho manuscript)

It is not clear why repeating this experiment would be informative. We performed this experiment previously (Carvalho et al., 2009) and subsequent work employing a temperature-sensitive mutant in the cytokinesis formin at the one-cell stage (PMID: 25073157) has also yielded the same result.

12) Inconsistency between the scheme in Figure 4 and their own observation (Naive cortex?).The model doesn't match with what was observed by the authors. It is ambiguous what the 'naive cortex' is. In the scheme in Figure 4A left box, it is placed between the equatorial Rho zone and the polar cortexes and treated as empty boxes without myosin. The only route of recruitment of myosin to the cortex is the direct recruitment of myosin II from the cytoplasm in the Rho zone. However, this picture is not consistent with their own observations.In Figures 1 and 3, they observed the flow of myosin II in the cortical regions at the surface of the embryos, which largely correspond to the regions marked in grey in Figure 4 (and the green Rho zone before furrow ingression starts). Myosin DOES exist in these regions of "naive cortex". The authors may argue that myosin in these regions is inactive. However, at the beginning of furrow ingression (Figure 3, t/tt_ck_ -0.1 and Figure 1t/t_ck_ 0.2 'bottom'), the gradients are observed in regions wider than the ~10 µm wide 'Rho zone'. In later stages, the authors attribute the velocity gradients in the regions flanking the furrow to a projection artifact (dotted segments in Figure1) although it is not very clear how convincing this interpretation is. In the regions where the flow is represented by a solid line, the velocity is largely constant. However, an absence of compression doesn't necessarily mean that myosin is inactive. The tension generated by myosin might just be balanced. Indeed, laser ablation caused outward flows in all the directions (not limited to the direction towards the furrow), indicating that the 'naive cortex' is under active tension although it would be lower than that in the Rho zone.(Editors recommendation: rewrite significantly for clarity)

Myosin foci, which we use as fiduciary marks to monitor cortical movements are found on the entire cortex. However, as highlighted in the first round of review by our other reviewer there is substantially more myosin inside the Rho zone than outside the Rho zone (see examples in Author response image 6). Prior work has also suggested that at anaphase onset, the myosin in the Rho zone is recruited directly to the cortex from the cytoplasm downstream of RhoA-based signaling (PMID: 19720876; PMID: 17959823; PMID:11448996). Thus, in response to the strong preference of reviewer #1 that we align our mathematical model with the most up-to-date view of Rho A-based signaling, we changed how we depict our model so that myosin is loaded onto the cortex upon entry into the Rho zone at a concentration = m_rho_. That said, the only thing that matters in our model equations is the concentration of myosin on the cortex at the point where the cortex enters the Rho zone/contractile ring. Whether the myosin is already on the cortex when it enters the Rho zone as we had suggested in the original version of our model, when we called this constant m_cort_, or is loaded onto the cortex after it enters the ring, as in the current version of the model (m_rho_), does not alter the model equations. We agree with reviewer #1 that the updated version of the model makes more biological sense in terms of the current views on RhoA signaling. Setting the model up this way with myosin (and other contractile ring components) being loaded onto the cortex upon entry to the Rho zone, required us to have a name for the cortex outside the Rho zone that has not yet been subjected to RhoA-based signaling. We therefore decided to call this cortex “naive cortex”.

We do not think that the myosin on the naive cortex outside of the Rho zone/contractile ring is inactive or that the lack of compression means a lack of actin myosin. As the reviewer highlights the cortex outside of the ring is under tension and generation of this tension likely depends on cortical myosin. We suspect that the fact that the cortex in the Rho zone/ring is compressed and the cortex outside the ring is not is related to the fact that the properties of the myosin on the naive cortex are different than the properties of the myosin in the ring. The myosin on the cortex has been shown to turn over rapidly (t_1/2_~30s; PMID: 20852613), whereas our experiments suggest that the myosin in the ring turns over very slowly or not at all, which may simply reflect the degree of cross-linking within the ring where you have high concentrations of myosin, forminnucleated actin, septins and anillin, which interact with each other to form cross-linked meshwork.

With respect to the reviewer’s suggestion that a gradient of flow velocity is observed across a region wider than the 10 µm wide Rho zone, for example at t/t_ck_= -0.1, which we show in Figure 3A (and in Author response image 6). It is not clear to us why the reviewer says this. The velocity plot has a linear slope in the central 10 µm-wide region indicating a zone of uniform compression and the slope is essentially zero outside of this region indicating flow at constant velocity without compression. This is also true on the bottom side at time points near t/t_ck_=0.2. The 0.2 timepoint in the figure looks like the region with the gradient might extend beyond this region, but inspection of nearby time points in Video 2 suggests that the gradient does indeed fall within the 10 µm wide region from -5 to +5 µm (see grey line in the snapshot from Video 2 at t/t_ck_=0.23 in Author response image 6).

**Author response image 6. respfig6:** (**A**) (left) Spinning disk confocal optics were used to collect a 4x1μm z-series containing the embryo cortex and a maximum intensity projection is shown 220s after nuclear envelope breakdown (NEBD). Reproduced from Figure 6 of Lewellyn et al., 2011. (right) en face view of the division plane reproduced from Figure 3C in our paper. Both illustrate the point that there is much more myosin in the Rho zone than on the adjacent cortex. (**B**) Panel reproduced from Figure 3A in our paper. The equatorial cortex is compressed during contractile ring assembly. Following the onset of spindle-based RhoA signaling, the initial recruitment of contractile ring proteins leads to uniform compression of cortical surface along the axis perpendicular to the forming ring across a 10 μm wide region spanning the cell equator. The surface velocity profile reveals a linear velocity gradient that spans the cell equator)-5 to +5 μm), indicating a uniform zone of cortical compression. Outside of this region velocity is essentially constant. (**C**) Panel reproduced from Video 2. An average cortical flow map was calculated from time lapse imaging of the cell surface in 93 control embryos expressing myosin::GFP. (top) schematic illustrates ring size and position at t/t_CK_= 0.23. The graph plots the magnitude of the component of surface velocity aligned along the anterior-posterior axis for the top (150-180°; black) and bottom (0-30°; grey) regions of the cortex at the t/t_CK_= 0.23.

The gradients observed in the regions of the curves that we mark with dotted lines where the reviewer is “not convinced about our interpretation” definitely represent an artificial gradient due to the fact that the surface goes from flowing in the plane perpendicular to the objective to flowing into the division plane, which is parallel to the objective. Our flow analysis only captures the x and y components of the flow vectors not the z-component. Therefore as the surface turns into the division plane, the x,y component of the velocity decreases even through the magnitude of the velocity does not change.

13) Inconsistency between the mathematical model and the scheme in Figure 4In the schematic in Figure 4, recruitment of myosin from the cytoplasm is depicted as the major route of the increase in the total amount of myosin in the Rho zone. However, in the mathematical model, a key assumption is that the mass of cortical myosin is conserved while it flows (formula 20). There is no term that corresponds to the recruitment from the cytoplasm.(Editors’ recommendation: rewrite significantly for clarity)

Equation 20 does not include a term with myosin recruitment because it describes the compression of the cortex within the Rho zone after it is already saturated with myosin motors. In our mathematical model, myosin is recruited to naive cortex at saturating levels as it enters the Rho zone; the concentration on the boundary of the zone (-w, w) is defined as m_rho_.

14) Lack of the effect of geometry change by furrow constriction, or the mechanism for coupling removal of myosin with furrow constrictionSimply due to the geometry change, even without new recruitment of myosin either via cortical flow or from the cytoplasm, furrow constriction results in an increase of the per-length-amount of myosin if there is no removal of myosin coupled with the disassembly of the contractile units. However, this fundamental fact hasn't been properly incorporated in their mathematical model.

In contrast to the reviewer’s assertion, the fact that furrow constriction results in an increase of the per-unit-length amount of myosin, due to a geometry change, has been properly taken into account in our model. Our division plane photobleaching data at the one-cell stage (Figure 6C) and our prior work at the four-cell stage (Carvalho et al., 2009) show that constriction in the around-the ring direction is coupled to disassembly which results in myosin removal (constriction-coupled disassembly). The revised text and figures now make this point significantly more clear. For example, in the methods where we describe the derivation of our model, we now state:

“Consistent with our division plane photobleaching experiment at the one-cell stage (Figure 6C) and our prior work (Carvalho et al., 2009), we propose that in the around-the-ring direction constriction-coupled disassembly leads to a reduction in ring components in proportion to the reduction in length. […] Changes in myosin levels are therefore determined solely by the rate of flow of naive cortex into the Rho zone along the direction perpendicular to the ring, which can be solved as a one-dimensional problem.”

In other words, one could include a term that increases myosin concentration due to the ring shortening and an equal, but negative, term that decreases myosin concentration due to disassembly. However, we chose to avoid this mathematical redundancy.

The consequence of this process is mentioned and depicted in the right box of the Figure 4A "Ring shortening is coupled to disassembly and does not change the per unit length amount of ring myosin". In Figure 4B, in box 3, under the lead "The per unit length rate of ring disassembly is proportional to the per unit length amount of ring myosin", the formula (24) is shown. However, this formula is about the relationship between the per unit length amount of ring myosin and the per unit length rate of ring constriction (instead of the per unit length rate of ring disassembly). The caption can be true only when the ring disassembly is proportional to the ring constriction. In their model, this is achieved by neglecting the effect of geometry change due to furrow constriction while they are discussing formula 18 to 23, and later on by using M_ring_ as equivalent to the per unit length amount of ring myosin without properly explaining that their model doesn't include a mechanism for coupling constriction and disassembly, which is not trivial and was a key discovery in Carvalho (2009).(Editors’ recommendation: rewrite significantly for clarity)

In our model, we explicitly specify that constriction is coupled to disassembly, consistent with our division plane photobleaching experiment at the one-cell stage (Figure 6C) and our prior work (Carvalho et al., 2009). Therefore, the rate of ring constriction and rate of ring disassembly are interchangeable. To avoid any confusion, we now put much more emphasis on this point, even including it in the name of the model, “Constriction-Coupled Disassembly with Compression Feedback”.

15) The velocity of flow of naive cortex into the Rho zone.The authors' theory predicts a linear relationship between v_flow, the velocity of flow of naive cortex into the Rho zone, and M_ring, the total per-unit-length amount of ring myosin (formula 22). Although, mathematically, v_flow_(t) is the speed of the flow at the boundary of the Rho zone, considering the continuity of the flow at the boundary of the Rho zone and largely uniform flow, it is reasonable to interpret v_flow_(t) as the velocity/speed of flow of naive cortex. In Figure 4, this is indicated by growing arrows labeled "Cortical flow" on the 'naive cortex'. However, the data in Figure 1 and 3 don't show such behavior. Instead, the flow seems to show rapid increase around t/t_ck_~0 and gradually slows down between t/t_ck_>0.2. It will be informative if the top and bottom speeds of cortical flow are plotted against time. Anyway, this pattern is inconsistent with the theoretical prediction.

The reviewer is correct in that it is reasonable to assume that v_flow_(t) is effectively the speed of flow at the boundary of the Rho zone. In our model we assume that cortical surface is compressed within the Rho zone/contractile ring. However, the problem in the reviewer’s logic is that the data in Figure 1 and 3 show the velocity of flow on the embryo surface, not at the boundary of the Rho zone/contractile ring. Throughout cytokinesis, the flux of cortical surface into the division plane is equal to the flux of cortical surface into the contractile ring. Thus, the velocity of flow on the embryo surface times the perimeter of the embryo is equal to the velocity of flow at the boundary of the ring times the perimeter of the ring. The velocity of flow at the boundary of the ring increases exponentially during constriction, but this is balanced by the decrease in the perimeter of the ring so that the flux of cortex into the ring remains roughly constant as the ring constricts (see right graph in Figure 3B). Since the perimeter of the embryo does not change during constriction, the average velocity of flow on the embryo surface therefore remains nearly constant. Thus, the pattern of flow that we observe on the surface is consistent with the prediction of our model.

In this case, their favourite trick to convert a constant quantity into an exponentially increasing one by dividing by the ring radius wouldn't work well since the radius that can be used here is the radius at the boundary of the Rho zone, which only decreases towards 5 µm, instead of the ring radius, which decreases towards 0 µm.(Editors’ recommendation: rewrite significantly for clarity)

The reviewer’s point is that our model predicts exponential velocity at the boundary of the Rho zone (R+w, where R is the radius of the ring and w is the width of the Rho zone) rather than at the central point in the ring (R) where maximum myosin fluorescence is observed. In our measurements we divide by ring perimeter (R) and not the outer boundary perimeter (R + w), because the width of the Rho zone is not straight forward to measure. The question is therefore how much of a difference this makes in our measurements. With respect to what the width of the Rho zone actually is, the best proxy we have to assess this is the distribution of myosin (since compression is proportional to the amount of myosin). Looking at the images in Figure 3C, we can visually assess the difference between the perimeter of the ring (red dashed circles) and the approximate boundary of the Rho zone (green dashed circles). Note that we only make measurements prior to the point where the ring comes into contact with the midzone (t/t_ck_~ 0.75). This assessment suggests that for t/t_ck_<=0.62, the ratio of radius of the boundary of the Rho zone (R_Rho zone_) to the radius of the ring (R_Ring_) is between 1.07 and 1.13. For t/t_ck_> 0.7 this ratio potentially becomes somewhat larger (Figure 3C), however this is at the very edge of our measurement zone (cortical compression panel in Figure 3C). Thus, we do not anticipate that the fact that we divide by R_Ring_, which is an easily measurable quantity versus R_Rho_ zone is going to have a large impact on our results.

16) 'Rho zone'.It is not clear what exactly the 'Rho zone' is especially after the furrow has deepened (the distance from the embryo surface to the contractile ring is larger than 5 µm). In the mathematical model, they assume that w is a constant. However, this is not realistic. The actual width of the distribution of active Rho in the cell is likely to be broader in the beginning and become narrower. The word "ring" stands for the contractile ring in most places while in some places it refers to a broader zone used for quantifying myosin (e.g. Figure 6A, the zone between the two boundaries marked with dotted lines). In the latter case, the 'ring' largely overlaps with the 'Rho zone' but not in the former case.(Editors’ recommendation: rewrite significantly for clarity)

We do not specify that w (which is the half width of the Rho zone) is a constant. Our model simply says that the rate of increase of the amount of myosin inside the Rho zone is proportional to the amount of myosin in the Rho zone. For simplicity, we assume that the contractile ring and the Rho zone are the same thing, a point that we have clarified in the text. As discussed above, the best proxy we have to assess the width of the Rho zone is the distribution of myosin. Looking at the images in Figure 3C, suggests that the Rho zone starts out quite narrow, and then increases a bit in width during constriction.

17) Feedback?The authors argue that exponential increase suggests a positive feedback. However, this is not necessarily true (even if their interpretation of exponential increase were true). For example, under an optimal condition, bacteria grow exponentially. Usually, this is not explained by a positive feedback loop. To confirm a feedback loop, an experiment to perturb a key step in the loop should be performed.(Editors’ recommendation: rewrite significantly for clarity or perform an experiment)

Exponential growth does imply positive feedback. Bacteria do indeed grow exponentially as the Reviewer highlights prior to limiting conditions in the media. However, population growth is a classic example of an exponential increase due to positive feedback. More population leads to more births and more births leads to more population. For example, see Chapter 2 of Human Ecology – Basics Concepts for Sustainable Development by Gerald G. Martin. It highlights the fact that positive feedback underlies exponential population growth.

18) "cortical surface".I support reviewer #2's original point about the terms "cortical surface" or "cortical surface area". The authors' rebuttal is not convincing. In 50's, the current concepts of "plasma membrane/cell membrane" and "cell cortex" were not established yet. The fluid mosaic model was established in 70's. The "membrane" in Swann and Mitchision (1958) refers to a combination of the lipid bilayer and the underlining cytoskeletal network.The markers listed are all added from the medium and attached on the cell surface (except for Dan's pigmented granules). The relations between these surface markers and the cortical cytoskeletal network haven't been clarified and can be variable. The expansion (an increase of the distance between the markers) can be caused between the markers that are not anchored to the cortical cytoskeleton by insertion of new membrane lipid bilayer. If markers are somehow anchored to the cortical cytoskeleton, the expansion can also be caused by relaxation of the cytoskeletal network (or radial pull by the neighboring cytoskeletons).The terms "cortical surface" and "cortical surface area" are confusing. The latter has been widely used to describe the geometry of brains. The usage of it in the context of cytokinesis seems to be a recent invention by the authors. At least, this word doesn't appear in Dan (1954) nor in Swann and Mitchision (1958) although they used "cortical layer", "cortical gel" etc.In most of the cases in this manuscript, just "cell cortex" instead of "cortical surface" seems to be appropriate.# an R script to plot unbleached myosin based on the fitted formulas in Figure 6Ct <- (0:100)/100y1 <- 0.22*exp(2.8*t)+0.78y2 <- 0.24*exp(2.8*t)-0.07y3 <- y1-y2quartz(width=4, height=4.5)plot(t, y3)plot(0,0, xlim=c(0,1), ylim=c(0,4.5), type="n", xlab="time", ylab="signal per length", xaxs="i", yaxs="i")lines(t, y1, col='magenta')lines(t, y2, col='green')lines(t, y3)

We disagree with the reviewer on this point. Although in some cases the ways in which the original authors used different terms was not clear, as is often the case as concepts evolve in science, the nature of their experiments is clear. Thus, we can now clearly interpret these experiments in the context of the large body of work and concepts that came after them. Given the confusion highlighted by reviewer #1, we think it is best to use the term cortical surface area because the surface area of the cortex is the property that we directly measure. This avoids the confusion that reviewer #1 has struggled with between the mass of the molecular components of the cortex and how much surface area the cortex occupies. Whereas the term cell cortex can mean either of these things, cortical surface area is very clear.